# Formulation of Scale Transformation in a Stochastic Data Assimilation Framework

Feng Liu[1, 3] and Xin Li[1, 2, 3]

Key Laboratory of Remote Sensing of Gansu Province, Cold and Arid Regions Environmental and Engineering Research Institute, Chinese Academy of Sciences, Lanzhou, 730000, China
[2]Center for Excellence in Tibetan Plateau Earth Sciences, Chinese Academy of Sciences
[3]University of Chinese Academy of Sciences, Beijing 100049, China

*Correspondence to*: Xin Li (lixin@lzb.ac.cn)

**Abstract:** Understanding the errors caused by spatial scale transformation in Earth observations and simulations requires a rigorous definition of scale. These errors are also an important component of representativeness errors in data assimilation. Several relevant studies have been conducted, but the theory of the scale associated with representativeness errors is still not well developed. We addressed these problems by reformulating the data assimilation framework using measure theory and stochastic calculus. First, measure theory is used to propose that the spatial scale is a Lebesgue measure with respect to the observation footprint or model unit, and the Lebesgue integration by substitution is used to describe the scale transformation. Second, a scale-dependent geophysical variable is defined to consider the heterogeneities and dynamic processes. Finally, the structures of the scale-dependent errors are studied in the Bayesian framework of data assimilation based on stochastic calculus. All the results were presented on the condition that the scale is one-dimensional, and the variations in these errors depend on the differences between scales. This new formulation provides a more general framework to understand the representativeness error in a nonlinear and stochastic sense and is a promising way to address the spatial scale issue.

## 1 Introduction

The spatial scale in Earth observations and simulations refers to the observation footprint or model unit in which a geophysical variable is observed or modelled(scale is used below as an abbreviation for spatial scale). Scale is traditionally defined in terms of distance, which is not adequate both because distance is a one-dimensional quantity while scale generally refers to a two- or three-dimensional space and because the scale may change in a very complicated manner (for example, from an irregular observation footprint to a square observation footprint). Generally, the scale is not explicitly expressed in the dynamics of a geophysical variable, partially because a rigorous definition of scale is difficult to find, except for an intuitive

conception (Goodchild and Proctor, 1997) and certain qualitative classifications of scale (Vereecken et al., 2007). This reflects the complexity of scale and requires a more rigorous mathematical conceptualization of scale.

The scale transformation of a geophysical variable may result in significant errors (Famiglietti et al., 2008; Crow et al., 2012; Gruber et al., 2013; Hakuba et al., 2013; Huang et al., 2016; Li and Liu, 2016; Ran et al., 2016). These errors are mainly caused by the strong spatial heterogeneities (Miralles et al., 2010; Li, 2014) and irregularities (Atkinson and Tate, 2000) that are associated with geophysical variables across different scales, and are also closely related to dynamic variations, e.g., in hydrological (Giménez et al., 1999; Vereecken et al., 2007; Merz et al., 2009; Narsilio, et al. 2009), soil (Ryu and Famiglietti, 2006; Lin et al., 2010) and ecological (Wiens, 1989) processes. How to develop mathematical tools to elucidate the scale transformation has yet to be fully addressed.

Data assimilation could be an ideal tool to explore the scale transformation because it presents a unified and generalized framework in Earth system modelling and observation (Talagrand, 1997). Geophysical data are typically observed by various Earth observations; thus, updating the observation data in a data assimilation system may result in scale transformations between the observation space and system state space. If observation operator is strongly nonlinear and complex, the errors caused by the scale transformation are even more serious (Li, 2014). An important concept that is related to the scale transformation in data assimilation is "representativeness error", which is associated with the inconsistency in the spatial and temporal resolutions between states, observations and operators (Lorenc, 1986; Janjić and Cohn, 2006; van Leeuwen, 2014; Hodyss and Nichols, 2015) and the missing physical information that is related to a numerical operator compared to the ideal operator (van Leeuwen, 2014), such as the discretization of a continuum model or neglect of necessary physical processes. The representativeness error and instrument error make up the observation error of data assimilation. Under the Gaussian assumption, they are independent of each other (Lorenc, 1995; van Leeuwen, 2014). This study will not consider the instrument error when formulating the scale transformation in data assimilation.

Recently, approaches have been developed to assess the representativeness error. Janjić and Cohn (2006) studied the representativeness error by treating system state as the sum of resolved and unresolved portions. Bocquet et al. (2011) used a pair of operators, namely, restriction and prolongation, to connect the relationship between the finest regular scale and a coarse scale, and determined the representativeness error using a multi-scale data assimilation framework. van Leeuwen (2014)

considered two complicated cases, i.e., conducting the observation vector in a finer resolution compared with system state vector and assimilating the retrieved variables. Their solutions were formulated using an agent in observation or state space, and a particle filter was proposed to treat the nonlinear relationship between observations, states and retrieved values. Hodyss and Nichols (2015) also estimated the representativeness error by investigating the difference between the truth and the inaccurate value that is generated by forecasting model.

Although these approaches explored the structure of the representativeness error and offered various solutions, improvements are still necessary to investigate the exact expression of the errors caused by scale transformation in data assimilation. The authors believe that these approaches are optimal in linear systems but may not be suitable when observations are heterogeneous and sparse, or when operators are nonlinear between states and observations, although the general equations in the nonlinear case were given. Without taking heterogeneities and nonlinear operators into account, the representativeness error cannot be fully understood. However, heterogeneity varies depending on the situation and is difficult to formulate in a general theoretical study.

Data assimilation studies based on stochastic processes (Apte et al., 2007; Miller, 2007) or a stochastic dynamic model (Miller et al., 1999; Eyink et al., 2004) have been proposed recently. Compared to deterministic models, stochastic data assimilation is more applicable in an integrated and time-continuous theoretical study (Bocquet et al., 2010) and creates an infinite sampling space of the system state (Apte et al., 2007). Although the theorems of calculus that are based on stochastic processes (or stochastic calculus) are different from those of ordinary calculus, these advantages suggest that stochastic data assimilation offers a more general framework to study scale transformation.

We attempt to explore the mathematic definitions of scale and scale transformation and then formulate the errors caused by the scale transformation in a general theoretical study on stochastic data assimilation. The next section introduces the basic concepts and theorems of measure theory, stochastic calculus and data assimilation. In Sect. 3, we present the definitions of scale and scale transformation. The posterior probability of system state is also reformulated by scale transformation in a stochastic data assimilation framework. In the final section, the contributions and deficiencies of this study are discussed.

## 2 Basic knowledge

The scale greatly depends on the geometric features of a certain observation footprint or model unit. The model unit is a specified subspace where a geophysical variable evolves in the model space; it could be a point, a rectangular grid, or an irregular unit such as a response unit (watershed, landscape patch, etc.). We offer a solution in which the definition of scale

uses measure theory and the expression of a geophysical variable as a stochastic process uses stochastic calculus. Therefore, we first introduce several basic concepts of measure theory and stochastic calculus.

## 2.1 Measure theory

Let $\Omega$ be an arbitrary non-empty space. $\mathcal{F}$ is a **σ-algebra** (or **σ-field**) of subsets of $\Omega$ that satisfies the following conditions:

(i)  $\Omega \in \mathcal{F}$, and the empty set $\Phi \in \mathcal{F}$;

(ii)  $A \in \mathcal{F}$ implies that its complementary set $A^c \in \mathcal{F}$;

(iii)  $A_1, A_2, \cdots \in \mathcal{F}$ implies their union $A_1 \cup A_2 \cup \cdots \in \mathcal{F}$.

A set function $\mu$ of $\mathcal{F}$ is called a **measure** if it satisfies the following conditions:

(1)  $\mu(A) \in [0, \infty)$ and $\mu(\Phi) = 0$;

(2)  If $A_1, A_2, \cdots \in \mathcal{F}$ is any disjoint sequence and $\bigcup_{k=1}^{\infty} A_k \in \mathcal{F}$, $\mu$ is countably additive such that $\mu\left(\bigcup_{k=1}^{\infty} A_k\right) = \sum_{k=1}^{\infty} \mu(A_k)$.

If $\mu(\Omega) = 1$, $\mu$ can be replaced by the probability measure $p$, and if $\mu$ is finite, $p$ can be calculated as $p(A) = \mu(A)/\mu(\Omega)$. The triples $(\Omega, \mathcal{F}, \mu)$ and $(\Omega, \mathcal{F}, p)$ are the **measure space** and **probability measure space**, respectively.

Let $\Omega$ be the set of real numbers $R$ and σ-algebra $\mathcal{B}$ be **Borel algebra**, which is generated by all closed intervals in $R$. Then, $\forall A = [a, b] \in B$, a **Lebesgue measure** on $R$ is defined as $I(A) = b - a$. Intuitively, the Lebesgue measure on $R$ coincides with the length.

An **n-dimensional Lebesgue volume** is defined to measure the standard volumes of the subsets in $R^n$ based on $I^n(A) = \prod_{k=1}^{n}(b_k - a_k)$, where $A = [x : a_k \leq x_k \leq b_k, k = 1, 2, \cdots, n]$ is an n-dimensional regular cell in $R^n$. The n-dimensional Lebesgue volume is an ordinary volume, such as length (n=1), area (n=2) and volume (n=3).

Next, the **outer measure** is defined as $m^n(A) = \inf\{\sum_{i=1}^{+\infty} I^n(A_i)\}$, where $\inf\{\cdot\}$ is the infimum, $A_i = [x : a_{i,k} \leq x_k \leq b_{i,k}, k = 1, 2, \cdots, n]$ is the n-dimensional regular cell in $R^n$, and $A \subseteq \bigcup_{i=1}^{+\infty} A_i$. Thus, if $A$ is any subset of $R^n$, one can collect many sets of n-dimensional regular cells $\{A_i\}$ to cover $A$. Among them, the outer measure denotes the set whose union has the smallest n-dimensional Lebesgue volume.

To match the two conditions of a measure, one can define the outer measure $m^n$ as a **Lebesgue measure** on measure spaces $(R^n, \mathcal{L}^n, m^n)$, where $\mathcal{L}^n$ is the **Lebesgue σ-algebra** of $R^n$. The construction of the Lebesgue σ-algebra is based on the Caratheodory condition (Bartle, 1995, definition 13.3). Fortunately, almost all of the observation footprints and model units are finite and closed, leading them to be Lebesgue measurable. This consequently ensures that the Lebesgue measure $m^n$ is a measure and the triple $(R^n, \mathcal{L}^n, m^n)$ is a measure space. The Lebesgue measure of a Lebesgue measurable subset in $R^n$ also coincides with its volume.

The n-dimensional **Lebesgue integral** in $(R^n, \mathcal{L}^n, m^n)$ is $\int f dm^n$, where $f$ is a real function on $R^n$. The Lebesgue integral can be further denoted by $\int f dm^n = \int f(x) dx$, where $x \in R^n$ and $x = (x_1, \cdots, x_n)$.

In the two-dimensional case ($n = 2$), the Lebesgue integral is

$$\iint_A f(x_1, x_2) dx_1 dx_2,$$

where $A \in \mathcal{L}^2$. Next, we consider the **Lebesgue integration by substitution** on $R^2$. Let $T(x_1, x_2) = [t_1(x_1, x_2), t_2(x_1, x_2)] = [y_1, y_2]$ be a one-to-one mapping of a subset $X$ onto another subset $Y$ on $R^2$. Assuming that $T$ is continuous and has a continuous partial derivative matrix $T_x = \begin{pmatrix} \partial t_1 / \partial x_1 & \partial t_1 / \partial x_2 \\ \partial t_2 / \partial x_1 & \partial t_2 / \partial x_2 \end{pmatrix}$, then

$$\iint_Y f(y_1, y_2) dy_1 dy_2 = \iint_X f(T(x_1, x_2)) |J(x_1, x_2)| dx_1 dx_2,$$

where the Jacobian determinant $|J(x_1, x_2)| = |\det T_x| = \begin{vmatrix} \partial t_1 / \partial x_1 & \partial t_1 / \partial x_2 \\ \partial t_2 / \partial x_1 & \partial t_2 / \partial x_2 \end{vmatrix}$. If $T$ is linear, the integral reduces to

$$\iint_Y f(y_1, y_2) dy_1 dy_2 = |J(x_1, x_2)| \iint_X f(T(x_1, x_2)) dx_1 dx_2.$$

By doing so, any observation footprint or model unit can be regarded as a Lebesgue measurable subset in a two-dimensional space $R^2$.

Additional details regarding measure theory can be found in the literature (for example, Billingsley, 1986; Bartle, 1995).

## 2.2 Stochastic calculus

We then introduce some necessary concepts and theorems of stochastic calculus. All the classic theorems are introduced without proofs; their detailed derivations can be found in the literature (Itô, 1944; Karatzas et al., 1991; Shreve, 2005).

**Stochastic calculus** is defined for ordinary integrals with respect to stochastic processes. One of the simplest stochastic processes is **Brownian motion**. The Brownian motion $W$ that is defined on a probability measure space $(\Omega, F, p)$ is characterized as follows:

1) $W(0) = 0$.

2) $\forall t_1 > s_1 \geq t_2 > s_2 > 0$, the increments $W(t_1) - W(s_1)$ and $W(t_2) - W(s_2)$ are independent.

3) $\forall t > s \geq 0, W(t) - W(s) \sim N(0, t - s)$.

The last two conditions represent that $\forall t_2 > s_2 > t_1 > s_1 \geq 0$, $W(t_2) - W(s_2)$ and $W(t_1) - W(s_1)$ are independent Gaussian random variables. Additionally, $W$ is related to the probability measure $p$.

Stochastic calculus based on Brownian motion produces an **Ito process**. The differential form of the time-dependent Ito process is

$$dI = \varphi(t)dt + \sigma(t)dW(t), \qquad (1)$$

where $\varphi(t)$, $\sigma(t)$ and $W(t)$ are the drift rate, volatility rate and Brownian motion, respectively. The integral form of Eq. (1) is

$$I(t) = I(0) + \int_0^t \varphi(u)du + \int_0^t \sigma(u)dW(u). \qquad (2)$$

**Theorem 1**: For any Ito process defined as in Eq. (1), the **quadratic variation** that is accumulated on the interval $[0, t]$ is

$$[I, I](t) = \int_0^t \sigma^2(u)du, \qquad (3)$$

and the **drift** of Eq. (1) is $I(0) + \int_0^t \varphi(u)du$.

As distinguishing features of stochastic calculus, the quadratic variation and drift can be regarded as stochastic versions of the variance and expectation, respectively. That is, the variance and expectation are instances of their random variable stochastic counterparts within a certain integral path. Therefore, rather than being constants, the quadratic variation and drift are given in terms of probability.

**Theorem 2 (Ito's Lemma)**: If the partial derivatives of function $f(u, x)$, viz. $f_u(u, x)$, $f_x(u, x)$ and $f_{xx}(u, x)$, are defined and continuous. If $t \geq 0$, we have

$$f(t, x(t)) = f(0, x(0)) + \int_0^t f_u(u, x(u))du + \int_0^t f_x(u, x(u))\sigma(u)dW(u) + \int_0^t f_x(u, x(u))\varphi(u)du +$$

$$\frac{1}{2}\int_0^t f_{xx}(u, x(u))\sigma^2(u)du. \qquad (4)$$

Ito's Lemma is typically used to build the differential of a stochastic model with Ito processes. In this study, Ito's Lemma is applied to study the scale-dependent relationship between the observation and state and the errors caused by scale transformation.

**2.3 Traditional formulation of data assimilation in the Bayesian theorem framework**

We use the well-accepted Bayesian theorem of data assimilation (Lorenc, 1995; van Leeuwen, 2015) to investigate its time- and scale-dependent errors. State and observation are first assumed to be one-dimensional.

A nonlinear forecasting system can be described by

$$X(t_k) = M_{k-1:k}\big(X(t_{k-1})\big) + \eta(t_k), \qquad (5)$$

where $M_{k-1:k}(\cdot)$, $X(t_k)$ and $\eta(t_k)$ represent a nonlinear forecasting operator that transits the state from the discrete time $k - 1$ to $k$, the state with prior probability distribution function (PDF) $p(X)$, and the model error at time $k$, respectively.

If a new observation is available at time $k$, the observation system is given by

$$Y^o(t_k) = H_k\big(X(t_k)\big) + \varepsilon(t_k), \qquad (6)$$

where $H_k(\cdot)$, $Y^o(t_k)$ and $\varepsilon(t_k)$ represent the nonlinear observation operator, true observation with prior PDF $p(Y)$, and observation error at time $k$, respectively.

Previous studies (e.g., Janjić and Cohn, 2006; Bocquet et al. 2011) described the origins of the components of $\varepsilon(t_k)$ and $\eta(t_k)$, such as white noise, the discretization error of a continuum model, the errors that are caused by missing physical processes, and the scale-dependent bias. In this study, we assume that both forecasting and observation operators are perfect models; thus, errors caused by missing physical processes are discarded.

According to Bayesian theory, the posterior PDF of the state based on the addition of a new observation into the system is

$$p(X|Y) = p(Y|X)p(X)/p(Y), \qquad (7)$$

where $p(X|Y)$ is the posterior PDF that presents the PDF value of state $X$ given an available observation $Y$. $p(Y|X)$ is a likelihood function, which is the probability that an observation is $Y$ given a state $X$. $p(X)$ and $p(Y)$ are the prior PDF values of the state and observation, respectively. Here, $p(X)$ is supposed to be known and $p(Y)$ is a normalization constant (van Leeuwen, 2014). The aim of data assimilation is equivalent to finding the posterior PDF $p(X|Y)$.

## 3 Reformulation of scale transformation in data assimilation framework

### 3.1 Definition of scale

We define the scale based on the measure theory that was introduced in Sect. 2. The relationship between Lebesgue measure in $(R^2, \mathcal{L}^2, m^2)$ and scale is first introduced by the following measures of Earth observations.

(i)   Measure of a single-point observation: When the observation footprint is very small and homogeneous, we assume that its footprint approaches zero, and its measure is accordingly zero under the condition of the Lebesgue measure.

(ii)   Measure along a line: The measure is a one-dimensional Lebesgue measure.

(iii)   Measure of a rectangular pixel (for example, remote sensing observation): $\forall A = [x: a_k \leq x_k \leq b_k, \mathrm{k} = 1,2]$, it is a two-dimensional Lebesgue volume, i.e., $\mu_{iii}(A) = I^2(A) = \prod_{k=1}^{2}(b_k - a_k)$.

(iv) Measure of a footprint-scale observation: The footprint is any bounded closed domain $A$, which is not necessary to be regular rectangles, but can also be circles or ellipses. We use Lebesgue measure on $R^2$, i.e., $\mu_{iv}(A) = m^2(A) = \inf\left\{\sum_{i=1}^{+\infty} I^2(A_i)\right\}$, where $A_i = [x: a_{i,k} \leq x_k \leq b_{i,k}, k = 1,2]$ and $A \subseteq \bigcup_{i=1}^{+\infty} A_i$. Clearly, measures (i)~(iii) are special cases of the measure of a footprint-scale observation.

All of the above measures depend mainly on the shape and size of $A$. The Lebesgue measure on $R^2$ coincides with the area; thus, the Lebesgue integral of $\mu_{iv}(A)$ is $\iint_A dx_1 dx_2$, where the real function $f \equiv 1$.

Now, we can generalize the above examples by defining the **scale** as the Lebesgue measure with respect to the observation footprint. This definition can also be extended to a certain model unit. Thus, for any subset $A \in \mathcal{L}^2$, the scale is $s = m^2(A) = \iint_A dx_1 dx_2$, where the real function $f \equiv 1$. From a geometric perspective, the measure function $m^2(\cdot)$ refers to the shape of the subset, and the scale further indicates its size.

We represent the scale as $s$, and let $s_0 = m_0^2(A_0) = \iint_{A_0} dx_1 dx_2 = 1$ be the **standard scale**, where $A_0 = [x: 0 \leq x_k \leq 1, k = 1,2]$ is the unit square in $R^2$. The standard scale can be regarded as a basic unit of scale. It presents a standard reference by which one can make a quantitative comparison between different scales. The standard scale is also the origin of scales that lets scales vary similarly to other physical quantities, such as time.

We can further define **scale transformation**. For $\forall A_1, A_2 \in \mathcal{L}^2$, if there are two different scales, $s_1 = m^2(A_1) = \iint_{A_1} dx_1 dx_2$ and $s_2 = m^2(A_2) = \iint_{A_2} dy_1 dy_2$, then we can obtain $s_2 = \iint_{A_2} dy_1 dy_2 = \iint_{A_1} |J(x_1, x_2)| dx_1 dx_2$ based on Lebesgue integration by substitution, where the Jacobian matrix $J(x_1, x_2)$ represents the geometric transformation from $A_1$ to $A_2$. In particular, if $J(x_1, x_2) = diag(\xi, \xi), \xi \in R$, which also indicates that the geometric transformation is linear, then the following expression is valid based on Lebesgue integration by substitution:

$$s_2 = |J(x_1, x_2)| \iint_{A_1} dx_1 dx_2 = \xi^2 s_1, \quad (8)$$

where $s_1$ and $s_2$ represent the change of the **one-dimensional rule**.

If two scales follow the one-dimensional rule, they are geometrically similar. This rule simplifies scale as a one-dimensional variable that corresponds to the scale transformations between most remote sensing images with various spatial resolutions. For example, $\forall A = [x: a \leq x_k \leq b, k = 1,2]$, where $A$ and the unit square $A_0$ are geometrically similar, and the scale $s =$

$\mu_{iii}(A)$ can be expressed by the one-dimensional rule of scale transformation: $s = \mu_{iii}(A) = |J(x_1, x_2)| \iint_{A_0} dx_1 dx_2 =$

$(b - a)^2 s_0$. For another example, let $s = \iint_A dy_1 dy_2$ be the scale of a disc footprint $A$ with radius $r$. The mapping function

between $A$ and $A_0$ is $T(x_1, x_2) = [rx_1 \cos(2\pi x_2), rx_1 \sin(2\pi x_2); 0 \le x_1 \le 1, 0 \le x_2 \le 1] = [y_1, y_2]$, and the Jacobian

determinant $\quad |J(x_1, x_2)| = \begin{vmatrix} r\cos(2\pi x_2) & -2\pi r x_1 \sin(2\pi x_2) \\ r\sin(2\pi x_2) & 2\pi r x_1 \cos(2\pi x_2) \end{vmatrix} = 2\pi r^2 x_1 \quad . \quad$ Therefore, $\quad s = \iint_A dy_1 dy_2 =$

$\iint_{A_0} |J(x_1, x_2)| dx_1 dx_2 = \pi r^2 s_0$, which is equal to its area. However, $s_0$ and $s$ do not obey the one-dimensional rule because

the Jacobian matrix is not diagonal.

Layer 1 in Figure 1 shows the relationship between the Lebesgue measure and scale. The measure space $\Omega =$

$[x: 0 \le x_k \le 4, k = 1,2]$ is regularly divided by the unit square $A_0$. Let scales $s_{C1} = m_{C1}^2(C1)$, $s_{C2} = m_{C2}^2(C2)$ and $s_{C3} =$

$m_{C3}^2(C3)$ be the Lebesgue measures of disc observation footprints $C1, C2$ and $C3$, respectively. Then, $m_{C1}^2(\cdot) = m_{C2}^2(\cdot) =$

$m_{C3}^2(\cdot)$ because they are the same Lebesgue measure functions. That is, if $\{A_i\}$ is the set with the smallest volume that

covers $C1$, then similar sets $\{A_i + 2\}$ and $\{A_i \times 3 + 2\}$ can be used (with the origin located in the upper-left corner) to cover $C_3$

and $C_2$ with the smallest volumes, respectively. Here, $A_i + 2 = [x_i: x_{i,k} + 2, x_{i,k} \in A_i, k = 1,2]$ and $A_i \times 3 + 2 = [x_i: x_{i,k} \times$

$3 + 2, x_{i,k} \in A_i, k = 1,2]$, which proves that functions $m_{C1}^2(\cdot)$, $m_{C2}^2(\cdot)$ and $m_{C3}^2(\cdot)$ collect the desired set based on the same

scheme; therefore, they are identical. Additionally, $s_{C2} = m_{C2}^2(C2) = \sum I^2(A_i \times 3 + 2)$ is much larger than $s_{C1} = m_{C1}^2(C1) =$

$\sum I^2(A_i)$ and $s_{C3} = m_{C3}^2(C3) = \sum I^2(A_i + 2)$. Therefore, the scale of $C2$ is not equal to the two other scales because the

volumes of their subsets are different. However, their scales are governed by one-dimensional rules because their measures

are identical and the Jacobian matrices between them are diagonal.

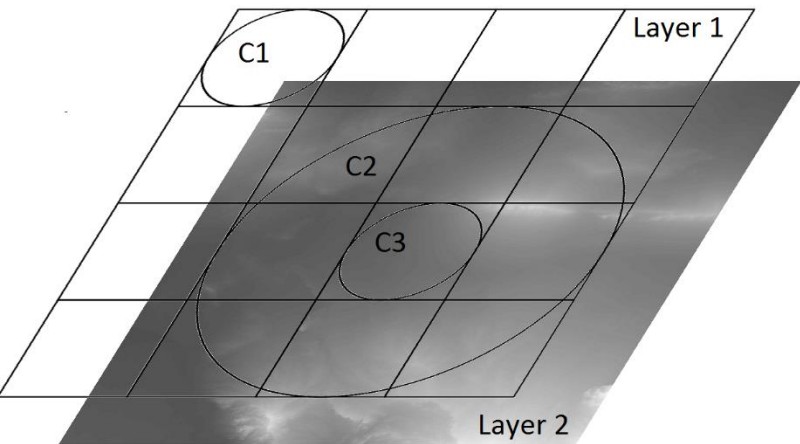

Figure 1. Diagram of the relationships among a Lebesgue measure, scale and geophysical variable

## 3.2 Stochastic variables in data assimilation

Instead of using Eqs. (5) and (6), which are discrete in time, we use Ito process-formed expressions with the one-dimensional
infinitesimals $ds$ and $dt$ to formulate a continuous-time (or continuous-scale) state and observation.

A geophysical variable can be regarded as a real function $V(s,t)$, and it maps the space $(R^2, \mathcal{L}^2, m^2)$ onto $R$, where $s$ is the
scale, $s = m^2(A), A \in \mathcal{L}^2$, and $t$ is the time. In n-dimensional data assimilation, a geophysical variable $V$ is related to an
element of state vector $X$ at a specific scale $s$ and time $t$.

In Figure 1, Layer 2 presents a heterogeneous geophysical variable in the entire region. If we aggregate Layer 2 into Layer
1 and let each pixel intensity be the value for a geophysical variable in that pixel, then the measure space $\Omega$ is heterogeneous.
A geophysical variable represents a spatial average in a specific observation footprint with a specific scale. Therefore, the
geophysical variables in $C1$ and $C3$ are not equal because their observation footprints are different, and the geophysical
variables in $C2$ and $C3$ are also different because the scale changes. The former introduces that the geophysical variables vary
with the location, and the latter states that the geophysical variables are scale-dependent.

If the statistical properties of the geophysical variable are available, we can construct an explicit stochastic equation for it.
We introduce the time-dependent Ito process Eq. (1) to define the geophysical variable process:

$$dV = p(t)dt + q(t)dW(t). \qquad (9)$$

Similarly, the geophysical variable is supposed to evolve via a stochastic process, for which the dynamic process and
uncertainty are allowed to vary with scale,

$$dV = \varphi(s)ds + \sigma(s)dW(s), \qquad (10)$$

where $\varphi(s)$ and $\sigma(s)$ are the scale-based drift rate and volatility rate, respectively. The geophysical variable is a probabilistic
process with respect to scale and thus has scale-dependent errors, where the scale should shift forward or backward based on
the condition that the scale follows the one-dimensional rule.

Eq. (9) can be regarded as a continuous-time version of Eq. (5), i.e., the estimation of the state is equal to the integral of Eq. (9) over a time interval. Here, $p(t)$ indicates the physical process with respect to time, and $q(t)$ is the error only caused by the evolution of time; thus, model error $\eta$ in Eq. (5) contains more parts than $q(t)$. Eq. (10) implies that the value and variance of a geophysical variable may change if the scale changes. The formulation of $\varphi(s)$ should consider the spatial heterogeneities and physical process variations among different scales, which together constitute the deterministic part of a geophysical variable. However, neither of them is well understood in a general theoretical study. Therefore, $\varphi(s)$ is conceptualized in Eq. (10). Particularly, if the study region is homogeneous, then the values of a variable that are observed at the same place are identical between the large scale and fine scale, and $\varphi(s)$ can be left out. Due to the integral over the space of Brownian motion, $\sigma(s)$ is the stochastic part, meaning that scale transformation produces uncertainties. The formulation of $\varphi(s)$ should consider the spatial heterogeneities and physical process variations among different scales, which together constitute the deterministic part of a geophysical variable. However, neither of them is well understood in a general theoretical study. Therefore, $\varphi(s)$ is conceptualized in Eq. (10). Particularly, if the study region is homogeneous, then the values of a variable that are observed at the same place are identical between the large scale and fine scale, and $\varphi(s)$ can be left out. Due to the integral over the space of Brownian motion, $\sigma(s)$ is the stochastic part, meaning that scale transformation produces uncertainties.

The state in the forecasting step can be expressed by Eq. (9) because only time is involved. In the analysis step of data assimilation, the state does not pertain to time, and we assume that the scale has a quantifiable effect on the errors in this step; thus, both the states and observations can be defined by Eq. (10).

## 3.3 Expression of scale transformation in a stochastic data assimilation framework

First, we provide the following lemma.

**Lemma 1**: For $\forall s_0 > 0$, let $W^*(0) = W(s_0) - W(s_0), \dots, W^*(s) = W(s_0 + s) - W(s_0)$; then, $W^*(s), s \geq 0$ is a Brownian motion.

**Proof.** First, $W^*(0) = W^*(s_0) - W^*(s_0) = 0$. $\forall s_{i+1} > s_i \geq 0, i = 1,2,3, \dots$, $W^*(s_{i+1}) - W^*(s_i) = [W(s_0 + s_{i+1}) - W(s_0)] - [W(s_0 + s_i) - W(s_0)] = W(s_0 + s_{i+1}) - W(s_0 + s_i)$, which suggests that the increments $[W^*(s_{i+1}) - W^*(s_i)]$ are equal to $[W(s_0 + s_{i+1}) - W(s_0 + s_i)]$ and are independent Gaussian distributed. Therefore, $W^*(s), s \geq 0$ is a Brownian motion, with $E[W^*(s_{i+1}) - W^*(s_i)] = 0$ and $Var[W^*(s_{i+1}) - W^*(s_i)] = s_{i+1} - s_i$. Q. E. D.

**Remark on Lemma 1**: Note that in the definition of Brownian motion, the parameter starts at zero. However, the scale is realistically greater than zero, which means that it cannot be directly applied in Brownian motion. Therefore, Lemma 1 is logical because it implies that $W(s), s \geq s_0$ is an equivalent expression of $W^*(s), s \geq 0$. Therefore, beginning with the standard scale, the Brownian motion and stochastic calculus with respect to scale can be further developed.

In the following content, we use Brownian motion with a parameter that starts at $s_0$ to define the scale-dependent geophysical variables; therefore, the classic expressions above are changed. According to Lemma 1, Eq. (3) is given by

$$[I, I](s) = \int_{s_0}^{s} \sigma^2(u)du. \tag{11}$$

Additionally, the integral form of Eq. (10) is

$$V(s) = V_0 + \int_{s_0}^{s} \varphi(u)du + \int_{s_0}^{s} \sigma(u)dW(u), \tag{12}$$

where $V_0 = V(s_0)$, and the drift of Eq. (12) is

$$V_0 + \int_{s_0}^{s} \varphi(u)du. $$

Similarly, Eq. (4) becomes

$$f(s, V(s)) = f(s_0, V(s_0)) + \int_{s_0}^{s} f_u(u, V(u))du + \int_{s_0}^{s} f_x(u, V(u))\sigma(u)dW(u) + \int_{s_0}^{s} f_x(u, V(u))\varphi(u)du +$$

$$\frac{1}{2}\int_{s_0}^{s} f_{xx}(u, V(u))\sigma^2(u)du.$$

Now, we make the following assumptions.

**Assumption 1**: The scale transformations between the state and observation spaces of data assimilation obey the one-dimensional rule as defined in Sect. 3.1.

**Assumption 2**: In the forecasting step, the model unit equals the scale of the state space, and both of them are constant.

**Assumption 3**: In the analysis step, the state, observation and observation operator are scale dependent. Only one observation is added into the data assimilation system at a time.

In assumption 1, the one-dimensional rule ensures that scale changes in a sense of geometrical similarity (for example, from a larger square observation footprint to a smaller square observation footprint, or from $C_2$ to $C_3$ as presented in Figure 1). Therefore, based on assumption 1, scale only varies in one-dimensional space, meaning that the corresponding scale transformation is an integral over one-dimensional space.

Assumption 2 indicates that the model unit and state scale are supposed to be the same and both invariant in space and time. Thus, there is no scale transformation in the forecasting step; Thus, Eq. (9) can adequately describe this step.

Based on assumption 3, the analysis step is related to the scale. According to Eq. (10), the state and observation in the analysis step are

$$dX = \varphi_X(s)ds + \sigma_X(s)dW(s) \tag{13}$$

and

$$dY = \varphi_Y(s)ds + \sigma_Y(s)dW(s), \tag{14}$$

where $\varphi_X(s)$, $\sigma_X(s)$, $\varphi_Y(s)$ and $\sigma_Y(s)$ represent the scale-dependent drift rates and volatility rates of state $X$ and observation $Y$, respectively. $\varphi(s)$ also implies the heterogeneities and physical processes from standard scale to a specific scale, which may be hard to formulate. $\sigma(u)$ can be regarded as the stochastic perturbation with respect to scale. Therefore, according to Eq. (12), a state is $X(s_X) = X_0 + \int_{s_0}^{s_X} \varphi(u)du + \int_{s_0}^{s_X} \sigma(u)dW(u)$ in the state space and $X(s_Y) = X_0 + \int_{s_0}^{s_Y} \varphi(u)du + \int_{s_0}^{s_Y} \sigma(u)dW(u)$ in the observation space. These formulas prove that the value of state varies with the changes of scale.

The scale transformation is only involved in the process of mapping the state vector from state space to observation space.

Based on the above discussion, the integral forms of the state are

$$X(s_X) = X_0 + \int_{s_0}^{s_X} \varphi_X(s)ds + \int_{s_0}^{s_X} \sigma_X(s)dW(s). \tag{15}$$

For the observation, we have

$$Y(s_Y) = Y_0 + \int_{s_0}^{s_Y} \varphi_Y(s)ds + \int_{s_0}^{s_Y} \sigma_Y(s)dW(s) \tag{16}$$

In Eq. (15) and Eq. (16), the time $t$ is omitted, and $s_X, s_Y, X_0$ and $Y_0$ represent the scale of the state space, scale of the observation space, state in $s_0$ and observation in $s_0$, respectively.

The Bayesian equation of data assimilation (Eq. (7)) produces the posterior PDF $p(X|Y)$ that is associated with the likelihood function $p(Y|X)$ and the distributions of the state and observation. Theorem 1 and Eqs. (15)~(16) yield $X \sim N\left(X_0, \int_{s_0}^{s_X} ds\right)$ and $Y \sim N\left(Y_0, \int_{s_0}^{s_Y} ds\right)$ under the condition that the variances exist. In addition, assumption 1 states that the scales vary in one-dimensional space, which results in

$$X \sim N\left(X_0 + \int_{s_0}^{s_X} \varphi_X(s)ds, \int_{s_0}^{s_X} \sigma_X^2(s)ds\right) \tag{17}$$

$$\text{and } Y \sim N\left(Y_0 + \int_{s_0}^{s_Y} \varphi_Y(s)ds, \int_{s_0}^{s_Y} \sigma_Y^2(s)ds\right). \tag{18}$$

Eq. (17) and Eq. (18) are the prior PDFs of state and observation with respect to scale in state space and observation space, respectively. Compared with the PDFs with respect to time, their expectations are equal to the value at the standard scale, and the variances depend on the differences between the standard scale and the scale in state or observation space. These two prior PDFs are introduced into the Bayesian theorem that is reformulated by scale.

Then, we calculate the posterior PDF.

The scale-dependent observation operator is $H(s, x)$, which suggests that the observation operator and its parameters are both susceptible to the scale. If $H(s, x)$ is defined, its continuous partial derivatives are $H_s(s, x), H_x(s, x)$ and $H_{xx}(s, x)$. In line with Ito's Lemma, we get an estimation of observation in the observation space, which is related to the state $X(s_X)$ defined in the state space

$$H\big(s_X, X(s_X)\big) = H(s_0, X_0) + \int_{s_0}^{s_X} H_s\big(u, X(u)\big)\,du + \int_{s_0}^{s_X} H_x\big(u, X(u)\big)\,dW(u) + \frac{1}{2}\int_{s_0}^{s_X} H_{xx}\big(u, X(u)\big)\,du$$

$$= H(s_0, X_0) + \int_{s_0}^{s_X}\left[H_s\big(u, X(u)\big) + \frac{1}{2}H_{xx}\big(u, X(u)\big)\right]du + \int_{s_0}^{s_X} H_x\big(u, X(u)\big)\,dW(u). \tag{19}$$

Assumption 1 suggests that the observation and state spaces have the same probability measure; thus, the Brownian motions in these two spaces are equivalent. Eq. (19) can also be rewritten by replacing $s_0$ with $s_Y$, namely $H\big(s_X, X(s_X)\big) = H\big(s_Y, X(s_Y)\big) + \int_{s_Y}^{s_X}\left[H_s\big(u, X(u)\big) + \frac{1}{2}H_{xx}\big(u, X(u)\big)\right]du + \int_{s_Y}^{s_X} H_x\big(u, X(u)\big)\,dW(u)$, and then we obtain

$$Y(s_Y) - H\big(s_X, X(s_X)\big)$$

$$= Y(s_Y) - \left[H\big(s_Y, X(s_Y)\big) + \int_{s_Y}^{s_X}\left[H_s\big(u, X(u)\big) + \frac{1}{2}H_{xx}\big(u, X(u)\big)\right]du + \int_{s_Y}^{s_X} H_x\big(u, X(u)\big)\,dW(u)\right]$$

$$= Y(s_Y) - \left[H\big(s_X, X(s_X)\big) + \frac{1}{2}\int_{s_Y}^{s_X} H_{xx}\big(u, X(u)\big)\,du\right] + \int_{s_X}^{s_Y} H_x\big(u, X(u)\big)\,dW(u). \tag{20}$$

Equation (20) can be regarded as an Ito process, and its drift is

$$Y(s_Y) - \left[ H(s_X, X(s_X)) + \frac{1}{2} \int_{s_Y}^{s_X} H_{xx}(u, X(u)) \, du \right]. \tag{21}$$

The integral term in Eq. (21) is the difference in the first-order differential observation operator between the state scale $s_X$ and the observation scale $s_Y$. This term illustrates that the mapping process should consider not only the observation operator but also the first-order differential term when state is mapped to the observation space. The former is typically determined from the literature, whereas the latter was derived in this study for the first time. This result prompted us to further consider the first-order differential of the observation operator when calculating the representativeness error.

The quadratic variation of Eq. (20) is

$$\int_{s_X}^{s_Y} H_x^2(u, X(u)) du. \tag{22}$$

This equation suggests that the uncertainty in the observation error includes the change in the observation operator from scale $s_X$ to $s_Y$. Therefore, Eq. (21) and Eq. (22) can be combined to produce

$$p(Y|X) = N\left( Y(s_Y) - \left[ H(s_X, X(s_X)) + \frac{1}{2} \int_{s_Y}^{s_X} H_{xx}(u, X(u)) \, du \right], \int_{s_X}^{s_Y} H_x^2(u, X(u)) du \right). \tag{23}$$

Based on Eqs. (17), (18) and (23), $p(Y|X)$, $p(X)$ and $p(Y)$ are stochastic functions that depend on the scale; thus, the posterior PDF of the state is scale-dependent as well.

In particular, if $Y$ is a direct observation, which means that the observation is of the same physical quantity and scale as the state, viz. $H(s, X(s)) = X(s)$. Then the result becomes

$$Y(s_Y) - X(s_X) = Y(s_Y) - X(s_X) + \int_{s_X}^{s_Y} dW(u) \tag{24}$$

$$\text{and } p(Y|X) = N\{Y(s_Y) - X(s_X), |s_Y - s_X|\}. \tag{25}$$

In Eq. (24), the integral $\int_{s_X}^{s_Y} dW(u)$ can be regarded as the noise based on the increment of Brownian motion with respect to scale, and its expectation equals zero.

The significance of Eqs. (20)~(25) is that the effect of scale on the posterior PDF can be determined quantitatively. In addition to the model error and instrument error (both were not introduced explicitly in this study because they have little influence on the error caused by scale transformation), a new type of error in data assimilation was discovered in the analysis step. The expectation of the posterior PDF may vary with the scale of the state space if $Y$ is an indirect observation, and the variance of the drift depends on the difference between $s_Y$ and $s_X$ (based on Eq. (22)). In addition, if $Y$ is a direct observation (Eq. (24) and Eq. (25)), the expectation of the posterior PDF is the difference between $Y$ and $X$, and the variance is equal to the increment of Brownian motion with respect to the scale. Additionally, if the results are not derived from assumption 1, i.e., the scale varies randomly, the posterior PDF is more complex because the Jacobian matrix in the Lebesgue integration of scale transformation is arbitrary.

### 3.4 Example: the stochastic radiative transfer equation (SRTE)

To explicitly show how the stochastic scale transformations impact assimilation, we introduce an illustrative example based on the scales presented in Figure 1. Assume that in the analysis step, the state has the standard scale $s_0$, whose observation footprint is the unit square $A_0$. If the scale of observation space is $s_{C1}$ and its observation footprint is the disc $C_1$, then the Jacobian matrix of the transformation between the scales of the state space and observation space is not diagonal according to the statements in Sect. 3.1, leading the two scales to not obey the one-dimensional rule, which is against assumption 1. However, if the scales of state space and observation space are $s_{C1}$ and $s_{C2}$, respectively, assumption 1 is met, and it can be determined that $s_X = s_{C1} = \frac{\pi}{4} s_0$ and $s_Y = s_{C2} = \frac{9\pi}{4} s_0$.

Now the scales of state space and observation space obey the one-dimensional rule, and we further presume that the measure space $\Omega$ in Figure 1 is free of spatial heterogeneities and dynamic process variations depending on scale. Consequently, the drift rate $\varphi(s) = 0$. If the value of state in the standard scale is denoted as $X_0$ and assuming that $\sigma(s) = 1$, then the prior PDF of state is $X \sim N\left(X_0, \frac{\pi}{4} s_0 - s_0\right)$ according to Eq. (17). It should be noted that $\frac{\pi}{4} s_0 - s_0$ is not a real number and is only used to indicate the variation when the scale changes.

If $H(s, X(s)) = X(s)$, the observation has the same physical quantity as the state, and according to Eq. (25), the likelihood function is $p(Y|X) = N\{Y(s_Y) - X(s_X), |s_Y - s_X|\} = N\{Y(s_Y) - X(s_X), |s_{C2} - s_{C1}|\} = N\left\{Y(s_Y) - X(s_X), \left|\frac{9\pi}{4} s_0 - \frac{\pi}{4} s_0\right|\right\}$.

To formulate the likelihood function in the case that the observation is different from the state, the SRTE will be employed in the following text. The SRTE is a stochastic integral-differential equation that describes the radiative transfer phenomena through a stochastically mixed immiscible media. Scientists have developed analytical or numerical methods for finding the stochastic moments of the solution, such as the ensemble averaged and the variance of the radiation intensity (Pomraning, 1998; Shabanov et al., 2000; Kassianov et al., 2011).

Consider the general expression of the SRTE (leaving out the scattering and emission),

$$-\mu \frac{dI(\tau)}{d\tau} = -I(\tau), \tag{26}$$

where $I(\tau)$, $\mu$ and $\tau$ are the radiation intensity, coefficient of radiation direction and optical depth, respectively.

To tie into more substantial random optical properties of the transfer media, such as absorption and scattering, the optical depth $\tau$ is assumed to be stochastic. This suggests that the optical depth is a scale-dependent Ito process and can be expressed as

$$d\tau(s) = \varphi_\tau(s)ds + \sigma_\tau(s)dW(s), \qquad (27).$$

This causes the radiation intensity to depend on scale.

The analytical solution of Eq. (26) is $I(\tau) = I_0 e^{\tau/\mu}$, where $I_0 = I(\tau(s_0))$.

SRTE can be considered as a concrete instance of a stochastic observation operator by defining $H(s, x(s)) = I(x) = I_0 e^{x/\mu}$. Therefore, $H_s(s, x(s)) = 0$, $H_x(s, x(s)) = \frac{1}{\mu} I_0 e^{x/\mu}$ and $H_{xx}(s, x(s)) = \frac{1}{\mu^2} I_0 e^{x/\mu}$. Based on Ito's Lemma,

$$
\begin{aligned}
dI\big(\tau(s)\big) = dH\big(s,\tau(s)\big) &= H_s\big(s,\tau(s)\big)ds + H_x\big(s,\tau(s)\big)d\tau(s) + \frac{1}{2}H_{xx}\big(s,\tau(s)\big)d\tau(s)d\tau(s) \\
&= \frac{1}{\mu}I_0 e^{\tau(s)/\mu}d\tau(s) + \frac{1}{2\mu^2}I_0 e^{\tau(s)/\mu}d\tau(s)d\tau(s) \\
&= \frac{1}{\mu}I\big(\tau(s)\big)d\tau(s) + \frac{1}{2\mu^2}I\big(\tau(s)\big)d\tau(s)d\tau(s) \\
&= \left(\frac{1}{\mu}I\big(\tau(s)\big)\right)\sigma_\tau(s)dW(s) + \left(\frac{1}{\mu}I\big(\tau(s)\big)\right)\varphi_\tau(s)ds + \left(\frac{1}{2\mu^2}I\big(\tau(s)\big)\right)\sigma_\tau^2(s)ds \\
&= \left(\frac{\sigma_\tau^2(s)}{2\mu^2} + \frac{\varphi_\tau(s)}{\mu}\right)I\big(\tau(s)\big)ds + \left(\frac{\sigma_\tau(s)}{\mu}\right)I\big(\tau(s)\big)dW(s)
\end{aligned} \qquad . (28)
$$

The radiation intensity is a scale-dependent Ito process. The difference between Eq. (28) and the general Ito process is that there is a primitive function $I(\tau(s))$ in the integral term. Therefore, the uncertainty of the radiation intensity is more complex because it is related to both the change of scale and the primitive function.

Integrating both sides of Eq. (28) yields the general solution of the radiation intensity,

$$I\big(\tau(s)\big) = C \cdot \exp\left[\int\left(\frac{\sigma_\tau^2(s)}{2\mu^2} + \frac{\varphi_\tau(s)}{\mu}\right)ds + \int\left(\frac{\sigma_\tau(s)}{\mu}\right)dW(s)\right], \qquad (29)$$

where the constant $C \in R$. Eq. (29) further indicates that $I(\tau(s))$ is a scale-dependent Ito process.

Considering that the optical depth $\tau$ is the state, the radiation intensity $I$ is the observation and $I(\tau(s))$ is the observation operator, the results in Sect. 3.3 could easily be applied here. For example, Eq. (20) and Eq. (23) become

$$Y(s_Y) - H\big(s_X, X(s_X)\big) = I\big(\tau(s_Y)\big) - I\big(\tau(s_X)\big) + \frac{1}{2}\int_{s_X}^{s_Y}\frac{1}{\mu^2}I(\tau)du + \int_{s_X}^{s_Y}\frac{1}{\mu}I(\tau)dW(u), \qquad (30)$$

$$p(Y|X) = N\left(I\big(\tau(s_Y)\big) - I\big(\tau(s_X)\big) + \frac{1}{2}\int_{s_X}^{s_Y}\frac{1}{\mu^2}I(\tau)du, \int_{s_X}^{s_Y}\frac{1}{\mu^2}I^2(\tau)du\right). \qquad (31)$$

Then, the posterior PDF of the data assimilation can be determined by Eq. (27), (29) and (31).

# 4 Discussion & conclusions

## 4.1 Discussion

Our study offered a stochastic data assimilation framework to formulate the errors that are caused by scale transformations. The necessity of the methodology, the difference from previous works by other investigators, and the advantages and limitations of this study are discussed as follows.

The reasons that the methodology focuses on a stochastic framework are as follows. First, the stochastic data assimilation framework is essentially consistent with the concepts of scale and scale transformation; both are associated with corresponding measure spaces $(\Omega, \mathcal{F}, \mu)$. Therefore, it is natural to regard the state space and observation space as two different measure spaces, and each element of state (or observation) vector can be seen as a geophysical variable that maps the state (or observation) measure space onto $R$. Correspondingly, as the integrals of random processes with respect to random processes, stochastic calculus was ultimately adopted. Second, using stochastic calculus can also formulate the errors caused by scale transformations. The study proceeds with  and improves the understanding of representativeness error in terms of scale. The results did not only prove the conventional point that the uncertainties of these errors mainly depend on the differences between scales but also indicated that the first-order differential of the nonlinear observation operator should be incorporated in representativeness error. Third, the error caused by scale transformation was presented in a general form. The drift and quadratic variation of error were formulated by Eq. (21) and Eq. (22), respectively, and both defined the probability distribution space of $p(Y|X)$. Last, stochastic calculus can be extended to meet a general scale transformation and formulate the corresponding representativeness error, which was unattainable in previous work. For example, if the scale changes randomly, say, from an irregular footprint to another irregular footprint, the stochastic equation can offer a multiple integral to present this type of scale transformation, such as $V(x,y) = V_0 + \int_{Y_0}^{Y} \int_{X_0}^{X} \varphi(x,y) dx dy + \int_{Y_0}^{Y} \int_{X_0}^{X} \sigma(x,y) dW_1(x) dW_2(y)$, where $W_1(x)$ and $W_2(y)$ are two independent Brownian motions.

The significant innovation of this work is as follows. We developed a more rigorous formulation of the scale and scale transformation based on Lebesgue measure, which places the related concepts in a rigorous mathematical framework and then provides a new understanding of the errors caused by scale transformation. In addition, due to the Ito process-formed state and observation, a stochastic data assimilation framework was proposed by considering the nonlinear operators, heterogeneity of a geophysical variable and a general Gaussian representativeness error. The scale transformation is also nonlinear if the one-dimensional rule is not applied. Additionally, Ito process-formed state and observation offer the drift rate (i.e., $\varphi(s)$ in Eq. (10)) to formulate the heterogeneity associated with scale transformation. It also permits the representativeness error to be general Gaussian in this framework. If all the integrands in Eq. (13) and Eq. (14) are nonlinear functions instead of constants, then these two equations can be integrated over the field of Brownian motion, and state and observation are the general Gaussian processes of scale. Based on these functions, the representativeness error is a general Gaussian process.

As a theoretical exploration towards scale transformation and stochastic data assimilation, there is still much room for improvement. First, we reduced the scale transformation by the one-dimensional rule, and let the variables in data assimilation evolve regularly according to assumptions 1~3; thus, only the ideal result was investigated. Therefore, an in-depth and comprehensive exploration should be conducted in the future to describe other situations in the real world. However, the use of either an arbitrary scale transformation or the geophysical variable without ignoring the drift rates will obtain lengthy results. Therefore, the second improvement focuses on how to make the formulation more concise. Last, noting that all the results in our framework were given in terms of probability, it is necessary to implement real-world applications of these theoretical results, such as introducing some concrete dynamic models to formulate the Ito process-formed geophysical variable of scale.

## 4.2 Conclusions

In this study, we mainly addressed two basic problems associated with scale transformation in Earth observation and simulation. First, we produced a mathematical formalism of scale and scale transformation by employing measure theory. Second, we demonstrated how scale transformation and its associated errors could be presented in a stochastic data assimilation framework.

We revealed that the scale is the Lebesgue measure with respect to the observation footprint or model unit. The scale is related to the shape and size of a footprint, and scale transformation depends on the spatial change between different footprints. We then defined the geophysical variable, which further considers the heterogeneities and physical processes. A geophysical variable consequently expresses the spatial average at a specific scale.

We formulated the expression of scale transformation and investigated the error structure that is caused by scale transformation in data assimilation using basic theorems of stochastic calculus. The formulations explicate that the first-order differential of the nonlinear observation operator should be considered in representativeness error, and the uncertainty of representativeness error is directly associated with the difference between scales. A concrete physical models (SRTE) was introduced to demonstrate the results when observation operator is nonlinear.

This work conducted a theoretical exploration of formulating the errors caused by scale transformation in a stochastic data assimilation framework. We hope that the stochastic methodology can benefit the study of these errors.

## 5 Notation

### 5.1 Basic notations

| | |
|---|---|
| $\Omega$ | Non-empty space |
| $\mathcal{F}$ | σ-algebra |
| $\mu$ | Measure |
| $dV$ | Variable process |

| | | | |
|---|---|---|---|
| $W(s)$ | Brownian motion | | |
| $(\Omega, \mathcal{F}, \mu)$ | Measure space | | |
| $I^n$ | N-dimensional Lebesgue volume | | |
| $m^n$ | Lebesgue measure or an outer measure on $R^n$ | | |
| $\mathcal{L}^n$ | Lebesgue σ-algebra of $R^n$ | | |
| $\int f dm^n$ | Lebesgue integral | | |
| $|J(\cdot)|$ | Jacobian determinant | | |
| $(\Omega^n, \mathcal{F}^n)$ | Product space | | |

## 5.2 New notations

| Notation | Name | Explanation | Index |
|---|---|---|---|
| $s$ | Scale | The observation footprint or model unit to observe or model a geophysical variable | Sect. 1 & Sect. 3.1 |
| $A_0$ | Unit square in $R^2$ | | Sect. 3.1 |
| $s_0$ | Standard scale | A Lebesgue integral where $A_0$ is the unit area | Sect. 3.1 |
| | One-dimensional rule | Two scales are geometrically similar | Eq. (8) |
| $V$ | Geophysical variable | Estimation of a variable at a specific scale | Sect. 3.2 |
| $dX$ | State process | Ito process-formed state | Eq. (13) |
| $dY$ | Observation process | Ito process-formed observation | Eq. (14) |
| $X_0$ | State in $s_0$ | | Eq. (15) |
| $Y_0$ | Observation in $s_0$ | | Eq. (16) |
| $s_X$ | Scale of state space | | Eq. (15) |
| $s_Y$ | Scale of observation space | | Eq. (16) |

10 **Acknowledgements**

We thank the editor-in-chief of NGP, Prof. Talagrand, and his kind help and valuable comments on our manuscript. We also thank Dr. van Leeuwen and another anonymous reviewer for their valuable comments and suggestions. This work was supported by the NSFC projects (grant numbers 91425303 & 91625103) and the CAS Interdisciplinary Innovation Team of the Chinese Academy of Sciences.

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

# Formulation of Scale Transformation in a Stochastic Data Assimilation Framework

Feng Liu[1, 3] and Xin Li[1, 2, 3]

Key Laboratory of Remote Sensing of Gansu Province, Cold and Arid Regions Environmental and Engineering Research Institute, Chinese Academy of Sciences, Lanzhou, 730000, China
[2]Center for Excellence in Tibetan Plateau Earth Sciences, Chinese Academy of Sciences
[3]University of Chinese Academy of Sciences, Beijing 100049, China

*Correspondence to*: Xin Li (lixin@lzb.ac.cn)

**Abstract:** Understanding ~~of~~ the errors caused by spatial scale transformation in Earth observations and simulations requires a rigorous definition of scale. These errors are also an important component of representativeness errors in data assimilation. Several relevant studies have been conducted, but the ~~theorization~~ theory of the scale associated with representativeness errors is still not well developed. We addressed these problems by reformulating the data assimilation framework using measure theory and stochastic calculus. First, ~~the~~ measure theory is used to propose that the spatial scale is ~~the~~ a Lebesgue measure

with respect to the observation footprint or model unit, and the Lebesgue integration by substitution is used to describe the scale transformation. Second, a scale-dependent geophysical variable is defined to consider the heterogeneities and dynamic processes. Finally, the structures of the scale-dependent errors are studied in the Bayesian framework of data assimilation based on stochastic calculus. ~~–~~All the results were presented on the condition that the scale is one-dimensional, and the variations in these errors depend on the differences between scales. This new formulation provides a more general framework

to understand the representativeness error in a nonlinear and stochastic sense and is a promising way to address the spatial scale issue.

## 1 Introduction

The spatial scale in Earth observations and simulations refers to the observation footprint or model unit in which a geophysical variable is observed or modelled (scale is used below ~~to~~ as an abbreviation for ~~abbreviate~~ spatial scale). Scale is

25 traditionally defined in terms of distance, which is not adequate both because distance is a one-dimensional quantity ~~but~~ while scale generally refers to a two- or three-dimensional space~~,~~ and because the scale may change ~~much~~ in a very complicated manner (for example, ~~–~~from an irregular observation footprint to a square observation footprint). Generally, the scale is not explicitly expressed in the dynamics of a geophysical variable, partially because a rigorous definition of scale is difficult to

find, except for an intuitive conception (Goodchild and Proctor, 1997) and certain qualitative classifications of scale (Vereecken et al., 2007). This reflects the complexity of scale and requires a more rigorous mathematical conceptualization of scale.

~~Scale~~ The scale transformation of a geophysical variable may result in significant errors (Famiglietti et al., 2008; Crow et al., 2012; Gruber et al., 2013; Hakuba et al., 2013; Huang et al., 2016; Li and Liu, 2016; Ran et al., 2016). These errors are mainly caused by the strong spatial heterogeneities (Miralles et al., 2010; Li, 2014) and irregularities (Atkinson and Tate, 2000) that are associated with geophysical variables across different scales, and are also closely related to dynamic variations, e.-g., in hydrological (Giménez et al., 1999; Vereecken et al., 2007; Merz et al., 2009; Narsilio, et al. 2009), soil (Ryu and Famiglietti, 2006; Lin et al., 2010) and ecological (Wiens, 1989) processes. How to develop mathematical tools to elucidate the scale transformation has yet to be fully addressed.

Data assimilation could be an ideal tool to explore the scale transformation because it presents a unified and generalized framework in Earth system modelling and observation (Talagrand, 1997). Geophysical data are typically observed by various Earth observations~~,~~; thus, updating~~therefore to update~~ the observation data in a data assimilation system may result in scale transformations between the observation space and system state space. If observation operator is strongly nonlinear and complex, the errors caused by the scale transformation ~~is~~ are even more serious (Li, 2014). An important concept that is related to the scale transformation in data assimilation is "representativeness error", which is associated with the inconsistency in the spatial and temporal resolutions between states, observations and operators (Lorenc, 1986; Janjić and Cohn, 2006; van Leeuwen, 2014; Hodyss and Nichols, 2015)~~,~~ and the missing physical information that is related to a numerical operator compared to the ideal operator (van Leeuwen, 2014), such as the discretization of a continuum model or neglect of necessary physical processes. The representativeness error and instrument error make up the observation error of data assimilation. Under the Gaussian assumption, they are independent of each other (Lorenc, 1995; van Leeuwen, 2014). This study will not ~~introduce~~ consider the instrument error when ~~formulate~~ formulating the scale transformation in data assimilation.

Recently, approaches have been developed to assess the representativeness error. Janjić and Cohn (2006) studied the representativeness error by treating system state as the sum of resolved and unresolved portions. Bocquet et al. (2011) used a pair of operators, namely, restriction and prolongation, to connect the relationship between the finest regular scale and a coarse

scale, and determined the representativeness error using a multi-scale data assimilation framework. van Leeuwen (2014) considered two complicated cases, i.e., conducting the observation vector in a finer resolution compared with system state vector and assimilating the retrieved variables. Their solutions were formulated using an agent in observation or state space, and a particle filter was proposed to treat the nonlinear relationship between observations, states and retrieved values. Hodyss and Nichols (2015) also estimated the representativeness error by investigating the difference between the truth and the inaccurate value that is generated by forecasting model.

Although these approaches explored the structure of the representativeness error and offered various solutions, improvements are still necessary to investigate ~~what is~~ the exact expression of the errors caused by scale transformation in data assimilation. The authors believe that these approaches are optimal in linear systems~~,~~ but may not be suitable when observations are heterogeneous and sparse, or when operators are nonlinear between states and observations, although the general equations in the nonlinear case were given.Without taking heterogeneities and nonlinear operators into account, the representativeness error cannot be fully understood. However, heterogeneity varies depending on the situation and is difficult to ~~be~~ formulated in a general ~~theory~~ theoretical study.

Data assimilation studies based on stochastic processes (Apte et al., 2007; Miller, 2007~~; Apte et al., 2007~~) or a stochastic dynamic model (Miller et al., 1999; Eyink et al., 2004) have been proposed recently. Compared to deterministic models, stochastic data assimilation is more applicable in an integrated and time-continuous theoretical study (Bocquet et al., 2010)~~,~~ and creates an infinite sampling space of the system state (Apte et al., 2007). Although the theorems of calculus that are based on stochastic processes (or stochastic calculus) are different from those of ordinary calculus, these advantages suggest that stochastic data assimilation offers a more general framework to study scale transformation.

We attempt to explore the mathematic definitions of scale and scale transformation~~,~~ and then formulate the errors caused by the scale transformation in a general theoretical~~y~~ study on stochastic data assimilation. The next section introduces the basic concepts and theorems of measure theory, stochastic calculus and data assimilation. In Sect. 3, we present the definitions of scale and scale transformation. The posterior probability of system state ~~was~~ is also reformulated by scale transformation in a stochastic data assimilation framework. In the final section, the contributions and deficiencies of this study ~~were~~ are discussed.

## 2 Basic knowledge

The scale greatly depends on the geometric features of a certain observation footprint or model unit. The model unit is a specified subspace where a geophysical variable evolves in the model space~~. I;~~ it could be a point, a rectangular grid, or an

irregular unit such as a response unit (watershed, landscape patch ~~and so on,~~ etc.). We offer a solution in which the definition of scale uses measure theory and the expression of a geophysical variable as a stochastic process uses stochastic calculus. Therefore, we first introduce several basic concepts of measure theory and stochastic calculus.

## 2.1 Measure theory

Let $\Omega$ be an arbitrary non-empty space. $\mathcal{F}$ is a **σ-algebra** (or **σ-field**) of subsets of $\Omega$ that satisfies the following conditions:

(i)  $\Omega \in \mathcal{F}$, and the empty set $\Phi \in \mathcal{F}$;

(ii)  $A \in \mathcal{F}$ implies that its complementary set $A^c \in \mathcal{F}$;

(iii)  $A_1, A_2, \cdots \in \mathcal{F}$ implies their union $A_1 \cup A_2 \cup \cdots \in \mathcal{F}$.

A set function $\mu$ of $\mathcal{F}$ is called a **measure** if it satisfies the following conditions:

(1)  $\mu(A) \in [0, \infty)$ and $\mu(\Phi) = 0$;

(2)  If $A_1, A_2, \cdots \in \mathcal{F}$ is any disjoint sequence and $\bigcup_{k=1}^{\infty} A_k \in \mathcal{F}$, $\mu$ is countably additive such that $\mu(\bigcup_{k=1}^{\infty} A_k) = \sum_{k=1}^{\infty} \mu(A_k)$.

If $\mu(\Omega) = 1$, $\mu$ can be replaced by the probability measure $p$, and if $\mu$ is finite, $p$ can be calculated as $p(A) = \mu(A)/\mu(\Omega)$. The triples $(\Omega, \mathcal{F}, \mu)$ and $(\Omega, \mathcal{F}, p)$ are the **measure space** and **probability measure space**, respectively.

Let $\Omega$ be the set of real numbers $R$ and σ-algebra $\mathcal{B}$ be **Borel algebra**, which is generated by all closed intervals in $R$. Then, $\forall A = [a, b] \in B$, a **Lebesgue measure** on $R$ is defined as $I(A) = b - a$. Intuitively, the Lebesgue measure on $R$ coincides with the length.

An **n-dimensional Lebesgue volume** is defined to measure the standard volumes of the subsets in $R^n$ based on $I^n(A) = \prod_{k=1}^{n}(b_k - a_k)$, where $A = [x: a_k \leq x_k \leq b_k, k = 1, 2, \cdots, n]$ is an n-dimensional regular cell in $R^n$. The n-dimensional Lebesgue volume is an ordinary volume, such as length (n=1), area (n=2) and volume (n=3).

Next, the **outer measure** is defined as $m^n(A) = \inf\{\sum_{i=1}^{+\infty} I^n(A_i)\}$, where $\inf\{\cdot\}$ is the infimum, $A_i = [x: a_{i,k} \leq x_k \leq b_{i,k}, k = 1, 2, \cdots, n]$ is the n-dimensional regular cell in $R^n$, and $A \subseteq \bigcup_{i=1}^{+\infty} A_i$. Thus, if $A$ is any subset of $R^n$, one can collect many sets of n-dimensional regular cells $\{A_i\}$ to cover $A$. Among them, the outer measure denotes the set whose union has the smallest n-dimensional Lebesgue volume.

To match the two conditions of a measure, one can define the outer measure $m^n$ as a **Lebesgue measure** on measure spaces $(R^n, \mathcal{L}^n, m^n)$, where- $\mathcal{L}^n$ is the **Lebesgue σ-algebra** of $R^n$. The construction of the Lebesgue σ-algebra is based on the Caratheodory condition (Bartle, 1995, definition 13.3). Fortunately, almost all of the observation footprints and model units are finite and ~~closed, leading~~closed, leading them to be Lebesgue measurable. This consequently ensures that the Lebesgue measure $m^n$ is a measure and the triple $(R^n, \mathcal{L}^n, m^n)$ is a measure space~~s~~. The Lebesgue measure of a Lebesgue measurable subset in $R^n$ also coincides with its volume.

The n-dimensional **Lebesgue integral** in $(R^n, \mathcal{L}^n, m^n)$ is $\int f dm^n$ , where $f$ is a real function on $R^n$. The Lebesgue integral can be further denoted by $\int f dm^n = \int f(x)dx$, where $x \in R^n$ and $x = (x_1, \cdots, x_n)$.

In the two-dimensional case ($n = 2$), the Lebesgue integral is

$$\iint_A f(x_1, x_2)dx_1 dx_2,$$

where $A \in \mathcal{L}^2$. Next, we consider the **Lebesgue integration by substitution** on $R^2$. Let $T(x_1, x_2) = [t_1(x_1, x_2), t_2(x_1, x_2)] = [y_1, y_2]$ be a one-to-one mapping of a subset $X$ onto another subset $Y$ on $R^2$. Assuming that $T$ is continuous and has a continuous partial derivative matrix $T_x = \begin{pmatrix} \partial t_1/\partial x_1 & \partial t_1/\partial x_2 \\ \partial t_2/\partial x_1 & \partial t_2/\partial x_2 \end{pmatrix}$, then

$$\iint_Y f(y_1, y_2)dy_1 dy_2 = \iint_X f(T(x_1, x_2))|J(x_1, x_2)|dx_1 dx_2 ,$$

where the Jacobian determinant- $|J(x_1, x_2)| = |\det T_x| = \begin{vmatrix} \partial t_1/\partial x_1 & \partial t_1/\partial x_2 \\ \partial t_2/\partial x_1 & \partial t_2/\partial x_2 \end{vmatrix}$. If $T$ is linear, the integral reduces to

$$\iint_Y f(y_1, y_2)dy_1 dy_2 = |J(x_1, x_2)| \iint_X f(T(x_1, x_2))dx_1 dx_2.$$

By doing so, any observation footprint or model unit can be regarded as a Lebesgue measurable subset in a two-dimensional space $R^2$.

Additional details regarding measure theory can be found in the literature (for example, Billingsley, 1986; Bartle, 1995).

## 2.2 Stochastic calculus

We then introduce some necessary concepts and theorems of stochastic calculus. All the classic theorems are introduced without proofs; their detailed derivations can be found in the literature (Itô, 1944; Karatzas et al., 1991; Shreve, 2005).

**Stochastic calculus** is defined for ordinary integrals with respect to stochastic processes. One of the simplest stochastic processes is **Brownian motion**. The Brownian motion $W$ that is defined on a probability measure space $(\Omega, F, p)$ is characterized as follows:

1) $W(0) = 0$.

2) $\forall t_1 > s_1 \geq t_2 > s_2 > 0$, the increments $W(t_1) - W(s_1)$ -and $W(t_2) - W(s_2)$ are independent.

3) $\forall t > s \geq 0, W(t) - W(s) \sim N(0, t - s)$.

The last two conditions represent that $\forall t_2 > s_2 > t_1 > s_1 \geq 0$, $W(t_2) - W(s_2)$ and $W(t_1) - W(s_1)$ are independent Gaussian random variables. Additionally, $W$ is related to the probability measure $p$.

Stochastic calculus based on Brownian motion produces an **Ito process**. The differential form of the time-dependent Ito process is

$$dI = \varphi(t)dt + \sigma(t)dW(t), \qquad\qquad (1)$$

where $\varphi(t)$, $\sigma(t)$ and $W(t)$ are the drift rate, volatility rate and Brownian motion, respectively. The integral form of Eq. (1) is

$$I(t) = I(0) + \int_0^t \varphi(u)du + \int_0^t \sigma(u)dW(u). \qquad (2)$$

**Theorem 1**: For any Ito process defined as in Eq. (1), the **quadratic variation** that is accumulated on the interval $[0, t]$ is

$$[I, I](t) = \int_0^t \sigma^2(u)du, \qquad\qquad (3)$$

and the **drift** of Eq. (1) is $I(0) + \int_0^t \varphi(u)du$.

As distinguishing features of stochastic calculus, the quadratic variation and drift can be regarded as stochastic versions of the variance and expectation, respectively. That is, the variance and expectation are instances of their random variable stochastic counterparts within a certain integral path. Therefore, rather than being constants, the quadratic variation and drift are given in terms of probability.

**Theorem 2 (Ito's Lemma)**: If the partial derivatives of function $f(u, x)$, viz. $f_u(u, x)$, $f_x(u, x)$ and $f_{xx}(u, x)$, are defined and continuous, and if. If $t \geq 0$, we have

$$f(t, x(t)) = f(0, x(0)) + \int_0^t f_u(u, x(u))du + \int_0^t f_x(u, x(u))\sigma(u)dW(u) + \int_0^t f_x(u, x(u))\varphi(u)du +$$

$$\frac{1}{2}\int_0^t f_{xx}(u, x(u))\sigma^2(u)du. \qquad\qquad (4)$$

Ito's Lemma is typically used to build the differential of a stochastic model with Ito processes. In this study, Ito's Lemma is applied to study the scale-dependent relationship between the observation and state, and the errors caused by scale transformation.

## 2.3 Traditional formulation of data assimilation in the Bayesian theorem framework

We use the ~~well~~ well-accepted Bayesian theorem of data assimilation (Lorenc, 1995; van Leeuwen, 2015) to investigate its time- and scale-dependent errors. State and observation are first assumed to be one-dimensional. ~~In Sect. 3.5, the results are extended to n-dimensional state vectors and observation vectors.~~

A nonlinear forecasting system can be described by

$$X(t_k) = M_{k-1:k}\big(X(t_{k-1})\big) + \eta(t_k), \qquad (5)$$

where $M_{k-1:k}(\cdot)$, $X(t_k)$ and $\eta(t_k)$ represent a nonlinear forecasting operator that transits the state from the discrete time $k-1$ to $k$, the state with prior probability distribution function (PDF) $p(X)$, and the model error at time $k$, respectively.

If a new observation is available at time $k$, the observation system is given by

$$Y^o(t_k) = H_k\big(X(t_k)\big) + \varepsilon(t_k), \qquad (6)$$

where $H_k(\cdot)$, $Y^o(t_k)$ and $\varepsilon(t_k)$ represent the nonlinear observation operator, true observation with prior PDF $p(Y)$, and observation error at time $k$, respectively.

Previous studies (e.g., Janjić and Cohn, 2006; Bocquet et al. 2011) described the origins of the components of $\varepsilon(t_k)$ and $\eta(t_k)$, such as white noise, the discretization error of a continuum model, the errors that are caused by missing physical processes, and the scale-dependent bias. In this study, we assume that both forecasting and observation operators are perfect models; thus, ~~so~~ errors ~~that are~~ caused by missing physical processes are discarded.

According to Bayesian theory, the posterior PDF of the state based on the addition of a new observation into the system is

$$p(X|Y) = p(Y|X)p(X)/p(Y), \qquad (7)$$

where $p(X|Y)$ is the posterior PDF that presents the PDF value of state $X$ given an available observation $Y$. $p(Y|X)$ is a likelihood function, which is the probability that an observation is $Y$ given a state $X$. $p(X)$ and $p(Y)$ are the prior PDF values of the state and observation, respectively. Here, $p(X)$ is supposed to be known and $p(Y)$ is a ~~normalisation~~ normalization constant (van Leeuwen, 2014). The aim of data assimilation is equivalent to finding the posterior PDF $p(X|Y)$.

## 3 Reformulation of scale transformation in data assimilation framework

### 3.1 Definition of scale

We define the scale based on the measure theory that was introduced in Sect. 2. The relationship between Lebesgue measure in $(R^2, \mathcal{L}^2, m^2)$ and scale is first~~ly~~ introduced by the following measures of Earth observations.

(i) Measure of a ~~single~~ single-point observation: When the observation footprint is very small and homogeneous, we assume that its footprint approaches zero, and its measure is accordingly zero under the condition of the Lebesgue measure.

(ii) Measure ~~in~~ along a line: The measure is a one-dimensional Lebesgue measure.

(iii) Measure of a rectangular pixel (for example, remote sensing observation): $\forall A = [x: a_k \leq x_k \leq b_k, k = 1,2]$, it is a two-dimensional Lebesgue volume, i.e., $\mu_{iii}(A) = I^2(A) = \prod_{k=1}^{2}(b_k - a_k)$.

(iv) Measure of a footprint-scale observation: The footprint is any bounded closed domain $A$, which is not necessary to be regular rectangles, but ~~as~~ can also be circles or ellipses. We use Lebesgue measure on $R^2$, i.e., $\mu_{iv}(A) = m^2(A) = \inf\left\{\sum_{i=1}^{+\infty} I^2(A_i)\right\}$, where $A_i = [x: a_{i,k} \leq x_k \leq b_{i,k}, k = 1,2]$ and $A \subseteq \bigcup_{i=1}^{+\infty} A_i$. ~~Obviously~~Clearly, measures (i)~(iii) are ~~the~~ special cases of the measure of a footprint-scale observation.

~~Actually, all~~All of the above measures depend mainly ~~depend~~ on the shape and size of $A$. The Lebesgue measure on $R^2$ coincides with the area~~, so~~; thus, the Lebesgue integral of $\mu_{iv}(A)$ is $\iint_A dx_1 dx_2$, where the real function $f \equiv 1$.

Now, we can generalize the above examples by defining the **scale** as the Lebesgue measure with respect to the observation footprint. This definition can also be extended to a certain model unit. Thus, for any subset $A \in \mathcal{L}^2$, the scale is $s = m^2(A) = \iint_A dx_1 dx_2$, where the real function $f \equiv 1$. From a geometric perspective, the measure function $m^2(\cdot)$ refers to the shape of the subset, and the scale further indicates its size.

We represent the scale as $s$, and let $s_0 = m_0^2(A_0) = \iint_{A_0} dx_1 dx_2 = 1$ be the **standard scale**, where $A_0 = [x: 0 \leq x_k \leq 1, k = 1,2]$ is the unit square in $R^2$. The standard scale can be regarded as a basic unit of scale. It presents a standard reference~~,~~ by which one can make a quantitative comparison between different scales. The standard scale is also the origin of scales that lets scales vary similarly to other physical quantities, such as time.

We can further define **scale transformation**. For $\forall A_1, A_2 \in \mathcal{L}^2$, if there are two different scales, $s_1 = m^2(A_1) = \iint_{A_1} dx_1 dx_2$ and $s_2 = m^2(A_2) = \iint_{A_2} dy_1 dy_2$, then we can obtain $s_2 = \iint_{A_2} dy_1 dy_2 = \iint_{A_1} |J(x_1, x_2)| dx_1 dx_2$ based on Lebesgue integration by substitution, where the Jacobian matrix $J(x_1, x_2)$ represents the geometric transformation from $A_1$

to $A_2$. In particular, if $J(x_1, x_2) = diag(\xi, \xi), \xi \in R$, which also indicates that the geometric transformation is linear, then the following expression is valid based on Lebesgue integration by substitution:

$$s_2 = |J(x_1, x_2)| \iint_{A_1} dx_1 dx_2 = \xi^2 s_1, \quad (8)$$

where $s_1$ and $s_2$ represent the change of the **one-dimensional rule**.

If two scales follow the one-dimensional rule, they are geometrically similar. This rule simplifies scale as a one-dimensional variable that corresponds to the scale transformations between most remote sensing images with various spatial resolutions. For example, $\forall A = [x: a \leq x_k \leq b, k = 1,2]$, where $A$ and the unit square $A_0$ are geometrically similar, and the scale $s = \mu_{iii}(A)$ can be expressed by the one-dimensional rule of scale transformation: $s = \mu_{iii}(A) = |J(x_1, x_2)| \iint_{A_0} dx_1 dx_2 = (b-a)^2 s_0$. For another example, let $s = \iint_A dy_1 dy_2$ be the scale of a disc footprint $A$ with radius $r$. The mapping function

between $A$ and $A_0$ is $T(x_1, x_2) = [rx_1 \cos(2\pi x_2), rx_1 \sin(2\pi x_2); 0 \leq x_1 \leq 1, 0 \leq x_2 \leq 1] = [y_1, y_2]$, and the Jacobian

determinant $|J(x_1, x_2)| = \begin{vmatrix} r\cos(2\pi x_2) & -2\pi rx_1 \sin(2\pi x_2) \\ r\sin(2\pi x_2) & 2\pi rx_1 \cos(2\pi x_2) \end{vmatrix} = 2\pi r^2 x_1$ . Therefore, $s = \iint_A dy_1 dy_2 = \iint_{A_0} |J(x_1, x_2)| dx_1 dx_2 = \pi r^2 s_0$, which is equal to its area. However, $s_0$ and $s$ do not obey the one-dimensional rule because the Jacobian matrix is not diagonal.

     The Layer 1 in Figure 1 shows the relationship between the Lebesgue measure and scale. The measure space $\Omega =$

$[x: 0 \leq x_k \leq 4, k = 1,2]$ is regularly divided by the unit square $A_0$. Let scales $s_{C1} = m_{C1}^2(C1)$, $s_{C2} = m_{C2}^2(C2)$ and $s_{C3} = m_{C3}^2(C3)$ be the Lebesgue measures of disc observation footprints $C1, C2$ and $C3$, respectively. Then, $m_{C1}^2(\cdot) = m_{C2}^2(\cdot) = m_{C3}^2(\cdot)$ because they are the same Lebesgue measure functions. That is, if $\{A_i\}$ is the set with the smallest volume that covers $C1$, then similar sets $\{A_i + 2\}$ and $\{A_i \times 3 + 2\}$ can be used (with the origin located in the upper-left corner) to cover $C_3$ and $C_2$ with the smallest volumes, respectively. Here, $A_i + 2 = [x_i: x_{i,k} + 2, x_{i,k} \in A_i, k = 1,2]$ and $A_i \times 3 + 2 = [x_i: x_{i,k} \times$

$3 + 2, x_{i,k} \in A_i, k = 1,2]$, which proves that functions $m_{C1}^2(\cdot)$, $m_{C2}^2(\cdot)$ and $m_{C3}^2(\cdot)$ collect the desirable desired set based on the same scheme, so; therefore, they are identical. Additionally, $s_{C2} = m_{C2}^2(C2) = \sum I^2(A_i \times 3 + 2)$ is much larger than $s_{C1} = m_{C1}^2(C1) = \sum I^2(A_i)$ and $s_{C3} = m_{C3}^2(C3) = \sum I^2(A_i + 2)$. Therefore, the scale of $C2$ is not equal to the two other scales

because the volumes of their subsets are different. However, their scales are governed by one-dimensional rules because their measures are identical and the Jacobian matrices between them are diagonal.

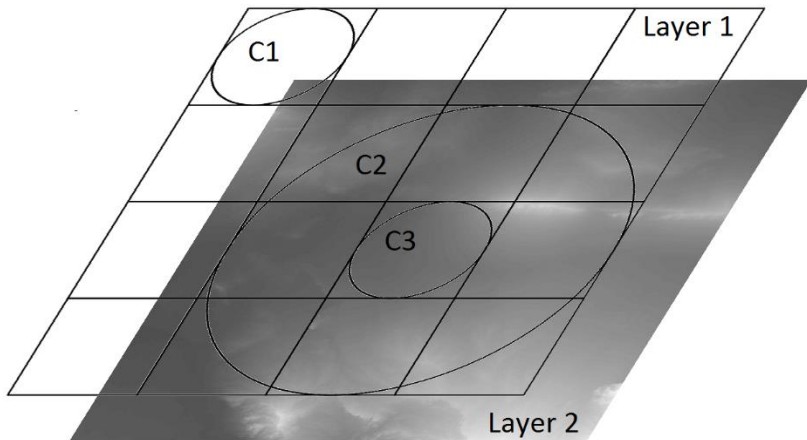

Figure 1. Diagram of the relationships among a Lebesgue measure, scale and geophysical variable

## 3.2 Stochastic variables in data assimilation

Instead of using Eqs. (5) and (6), which are discrete in time, we use Ito process-formed expressions with the one-dimensional infinitesimals $ds$ and $dt$ to formulate a continuous-time (or continuous-scale) state and observation.

~~Geophysical~~ A geophysical variable can be regarded as a real function $V(s,t)$~~-~~, and it maps the space $(R^2, \mathcal{L}^2, m^2)$ onto $R$, where $s$ is the scale, $s = m^2(A), A \in \mathcal{L}^2$, and $t$ is the time. In n-dimensional data assimilation, a geophysical variable $V$ is related to an element of state vector $X$ at a specific scale $s$ and time $t$.

In Figure 1, Layer 2 presents a heterogeneous geophysical variable in the entire region. If ~~aggregating~~ we aggregate Layer 2 into Layer 1 and let each pixel intensity ~~is~~ be the value for a geophysical variable in ~~each~~ that pixel, then the measure space $\Omega$ is heterogeneous. A geophysical variable represents a spatial average in a specific observation footprint with a specific scale. Therefore, the geophysical variables in $C1$ and $C3$ are not equal because their observation footprints are different, and the geophysical variables in $C2$ and $C3$ are also different because the scale changes. The former introduces that the geophysical variables vary with the location, and the latter states that the geophysical variables are scale-dependent.

If the statistical properties of the geophysical variable are available, we can construct an explicit stochastic equation for ~~the geophysical variable~~it. We introduce the time-dependent Ito process Eq. (1) to define the geophysical variable process:

$$dV = p(t)dt + q(t)dW(t). \qquad (9)$$

Similarly, the geophysical variable is supposed to evolve via a stochastic process, for which the dynamic process and uncertainty are allowed to vary with scale~~,~~.

$$dV = \varphi(s)ds + \sigma(s)dW(s), \qquad (10)$$

where $\varphi(s)$ and $\sigma(s)$ are the scale-based drift rate and volatility rate, respectively. The geophysical variable is a probabilistic process with respect to scale and thus has scale-dependent errors, where the scale should shift forward or backward based on the condition that the scale follows the one-dimensional rule.

Eq. (9) can be regarded as a continuous-time version of Eq. (5), i.e., ~~to estimate~~the estimation of the state is equal to the integral of Eq. (9) over a time interval. Here, $p(t)$ indicates the physical process with respect to time, and $q(t)$ is the error only caused by the evolution of time~~, so~~; thus, model error $\eta$ in Eq. (5) contains more parts than $q(t)$. Eq. (10) implies that the value and variance of a geophysical variable may change if the scale changes. ~~To formulate~~The formulation of $\varphi(s)$ should consider ~~both~~ the spatial heterogeneities and physical process variations among different scales, which together constitute the deterministic part of a geophysical variable. However, neither of them is well understood in a general ~~theory~~theoretical study. Therefore, $\varphi(s)$ is conceptualized in Eq. (10). Particularly, if the study region is homogeneous, then the values of a variable that are observed at the same place are identical between the large scale and fine scale, and $\varphi(s)$ can be left out. Due to the integral over the space of Brownian motion, $\sigma(s)$ is the stochastic part, meaning that scale transformation produces uncertainties. ~~error caused by t~~ The formulation of $\varphi(s)$ should consider the spatial heterogeneities and physical process variations among different scales, which together constitute the deterministic part of a geophysical variable. However, neither of them is well understood in a general theoretical study. Therefore, $\varphi(s)$ is conceptualized in Eq. (10). Particularly, if the study region is homogeneous, then the values of a variable that are observed at the same place are identical between the large scale and fine scale, and $\varphi(s)$ can be left out. Due to the integral over the space of Brownian motion, $\sigma(s)$ is the stochastic part, meaning that scale transformation produces uncertainties. ~~he scale transformation.~~

The state in the forecasting step can be expressed by Eq. (9) because only time is involved. In the analysis step of data assimilation, the state does not pertain to time, and we assume that the scale has a quantifiable effect on the errors in this step; thus, both the states and observations can be defined by Eq. (10).

### 3.3 Expression of scale transformation in a stochastic data assimilation framework

First, we provide the following lemma.

**Lemma 1**: For $\forall s_0 > 0$, let $W^*(0) = W(s_0) - W(s_0), \dots, W^*(s) = W(s_0 + s) - W(s_0)$; then, $W^*(s), s \geq 0$ is a Brownian motion.

**Proof.** First, $W^*(0) = W^*(s_0) - W^*(s_0) = 0$. $\forall s_{i+1} > s_i \geq 0, i = 1,2,3, \dots$, $W^*(s_{i+1}) - W^*(s_i) = [W(s_0 + s_{i+1}) - W(s_0)] - [W(s_0 + s_i) - W(s_0)] = W(s_0 + s_{i+1}) - W(s_0 + s_i)$, which suggests that the increments $[W^*(s_{i+1}) - W^*(s_i)]$ are equal to $[W(s_0 + s_{i+1}) - W(s_0 + s_i)]$ and are independent Gaussian distributed. Therefore, $W^*(s), s \geq 0$ is a Brownian motion, with $E[W^*(s_{i+1}) - W^*(s_i)] = 0$ and $Var[W^*(s_{i+1}) - W^*(s_i)] = s_{i+1} - s_i$. Q. E. D.

**Remark on Lemma 1**: Note that in the definition of Brownian motion, the parameter starts at zero. However, the scale is realistically greater than zero, which ~~results~~means that it cannot be directly applied in Brownian motion. ~~So~~Therefore, Lemma

1 is logical because it implies that $W(s), s \geq s_0$ is an equivalent expression of $W^*(s), s \geq 0$. Therefore, beginning with the standard scale, the Brownian motion and stochastic calculus with respect to scale can be further developed.

In the following content, we use Brownian motion with a parameter that starts at $s_0$ to define the scale-dependent geophysical variables; therefore, the classic expressions above are changed. According to Lemma 1, Eq. (3) is given by

$$[I, I](s) = \int_{s_0}^{s} \sigma^2(u) du. \tag{11}$$

Additionally, the integral form of ~~the~~ Eq. (10) is ~~as follows:~~

$$V(s) = V_0 + \int_{s_0}^{s} \varphi(u) du + \int_{s_0}^{s} \sigma(u) dW(u), \tag{12}$$

where $V_0 = V(s_0)$, and the drift of Eq. (12) is

$$V_0 + \int_{s_0}^{s} \varphi(u) du.$$

Similarly, Eq. (4) becomes

$$f(s, V(s)) = f(s_0, V(s_0)) + \int_{s_0}^{s} f_u(u, V(u)) du + \int_{s_0}^{s} f_x(u, V(u)) \sigma(u) dW(u) + \int_{s_0}^{s} f_x(u, V(u)) \varphi(u) du +$$

$$\frac{1}{2} \int_{s_0}^{s} f_{xx}(u, V(u)) \sigma^2(u) du.$$

Now, we make the following assumptions.

**Assumption 1**: The scale transformations between the state and observation spaces of data assimilation obey the one-
dimensional rule as defined in Sect. 3.1.

**Assumption 2**: In the forecasting step, the model unit equals the scale of the state space, and both of them are constant.

**Assumption 3**: In the analysis step, the state, observation and observation operator are scale dependent. Only one observation is added into the data assimilation system at a time.

In assumption 1, the one-dimensional rule ensures that scale changes in a sense of geometrical similarity (for example, from
a larger square observation footprint to a smaller square observation footprint, or from $C_2$ to $C_3$ as presented in Figure 1). Therefore, based on assumption 1, scale only varies in one-dimensional space, meaning that the corresponding scale transformation is an integral over one-dimensional space. ~~Additionally, the formulations of scale transformation can be extremely reduced.~~

Assumption 2 indicates that the model unit and state scale are ~~both~~ supposed to be the same and both invariant in space and
time. ~~So~~Thus, there is no scale transformation in the forecasting step. ~~Thus;~~ Thus, Eq. (9) can adequately describe this step.

Based on assumption 3, the analysis step is related to the scale. According to Eq. (10), the state and observation in the analysis step are ~~as follows:~~

$$dX = \varphi_X(s) ds + \sigma_X(s) dW(s) \tag{13}$$

and

$$dY = \varphi_Y(s) ds + \sigma_Y(s) dW(s), \tag{14}$$

where $\varphi_X(s)$, $\sigma_X(s)$, $\varphi_Y(s)$ and $\sigma_Y(s)$ represent the scale-dependent drift rates and volatility rates of state $X$ and observation $Y$, respectively. $\varphi(s)$ also implies the heterogeneities and physical processes from standard scale to a specific scale, which ~~currently maybe~~may be hard to ~~be~~ formulate. $\sigma(u)$ can be regarded as the stochastic perturbation with respect to scale. Therefore, according to Eq. (12), a state is $X(s_X) = X_0 + \int_{s_0}^{s_X} \varphi(u)du + \int_{s_0}^{s_X} \sigma(u)dW(u)$ in the state space and ~~is~~ $X(s_Y) = X_0 + \int_{s_0}^{s_Y} \varphi(u)du + \int_{s_0}^{s_Y} \sigma(u)dW(u)$ in the observation space. These formulas prove that the value of state varies with the changes of scale.

The scale transformation is only ~~involves~~ involved in the process ~~that~~ of mapping the state vector from state space to observation space. ~~For simplicity, assume the scale-based drift rates of the state and observation do not exist, which leads to $\varphi_X(s) = 0$ and $\varphi_Y(s) = 0$. If the noises are Gaussian, we have $\sigma_X(s) = \sigma_Y(s) = 1$.~~

Based on the above discussion, the ~~differential and~~ integral forms of the state are

$$\sout{dX = dW(s) \text{ and }} X(s_X) = X_0 + \int_{s_0}^{s_X} \varphi_X(s)ds + \int_{s_0}^{s_X} \sigma_X(s)dW(s). \tag{15}$$

For the observation, we have

$$\sout{dY = dW(s) \text{ and }} Y(s_Y) = Y_0 + \int_{s_0}^{s_Y} \varphi_Y(s)ds + \int_{s_0}^{s_Y} \sigma_Y(s)dW(s) \tag{16}$$

In Eq. (15) and Eq. (16), the time $t$ is omitted, and $s_X$, $s_Y$, $X_0$ and $Y_0$ represent the scale of the state space, scale of the observation space, state in $s_0$ and observation in $s_0$, respectively.

The Bayesian equation of data assimilation (Eq. (7)) produces the posterior PDF $p(X|Y)$ that is associated with the likelihood function $p(Y|X)$ and the distributions of the state and observation. Theorem 1 and Eqs. (15)~(16) yield $X \sim N\left(X_0, \int_{s_0}^{s_X} ds\right)$ and $Y \sim N\left(Y_0, \int_{s_0}^{s_Y} ds\right)$ under the condition that the variances exist. In addition, assumption 1 states that the scales vary in one-dimensional space, which results in

$$X \sim N\left(X_0 + \int_{s_0}^{s_X} \varphi_X(s)ds, \int_{s_0}^{s_X} \sigma_X^2(s)ds \sout{s_X - s_0}\right) \tag{17}$$

$$\text{and } Y \sim N\left(Y_0 + \int_{s_0}^{s_Y} \varphi_Y(s)ds, \int_{s_0}^{s_Y} \sigma_Y^2(s)ds \sout{s_Y - s_0}\right). \tag{18}$$

Eq. (17) and Eq. (18) are the prior PDFs of state and observation with respect to scale in state space and observation space, respectively. Compared with the PDFs with respect to time, their expectations are equal to the value at the standard scale, and the variances depend on the differences between the standard scale and the scale in state or observation space. These two prior PDFs are introduced into the Bayesian theorem that is reformulated by scale.

Then, we calculate the posterior PDF.

The scale-dependent observation operator is $H(s, x)$, which suggests that the observation operator and its parameters are both susceptible to the scale. If $H(s, x)$ is defined, its continuous partial derivatives are $H_s(s, x)$, $H_x(s, x)$ and $H_{xx}(s, x)$. In line with Ito's Lemma, we get an estimation of observation in the observation space, which is related to the state $X(s_X)$ defined in the state space

$$H(s_X, X(s_X)) = H(s_0, X_0) + \int_{s_0}^{s_X} H_s(u, X(u))\, du + \int_{s_0}^{s_X} H_x(u, X(u))\, dW(u) + \frac{1}{2}\int_{s_0}^{s_X} H_{xx}(u, X(u))\, du$$

$$= H(s_0, X_0) + \int_{s_0}^{s_X}\left[H_s(u, X(u)) + \frac{1}{2}H_{xx}(u, X(u))\right] du + \int_{s_0}^{s_X} H_x(u, X(u))\, dW(u). \quad (19)$$

Assumption 1 suggests that the observation and ~~model~~ state spaces have the same probability measure; thus, the Brownian motions in these two spaces are equivalent. ~~Let Eq. (16) = Eq. (19)~~Eq. (19) can also be rewritten by replacing $s_0$ with $s_Y$, namely $H(s_X, X(s_X)) = H(s_Y, X(s_Y)) + \int_{s_Y}^{s_X}\left[H_s(u, X(u)) + \frac{1}{2}H_{xx}(u, X(u))\right] du + \int_{s_Y}^{s_X} H_x(u, X(u))\, dW(u)$, and then we obtain

$$~~Y(s_Y) - H(s_X, X(s_X))~~$$

$$~~= Y_0 + \int_{s_0}^{s_Y} dW(u) - \left[H(s_0, X_0) + \int_{s_0}^{s_X} H_s(u, X(u))\, du + \int_{s_0}^{s_X} H_x(u, X(u))\, dW(u) + \frac{1}{2}\int_{s_0}^{s_X} H_{xx}(u, X(u))\, du\right]~~$$

$$~~= Y_0 - H(s_0, X_0) + \int_{s_0}^{s_Y} dW(u) - \left[H(s_X, X(s_X)) - H(s_0, X(s_0))\right] - \frac{1}{2}\int_{s_0}^{s_X} H_{xx}(u, X(u))\, du - \int_{s_0}^{s_X} H_x(u, X(u))\, dW(u)~~$$

$$~~= Y_0 - \left[H(s_X, X(s_X)) + \frac{1}{2}\int_{s_0}^{s_X} H_{xx}(u, X(u))\, du\right] + \left\{\int_{s_0}^{s_Y} dW(u) - \int_{s_0}^{s_X} H_x(u, X(u))\, dW(u)\right\}. \quad (20)~~$$

$$Y(s_Y) - H(s_X, X(s_X))$$

$$= Y(s_Y) - \left[H(s_Y, X(s_Y)) + \int_{s_Y}^{s_X}\left[H_s(u, X(u)) + \frac{1}{2}H_{xx}(u, X(u))\right] du + \int_{s_Y}^{s_X} H_x(u, X(u))\, dW(u)\right]$$

$$= Y(s_Y) - \left[H(s_X, X(s_X)) + \frac{1}{2}\int_{s_Y}^{s_X} H_{xx}(u, X(u))\, du\right] + \int_{s_X}^{s_Y} H_x(u, X(u))\, dW(u). \quad (20)$$

Equation (20) can be regarded as an Ito process, and its drift is

$$Y(s_Y) - \left[H(s_X, X(s_X)) + \frac{1}{2}\int_{s_Y}^{s_X} H_{xx}(u, X(u))\, du\right]~~Y_0 - \left[H(s_X, X(s_X)) + \frac{1}{2}\int_{s_0}^{s_X} H_{xx}(u, X(u))\, du\right]~~. \quad (21)$$

The integral term in Eq. (21) is the difference in the first-order differential observation operator between the state scale $s_X$ and the ~~standard~~ observation scale $s_Y$ ~~$s_0$~~. This term illustrates that the mapping process should consider not only the observation operator but also the first-order differential term when state is mapped to the observation space. The former is typically determined from the literature, whereas the latter was derived in this study for the first time. This result prompted us to further consider the first-order differential of the observation operator when calculating the representativeness error.

The quadratic variation of Eq. (20) is

$$\int_{s_X}^{s_Y} H_x^2(u, X(u))\, du ~~(s_Y - s_0) + \int_{s_0}^{s_X} H_x^2(u, X(u))\, du~~. \quad (22)$$

This equation suggests that the uncertainty in the observation error includes ~~both the difference between scales $s_Y$ and $s_0$ and~~ the change in the observation operator from scale $s_X$ to $s_Y$ ~~$s_0$~~. Therefore, Eq. (21) and Eq. (22) can be combined to produce

$$~~p(Y|X) = N\left(Y_0 - \left[H(s_X, X(s_X)) + \frac{1}{2}\int_{s_0}^{s_X} H_{xx}(u, X(u))\, du\right], (s_Y - s_0) + \int_{s_0}^{s_X} H_x^2(u, X(u))\, du\right). \quad (23)~~$$

$$p(Y|X) = N\left(Y(s_Y) - \left[H(s_X, X(s_X)) + \frac{1}{2}\int_{s_Y}^{s_X} H_{xx}(u, X(u))\, du\right], \int_{s_X}^{s_Y} H_x^2(u, X(u))\, du\right). \quad (23)$$

Based on Eqs. (17), (18) and (23), $p(Y|X)$, $p(X)$ and $p(Y)$ are stochastic functions that depend on the scale; thus, the posterior PDF of the state is scale-dependent as well.

In particular, if $Y$ is a direct observation, which means ~~that~~ the observation is of the same physical quantity and scale as the state, viz. $H(s, X(s)) = X(s)$. The~~n the~~ result becomes

$$~~Y(s_Y) - X(s_X) = \begin{cases} Y_0 - X(s_X) + W(s_Y) - W(s_X), s_Y > s_X \\ Y_0 - X(s_X) + W(s_X) - W(s_Y), s_X > s_Y \end{cases}~~ \quad Y(s_Y) - X(s_X) = Y(s_Y) - X(s_X) + \int_{s_X}^{s_Y} dW(u) \qquad (24)$$

$$\text{and } p(Y|X) = N\{Y(s_Y)~~Y_0~~ - X(s_X), |s_Y - s_X|\}. \qquad (25)$$

In Eq. (24), the integral $\int_{s_X}^{s_Y} dW(u)$ can be regarded as the noise based on the increment of Brownian motion with respect to scale, and its expectation equals zero.

    ~~The quadratic variation in Eq. (22) can be further described by the scale from $s_X$ to $s_Y$. Under the condition $s_Y > s_X$ and because $W(s_Y) - W(s_X)$ and $W(s_X) - W(s_0)$ are independent, the quadratic variation of Eq. (20) is~~

$$~~s_Y - s_X + \int_{s_0}^{s_X} [1 - H_x(u, X(u))]^2 du. \qquad (26)~~$$

~~Similarly, if $s_X > s_Y$, the quadratic variation of Eq. (20) is~~

$$~~\int_{s_0}^{s_Y} \left(1 - H_x(u, X(u))\right)^2 du + \int_{s_Y}^{s_X} H_x^2(u, X(u)) du. \qquad (27)~~$$

    The significance of Eqs. (20)~(~~27~~25) is that the effect of scale on the posterior PDF can be determined quantitatively. In addition to the model error and instrument error (both ~~of them~~ were not introduced explicitly in this study because they have little influence on the error caused by scale transformation), a new type of error in data assimilation was discovered in the

analysis step. The expectation of the posterior PDF may vary with the scale of the state space if $Y$ is an indirect observation, and the variance of the drift depends on the difference between $s_Y$ and $s_X$ (based on Eq. (~~26~~22) ~~and Eq. (27)) or among $s_0$, $s_Y$ and $s_X$ (based on Eq. (22))~~. In addition, if $Y$ is a direct observation (Eq. (24) and Eq. (25)), the expectation of the posterior PDF is the difference between $Y$ and $X$, and the variance is equal to the increment of Brownian motion with respect to the scale. Additionally, if the results are not derived from assumption 1, i.e., the scale varies randomly, the posterior PDF is more

complex because the Jacobian matrix in the Lebesgue integration of scale transformation is arbitrary.

### 3.4 Example~~s~~: the stochastic radiative transfer equation (SRTE)

    To explicitly show how the stochastic scale transformations impact ~~on~~ assimilation, we introduce an illustrative example based on the scales presented in Figure 1. ~~Assuming~~ Assume that in the analysis step, the state ~~is with~~has the standard ~~scale scale~~ $s_0$, whose observation footprint is the unit square $A_0$. If the scale of observation space is $s_{C1}$ and its observation footprint

is the disc $C_1$, then the Jacobian matrix of the transformation between the scales of the state space and observation space is not

diagonal according to the statements in Sect. 3.1, leading the two scales ~~do~~ to not obey the one-dimensional rule ~~and~~, which is against assumption 1. However, if ~~let~~ the scales of ~~of~~ state space and observation space are $s_{C1}$ and $s_{C2}$, respectively, ~~the~~ assumption 1 is met, and it can be ~~counted~~ determined that $s_X = s_{C1} = \frac{\pi}{4} s_0$ and $s_Y = s_{C2} = \frac{9\pi}{4} s_0$.

Now the scales of state space and observation space obey the one-dimensional rule, and ~~then~~ we further presume that the

5 measure space $\Omega$ in Figure 1 is free of ~~the~~ spatial heterogeneities and dynamic process variations depending on scale. Consequently, the drift rate $\varphi(s) = 0$. If ~~denoting~~ the value of state in the standard scale is denoted as $X_0$ and assuming that $\sigma(s) = 1$, then the prior PDF of state is $X \sim N\left(X_0, \frac{\pi}{4} s_0 - s_0\right)$ ~~according to Eq. (17).~~ according to Eq. (17). ~~Noting~~ It should be noted that $\frac{\pi}{4} s_0 - s_0$ is not a real number and is only used to ~~indicates~~ indicate the variation when the scale changes.

If $H(s, X(s)) = X(s)$, the observation ~~is~~ has the same physical quantity as the state, and according to Eq. (25), the likelihood

function is $p(Y|X) = N\{Y(s_Y)\overline{Y_U} - X(s_X), |s_Y - s_X|\} = N\{Y(s_Y)\overline{Y_U} - X(s_X), |s_{C2} - s_{C1}|\} = N\left\{Y(s_Y)\overline{Y_U} - X(s_X), \left|\frac{9\pi}{4} s_0 - \frac{\pi}{4} s_0\right|\right\}$.

To formulate the likelihood function in the case that the observation is different from the state, the SRTE will be employed in the following text. The SRTE is a stochastic integral-differential equation that describes the radiative transfer phenomena through a stochastically mixed immiscible media. Scientists have developed analytical or numerical methods for finding the

15 stochastic moments of the solution, such as the ensemble- averaged ~~or~~ and the variance of the radiation intensity (Pomraning, 1998; Shabanov et al., 2000; Kassianov et al., 2011).

Consider the general expression of the SRTE (~~leave~~ leaving out the scattering and emission),

$$-\mu \frac{dI(\tau)}{d\tau} = -I(\tau) , \tag{2826}$$

where $I(\tau)$, $\mu$ and $\tau$ are the radiation intensity, coefficient of radiation direction and optical depth, respectively. ~~The analytical~~

20 ~~solution of Eq. (28) is $I(\tau) = -I(0)e^{\tau/\mu}$.~~

To tie into more substantial random optical properties of the transfer media, such as absorption and scattering, the optical depth $\tau$ is assumed to be stochastic. ~~So it~~ This suggests that the optical depth is a scale-dependent Ito process and can be expressed as

$$d\tau(s) = \varphi_\tau(s)ds + \sigma_\tau(s)dW(s), \tag{2927}.$$

This causes the radiation intensity to depend on scale.

The analytical solution of Eq. (26) is $I(\tau) = I_0 e^{\tau/\mu}$, where $I_0 = I(\tau(s_0))$.

SRTE can be considered as a concrete instance of a stochastic observation operator by defining $H(s, x(s)) = I(x) = I_0 \cancel{I(0)} e^{x/\mu}$. Therefore, $H_s(s, x(s)) = 0$, $H_x(s, x(s)) = \frac{1}{\mu} I_0 \cancel{I(0)} e^{x/\mu}$ and $H_{xx}(s, x(s)) = \frac{1}{\mu^2} I_0 \cancel{I(0)} e^{x/\mu}$. Based on Ito's Lemma,

$$
\begin{aligned}
dI(\tau(s)) = dH(s, \tau(s)) &= H_s(s, \tau(s)) ds + H_x(s, \tau(s)) d\tau(s) + \frac{1}{2} H_{xx}(s, \tau(s)) d\tau(s) d\tau(s) \\
&= \frac{1}{\mu} I_0 e^{\tau(s)/\mu} d\tau(s) + \frac{1}{2\mu^2} I_0 e^{\tau(s)/\mu} d\tau(s) d\tau(s) \\
&= \frac{1}{\mu} I(\tau(s)) d\tau(s) + \frac{1}{2\mu^2} I(\tau(s)) d\tau(s) d\tau(s) \\
&= \left( \frac{1}{\mu} I(\tau(s)) \right) \sigma_\tau(s) dW(s) + \left( \frac{1}{\mu} I(\tau(s)) \right) \varphi_\tau(s) ds + \left( \frac{1}{2\mu^2} I(\tau(s)) \right) \sigma_\tau^2(s) ds \\
&= \left( \frac{\sigma_\tau^2(s)}{2\mu^2} + \frac{\varphi_\tau(s)}{\mu} \right) I(\tau(s)) ds + \left( \frac{\sigma_\tau(s)}{\mu} \right) I(\tau(s)) dW(s)
\end{aligned}
$$
. (~~30~~28)

The ~~Radiation~~ radiation intensity is a scale-dependent Ito process. The difference between Eq. (~~30~~28) and the general Ito process is that there is a primitive function $I(\tau(s))$ in the integral term. Therefore, the uncertainty of the radiation intensity is more complex because it is related to both the change of scale and the primitive function.

Integrating both sides of Eq. (~~30~~28) yields the general solution of the radiation intensity,

$$
I(\tau(s)) = C \cdot \exp\left[ \int \left( \frac{\sigma_\tau^2(s)}{2\mu^2} + \frac{\varphi_\tau(s)}{\mu} \right) ds + \int \left( \frac{\sigma_\tau(s)}{\mu} \right) dW(s) \right],
$$
________(~~31~~29)

where the constant $C \in R$. Eq. (~~31~~29) further indicates that $I(\tau(s))$ is a scale-dependent Ito process.

 Considering that the optical depth $\tau$ is the state, the radiation intensity $I$ is the observation and $I(\tau(s))$ is the observation operator, ~~then~~the results in Sect. 3.3 could easily be applied here. For example, Eq. (20) and Eq. (23) become

$$
Y(s_Y) - H(s_X, X(s_X)) = I(\tau(s_Y)) - I(\tau(s_X)) + \frac{1}{2} \int_{s_X}^{s_Y} \frac{1}{\mu^2} I(\tau) du + \int_{s_X}^{s_Y} \frac{1}{\mu} I(\tau) dW(u), \quad (30)
$$

$$
p(Y|X) = N\left( I(\tau(s_Y)) - I(\tau(s_X)) + \frac{1}{2} \int_{s_X}^{s_Y} \frac{1}{\mu^2} I(\tau) du, \int_{s_X}^{s_Y} \frac{1}{\mu^2} I^2(\tau) du \right). \quad (31)
$$

~~the above results in Sect. 3.3 (For example, Eq. (20)) could be easily applied here to study the~~ Then, the posterior PDF of the data assimilation can be determined by Eq. (27), (29) and (31).

## 3.5 Extension to n-dimensional data assimilation

In the above discussion, we assumed that only one state existed in data assimilation. However, numerous states typically exist. This section further introduces the **product spaces** to extend the one-dimensional stochastic data assimilation to n-dimensions.

5 Assume that the independent states $X_k$ are the variables of the measure spaces $(\Omega_k, \mathcal{F}_k, \mu_k)$, $k = 1, 2, \ldots, n$, and $(\Omega^n, \mathcal{F}^n)$ is the product space, where $\Omega^n = \prod_{k=1}^{n} \Omega_k$ and $\mathcal{F}^n = \prod_{k=1}^{n} \mathcal{F}_k$. According to Fubini's theorem (Billingsley, 1986), only one product measure $\mu^n$ in $(\Omega^n, \mathcal{F}^n)$ exists, such that $\mu^n(\prod_{k=1}^{n} A_k) = \prod_{k=1}^{n} \mu_k(A_k)$, where $A_k \in \mathcal{F}_k$.

We define the state vector $X^n = (X_1, X_2, \ldots, X_n)^T$ as a variable vector of the product measure space $(\Omega^n, \mathcal{F}^n, \mu^n)$. In particular, if all the scales obey the one-dimensional rule, we have

$$\mu^n\left(\prod_{k=1}^{n} A_k\right) = \prod_{k=1}^{n} \xi_k^2 \mu_0(A_k) = \left(\prod_{k=1}^{n} \xi_k\right)^2 \mu_0^n\left(\prod_{k=1}^{n} A_k\right).$$

This expression proves that the product measure also obeys a one-dimensional rule. However, the above results may not hold without the assumption that the states $X_k$ are independent.

As discussed in Sect. 2.1, the Lebesgue measure $m^2$ is a measure and the triple $(R^2, \mathcal{L}^2, m^2)$ is a measure space. Therefore, the above extension is reasonable.

The analysis of a single state can also be applied to finite multiple states in the product measure space.

# 4 Discussion & ~~Conclusions~~conclusions

## 4.1 Discussion

Our study offered a stochastic data assimilation framework to formulate the errors that are caused by scale transformations. The necessity of the methodology, the difference ~~to~~ from previous works by other investigators, and the advantages and

20 limitations of this study are discussed as follows.

The reasons that the methodology focuses on a stochastic framework are as follows~~:~~. First, the stochastic data assimilation framework is essentially consistent with the conception~~s~~ of scale and scale transformation~~. Both of them~~; both are associated with corresponding measure spaces $(\Omega, \mathcal{F}, \mu)$. Therefore, it is natural to regard the state space and observation space as two different measure spaces, ~~respectively,~~ and each element of state (or observation) vector can be seen as a geophysical variable

that ~~mapping~~ maps the state (or observation) measure space onto $R$. Correspondingly, as the integrals of random processes with respect to random processes, stochastic calculus was ultimately adopted ~~ultimately~~. Second, using stochastic calculus can also formulate the errors caused by scale transformations. The study proceeds with and improves the understanding of

representativeness error in terms of scale. The resultsResults did not only prove the conventional point that the uncertainties of these errors mainly depend on the differences between scales, but also indicated that the first-order differential of the nonlinear observation operator should also be incorporated in representativeness error. Third, the error caused by scale transformation was presented in a general form. The drift and quadratic variation of error were formulated by Eq. (21) and Eq. (22), respectively, and both defined the probability distribution space of $p(Y|X)$. Last, stochastic calculus can be extended to meet a general scale transformation and formulate the corresponding representativeness error. This, which was unattainable in previous work. For example, if the scale changes randomly, say, from an irregular footprint to another irregular footprint, the stochastic equation can offer a multiple- integral to present this kind of atype of scale transformation, such as $V(x,y) = V_0 + \int_{Y_0}^{Y} \int_{X_0}^{X} \varphi(x,y)dxdy + \int_{Y_0}^{Y} \int_{X_0}^{X} \sigma(x,y)dW_1(x)dW_2(y)$ , where $W_1(x)$ and $W_2(y)$ are two independent Brownian Motionmotions.

The significant innovation of this work is as follows.: We developed a more rigorous formulation of the scale and the scale transformation based on Lebesgue measure, which places the related conceptions in a rigorous mathematical framework and then conduces provides a new understanding of the errors caused by scale transformation. In addition, due to the Ito process-formed state and observation, a stochastic data assimilation framework was proposed by considering the nonlinear operators, heterogeneity of a geophysical variable and a general Gaussian representativeness error. Scale The scale transformation is also nonlinear if the one-dimensional rule is not involvedapplied. Additionally, Ito processes-formed state and observation offer the drift rate (i.e., $\varphi(s)$ in Eq. (10)) to formulate the heterogeneity associated with scale transformation. It also permits the representativeness error to be general Gaussian in this framework. If all the integrands in Eq. (13) and Eq. (14) are nonlinear functions instead of constants (in this study we let $\varphi_X(s) = 0, \varphi_Y(s) = 0$ and $\sigma_X(s) = \sigma_Y(s) = 1$ for simplicity), then these two equations are can be integrated over the field of Brownian motion, and state and observation are the general Gaussian processes of scale. Based on these functions, the representativeness error is a general Gaussian process.

As a theoretical exploration towards scale transformation and stochastic data assimilation, there is still big much room for improvement. First, we reduced the scale transformation by the one-dimensional rule, and let the variables in data assimilation evolve regularly according to assumptions 1~3. So; thus, only the ideal result was investigated. Therefore, an in-depth and comprehensive exploration should be conducted in the future to describe other situations in the real world. However, the use of either an arbitrary scale transformation or the geophysical variable without ignoring the drift rates will deduce obtain lengthy results. Therefore, the second improvement focuses on how to make the formulation more concise. Last, noting that all the results in our framework were given in terms of probability, it is necessary to implement the real-world applications of these theoretical results, such as introducing some concrete dynamic models to formulate the Ito process-formed geophysical variable of scale.

## 4.2 Conclusions

In this study, we mainly addressed two basic problems associated with scale transformation in ~~earth~~ Earth observation and simulation. First, we produced a mathematical formalism of scale and scale transformation by employing measure theory. Second, we demonstrated how scale transformation and its associated errors could be presented in a stochastic data assimilation framework.

We revealed that the scale is the Lebesgue measure with respect to the observation footprint or model unit. ~~Scale~~ The scale is related to the shape and size of a footprint, and scale transformation depends on the spatial change between different footprints. We then defined the geophysical variable, which further considers the heterogeneities and physical processes. A geophysical variable consequently expresses the spatial average at a specific scale.

We formulated the expression of scale transformation and investigated the error structure that is caused by scale transformation in data assimilation using basic theorems of stochastic calculus. ~~Formulations~~ The formulations explicate that the first-order differential of the nonlinear observation operator should be considered in representativeness error, and the uncertainty of representativeness error is directly associated with the difference between scales. A concrete physical models (SRTE) was introduced to demonstrate the results when observation operator is nonlinear. ~~Extension the results to n-dimensional stochastic data assimilation was also presented.~~

This work conducted a theoretical exploration of formulating the errors caused by scale transformation in a stochastic data assimilation framework. We hope that the stochastic methodology can ~~essentially~~ benefit the study ~~on~~ of these errors.

## 5 Notation

### 5.1 Basic notations

| | |
|---|---|
| $\Omega$ | ~~Non~~ Non-empty space |
| $\mathcal{F}$ | $\sigma$-algebra |
| $\mu$ | Measure |
| $dV$ | Variable process |
| $W(s)$ | Brownian motion |
| $(\Omega, \mathcal{F}, \mu)$ | Measure space |
| $I^n$ | N-dimensional Lebesgue volume |
| $m^n$ | Lebesgue measure or an outer measure on $R^n$ |
| $\mathcal{L}^n$ | Lebesgue $\sigma$-algebra of $R^n$ |
| $\int f dm^n$ | Lebesgue integral |
| $|J(\cdot)|$ | Jacobian determinant |
| $(\Omega^n, \mathcal{F}^n)$ | Product space |

## 5.2 New notations

| Notation | Name | Explanation | Index |
|---|---|---|---|
| $s$ | Scale | The observation footprint or model unit to observe or model a geophysical variable | Sect. 1 & Sect. 3.1 |
| $A_0$ | Unit square in $R^2$ | | Sect. 3.1 |
| $s_0$ | Standard scale | A Lebesgue integral ~~of~~ where $A_0$ is the unit area | Sect. 3.1 |
| | One-dimensional rule | Two scales are geometrically similar | Eq. (8) |
| $V$ | Geophysical variable | Estimation of a variable at a specific scale | Sect. 3.2 |
| $dX$ | State process | Ito process-formed state | Eq. (13) |
| $dY$ | Observation process | Ito process-formed observation | Eq. (14) |
| $X_0$ | State in $s_0$ | | Eq. (15) |
| $Y_0$ | Observation in $s_0$ | | Eq. (16) |
| $s_X$ | Scale of state space | | Eq. (15) |
| $s_Y$ | Scale of observation space | | Eq. (16) |

**Acknowledgements**

We thank the editor-in-chief of NGP, Prof. Talagrand, and his kind help and valuable comments on our manuscript. We also thank Dr. van Leeuwen and another anonymous reviewer for their valuable comments and suggestions. This work was
5 supported by the NSFC projects (grant numbers 91425303 & 91625103) and the CAS Interdisciplinary Innovation Team of the Chinese Academy of Sciences.

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
