# Peer review of "Formulation of Scale Transformation in a Stochastic Data Assimilation Framework"

_Nonlinear Processes in Geophysics, 2016_

## Referee Comment (RC1) · Anonymous Referee #1 · 12 Jul 2016

We all carry with us the notion of "scale," and we have been performing scale transformations on mathematical formulations of problems since we were beginning students. Upon reflection, few would dispute the proposition that quantitative evaluation of uncertainty is fundamental to data assimilation, and that our evaluation of uncertainty is highly scale dependent. I don't recall seeing a rigorous treatment of the role of scale in uncertainty quantification as applied to data assimilation, so this article has the potential of introducing a useful new idea to the community. Setting down a rigorous definition of scale and scale transformation, and showing how the definition of scale can be applied to quantification of uncertainty would be a worthwhile contribution to the field of data assimilation. In addition, I welcome the explicit introduction of measure theory and stochastic calculus to the data assimilation community, which sees very little of either, especially the former.

[Figure]

Unfortunately, the exposition is extremely hard to follow, and, after reading the article through carefully, I still don't understand what the authors are doing. Part of the problem is probably the language. The manuscript would profit considerably if the authors could find a sympathetic native English speaker to read it over. More importantly, I find much of the exposition puzzling. I don't know the book by Billingsley. In my day students in the USA learned measure theory from the texts by Bartle and Royden, and, relative to my background, much of the material is written very unconventionally. I found this manuscript very hard to read, as much of it seemed to conflict with my most basic intuitions.

The usual intuition for the concept of measure is that measure is a generalization of the concepts of length, area and volume, and is thus a scalar valued set function. The authors' response at the end of section 2.1 to Prof. Talagrand's comment is inadequate. There is nothing intrinsically wrong with choosing a vector valued measure, but that choice requires more explanation than simply "the measure correspondingy turns to . . ." The authors should explain why they want to define measure as a vector valued set function, rather than simply defining the measure of a rectangle in Euclidean space as its area. Again, maybe Billingsley defines it differently, but the Lebesgue measure of a rectangle is its area, not a vector whose components are the lengths of its sides as the authors assert on line 16 of page 6.

I do not understand the definition of scale. First, measure is a function whose domain is the sigma field F, as noted at the very beginning of section 2.1. The integrals in the definition of scale, line 16, page 7, don't make sense to me. A_0 is a specific set. (Pardon my TeX, I don't know how to make superscripts, subscripts or special characters). In the analysis texts I learned from, \mu (A_0 ) is the area of the set A_0. I don't understand the expression \mu (A_0 )dA_0. I cannot make sense of the second integral. The domain of the measure \mu is the \sigma-field F, as noted in the beginning of section 2.1. The Lebesgue measure is not a point function.

It would help a great deal if there were more explanation of figure 1. In particular, after

reading and rereading the last paragraph on page 8, I can't understand how $C_2$ can have the same measure as $C_1$ and $C_3$, and $D_1$ has the same measure as $D_2$, though they have the same "scale." The problem may be the terminology: As I recall my long-ago analysis classes, the common intuition for measure was that measure corresponds to area, and, in particular, the Lebesgue measure of a geometrical figure in the plane is its area.

Finally, the manuscript seems inconsistent with itself. As examples, consider the abstract. "...measure theory was used to propose [a definition of] spatial scale ... [and the] Jacobian matrix [was used] to describe the change of scale. The Jacobian matrix is introduced on page 7 in the well known change of variables formula, change of scale by the Jacobian matrix is defined on page 8, and the Jacobian is not mentioned again until the summary. No further discussion of the effects of change of scale appears. Again, in the abstract, "...the variation range of this type of error is proportional to the scale gap, ..." I'm sure I'm not the only reader for whom the phrase "scale gap" conjures up ideas of inertial range from turbulence theory and similar notions. The term "scale gap" is never mentioned in the body of the article.

The Bayesian expression of DA in terms of the stochastic calculus appears in many places. The authors should consult the volume by Jazwinski and the recent work of P. J. van Leeuwen and M. Bocquet.

---

## Author Comment (AC1) · 28 Jul 2016

**Response to Reviewer #1**

We thank the anonymous Referee #1 for taking his/her valuable time to review our manuscript and provide us some very thoughtful and constructive comments. Here are the point-by-point responses to the comments.

**1.  The language problem.**

"Unfortunately, the exposition is extremely hard to follow, and, after reading the article through carefully, I still don't understand what the authors are doing. Part of the problem is probably the language. The manuscript would profit considerably if the authors could find a sympathetic native English speaker to read it over."

**Response:** Considering that we are non-native English speakers, our manuscript was thoroughly edited by a manuscript service company named American Journal Experts before submission (Certificate Verification Key: 53B9-681E-0C9B-BC77-FF4C). However, the exposition can be improved further. Overall, we tried to introduce our ideas in the scale problem with some classic theorem, so some expressions may become a little complicate to understand. We ensure that we will provide a modified version of this manuscript with higher quality English expression in the future.

**2.  The references problems.**

"More importantly, I find much of the exposition puzzling. I don't know the book by Billingsley. In my day students in the USA learned measure theory from the texts by Bartle and Royden, and, relative to my background, much of the material is written very unconventionally."

"The Bayesian expression of DA in terms of the stochastic calculus appears in many places. The authors should consult the volume by Jazwinski and the recent work of P. J. van Leeuwen and M. Bocquet."

**Response:** The literatures written by Billingsley, Bartle and Royden are all classic works of measure theory (Maybe the texts you mentioned are *The Elements of Integration and Lebesgue Measure* by Bartle R. G. and *Real analysis* by Royden H. L.), and they were all highly cited according to the search results of Google Scholar. In addition, I also got many help from the volume titled "Stochastic

Processes and Filtering Theory" by Jazwinski A. H. during my study, which will be listed in the 'References' of revised manuscript. And we also thanks for the recommended literature of the Bayesian Data Assimilation (DA) in terms of the stochastic calculus. They introduced the latest advances in this research field, and the related papers will be cited in our present and future works.

**3. Why the measure in manuscript was defined as a vector valued set function.**

"The usual intuition for the concept of measure is that measure is a generalization of the concepts of length, area and volume, and is thus a scalar valued set function. The authors' response at the end of section 2.1 to Prof. Talagrand's comment is inadequate. There is nothing intrinsically wrong with choosing a vector valued measure, but that choice requires more explanation than simply "the measure correspondingly turns to …" The authors should explain why they want to define measure as a vector valued set function, rather than simply defining the measure of a rectangle in Euclidean space as its area. Again, maybe Billingsley defines it differently, but the Lebesgue measure of a rectangle is its area, not a vector whose components are the lengths of its sides as the authors assert on line 16 of page 6."

**Response:** It is true that in the classic literatures on measure theory, measure is a scalar valued set function. However, when it comes to spatial scale, more information is necessary instead of one scalar value. For example, suppose that rectangle A is 4 meters long and 1 meter wide, and rectangle B is 2 meters long and 2 meters wide. Then if we define the area of rectangle is measure, their measures are equal but the shape of spatial scale is missed. So in our study both the length and width of rectangle composed the new measure, and they are also scalar values. However, it's our mistake that we use the notation of vector to present the new measure, and it has made confusion that the new measure is a mismatch with its basic definition. Therefore, in the revised manuscript, we will replace the old expression with the notation $\{a, b\}$.

Some texts will be updated correspondingly:

1) In line 3 of page 6, we will explain why we use the new measure. The sentence "In this case, the measure correspondingly turns into $\mu(A) = (a, b)^T, a, b \in [0, \infty)$, which should also obey the countable additivity." will be modified as "In this case, the subset of $\Omega$ evolves across two directions because $\Omega$ is two-dimensional. Therefore the measure should be of double scalar values so that the sufficient information of the subset can be presented. Correspondingly, we define the measure as $\mu(A) = \{a, b\}, a, b \in [0, \infty)$, which should also obey the countable additivity."

2) In line 16 of page 6, we also used the new form of measure to define the rectangle measure, not Lebesgue measure. The equation $\mu_{iii}(A) = b - a = (b_1 - a_1, b_2 - a_2)^T$ goes to

$\mu_{iii}(A) = b - a = \{b_1 - a_1, b_2 - a_2\}$.

3) Other changes of notations:

| Position | Original Text | Revised Text |
|---|---|---|
| line 20 of page 6 | $\frac{\sqrt{\pi}}{2}\left(b_1 - a_1, \quad b_2 - a_2\right)^T$ | $\left\{\frac{\sqrt{\pi}}{2}\left(b_1 - a_1\right), \quad \frac{\sqrt{\pi}}{2}\left(b_2 - a_2\right)\right\}$ |
| line 8 and 9 of page 7 | $(1,1)^T$ | $\{1,1\}$ |
| line 10 of page 7 | $\frac{2}{\sqrt{\pi}}\left(b_1 - a_1, \quad b_2 - a_2\right)^T$ | $\left\{\frac{2}{\sqrt{\pi}}\left(b_1 - a_1\right), \frac{2}{\sqrt{\pi}}\left[b_2 - a_2\right]\right\}$ |
| line 10 of page 7 | $\left(b_1 - a_1, \quad b_2 - a_2\right)^T$ | $\{b_1 - a_1, \quad b_2 - a_2\}$ |

**4. The definition of scale.**

"I do not understand the definition of scale. First, measure is a function whose domain is the sigma field F, as noted at the very beginning of section 2.1. The integrals in the definition of scale, line 16, page 7, don't make sense to me. A_0 is a specific set. (Pardon my TeX, I don't know how to make superscripts, subscripts or special characters). In the analysis texts I learned from, \mu (A_0) is the area of the set A_0. I don't understand the expression \mu (A_0)dA_0. I cannot make sense of the second integral. The domain of the measure \mu is the \sigma-field F, as noted in the beginning of section 2.1. The Lebesgue measure is not a point function"

**Response:** There are 2 elements should be concerned about in the definition of scale. The first one is the rectangular referential element $A_0$, which represents the unit of the subset and $\Omega$. For example, we can define the unit length and unit area as $A_0$ for one- and two-dimensional space, respectively. And we also introduced the other elementary concept of representative region $A$, which is the cumulative amount of $A_0$. And of course, $A \in \mathcal{F}$. Then the measure function $\mu(\cdot)$ means to calculate a specific feature of its domain, such as the area, the perimeter or others. So the scale $\mu(A)$ is a description of representative region $A$, and the unit of scale depends on the referential element $A_0$.

In line 16, page 7, we used the integral expression of scale. The reasons were that, as stated above, $A$ is the cumulative amount of $A_0$, and the measure function is with the countable additivity (the second condition of measure, line 21, page 5). So the measure with a domain $A$ can be calculated by the cumulative measures with $A_0$ in $A$, which confirms the first integral (it could be more clear if $A \gg A_0$). And if we want to get the reduction formula, then it's natural to replace the surface integral with a double integral, like $\iint_A \mu(A_0)dA_0 = \iint_A f(x,y)dxdy$. If $\mu(\cdot)$ is the Lebesgue measure, then $f(x,y) = 1$. Because here the measure corresponds to the area, its output is one-dimensional scalar. However, we let $f(x,y) = \mu(\cdot)$, which make the second integral cannot be deduced – partly because that generally $\mu(\cdot)$ is not equal to $f(x,y)$, and partly because the measure function, as you mentioned, is definitely not a point function, so $\mu(\cdot)$ is invalid in the second integral.

To correct this mistake, there should be some following changes:

1) From line 16 to line 18, page 7, the new content is "the scale is $s = \mu(A) = \iint_A \mu(A_0)dA_0$.

From a geometric perspective, the measure refers to the shape of the observation region, and the scale further indicates the size of the region; therefore, the scale increases with increases in the value of the measure. Specifically we further define that the measure is the area of its domain of integration, then $s = \mu(A) = \iint_A \mu(A_0)dA_0 = \iint_A dxdy$. This equation simplifies the measure by replacing the surface integral with a double integral, and will be applied to the following studies."

2) Related equations should be also changed. The equations in line 20, page 7 will be $s_1 = \mu_1(A_1) = \iint_{A_1} \mu_1(A_0)dA_0 = \iint_{A_1} dxdy$ and $s_2 = \mu_2(A_2) = \iint_{A_2} \mu_2(A_0)dA_0 = \iint_{A_2} dudv$, respectively. The equation in line 1, page 8 will be $s_1 = \iint_{A_1} dxdy = \iint_{A_2} |J(u,v)|dudv$. And the Eq. (1) in line 4, page 8 will be $s_1 = \xi^2 \iint_{A_2} dudv = \xi^2 \mu_2(A_2) = \xi^2 s_2$. The equation in line 2, page 9 will be $s_0 = \mu_0(A_0) = \iint_{A_0} dxdy = 1$.

5. **About figure 1.**

"It would help a great deal if there were more explanation of figure 1. In particular, after reading and rereading the last paragraph on page 8, I can't understand how C_2 can have the same measure as C_1 and C_3, and D_1 has the same measure as D_2, though they have the same "scale." The problem may be the terminology: As I recall my long-ago analysis classes, the common intuition for measure was that measure corresponds to area, and, in particular, the Lebesgue measure of a geometrical figure in the plane is its area."

**Response:** I feel very sorry because the instruction of this figure is not enough to make you understand, and also I think there may be some inconsistency problems of terminology between us. In my manuscript, measure refers to the function $\mu(\cdot)$, and its output $\mu(A)$ is scale. So measure is abstract, it becomes a real value when its argument (or domain) is confirmed. As noted in line 13 and 14, page 8, $C_1, C_2 \ and \ C_3$ have the same measures because they all calculate the area of the inscribed circle in a square region. And the outputs of measures cannot be obtained until the square regions are confirmed. Therefore, the output, which was defined as scale in our manuscript, is related to both the measure function and the function argument. The scale of $C_2$

is larger than the scales of $C_1 \ and \ C_3$ because the square region of $C_2$ is bigger.

However, as you stated, more explanation of figure 1 can help to make our manuscript more clear, so we decide to add more information in here. In addition, in order not to cause confusion, we replace the "measure" by "measure function" in the update version of explanation.

Based on the text from line 10 to line 16, page 8, the new explanation is as follows:

"The measure space $\Omega = [\alpha, \beta] = [x, y: 0 \leq x \leq 4, \ 0 \leq y \leq 4]$ is regularly divided by a referential element defined with unit area. Let $\mu_{C1}, \mu_{C2} \ and \ \mu_{C3}$ be the measure functions of the disc measurements $C_1, C_2 \ and \ C_3$, which present the calculation function of the area of the inscribed circle in a square region; and $\mu_{D1}, \mu_{D2}$ be the measure functions of the diamond measurements $D_1 \ and \ D_2$, which are also to calculate the area of the inscribed diamond in a square region, as shown in Figure 1. Then $\mu_{C1} = \mu_{C2} = \mu_{C3}$ because they are the same functions. And based on the definition, scale is related to both the measure function and the size of representative region. Therefore, the scale of $C_2$ is not equal to the two other scales because of their representative regions are different. However, their scales are in a one-dimensional law because their measure functions are identical and the Jacobian matrices are diagonal. Similarly, we have $\mu_{D1} = \mu_{D2}$; their scales are also different but are in a one-dimensional law. In addition, the value of scale is also based on the referential element, which is defined by the unit area. So if the referential element is changed, such as to increase to twice of unit area, the scale is also increased proportionally. However, scales which are in a one-dimensional law will still keep their relationship intact, regardless of whether the referential element changed or not."

**6.  Inconsistency problems.**

"Finally, the manuscript seems inconsistent with itself. As examples, consider the abstract. "…measure theory was used to propose [a definition of] spatial scale …[and the] Jacobian matrix [was used] to describe the change of scale. The Jacobian matrix is introduced on page 7 in the well known change of variables formula, change of scale by the Jacobian matrix is defined on page 8, and the Jacobian is not mentioned again until the summary. No further discussion of the effects of change of scale appears. Again, in the abstract, "…the variation range of this type of error is proportional to the scale gap, …" I'm sure I'm not the only reader for whom the phrase "scale gap" conjures up ideas of inertial range from turbulence theory and similar notions. The term "scale gap" is never mentioned in the body of the article."

**Response:** There are mainly two inconsistency problems you mentioned. The first one is why the Jacobian matrix and the change of scale disappeared after the Sect. 2. Actually, they were not omitted, they were simplified by the one-dimensional law (defined in line 5, page 8) to suit stochastic calculus. And then we used this simplified version of scale change to investigate the uncertainties in DA (we also mentioned this problem in the second paragraph of Sect. 5). Although it maybe the simplest case that how the change of scale can influence the evolution of uncertainties in DA, the results were still complicated and some new components of uncertainty were discovered (Eq. 21~27). However, it is comprehensive and universal to study the change of scale by Jacobian matrix, and it will be launched in our following work, but not in this study. The second one is the phrase "scale gap" in the abstract. I'm sorry for the trouble that this word did. Here the term "scale gap" stands for the quadratic variation between $s_X$ $and$ $s_Y$ (see Eq. 25~27). However, it is hard to fully explicate them in the abstract, so we had to use the term "scale gap". Maybe it's better to replace it with "the difference between scales" in next time.

---

## Referee Comment (RC2) · 31 Aug 2016

Although this is a potentially interesting paper I found it extremely hard to read. I realise that this is partly true because it stretches my knowledge of mathematics, but, as mentioned by others, the language is also not standard. A major revision is needed to make this readable for the interested NPG audience.

There are several recent works that describe the filtering problem for stochastic processes, and I'm not sure the results of the authors are new in this respect. An example is the book by Bain and Crisan, Fundamentals of Stochastic Filtering, doi:10.1007/978-0-387-76896-0. See also the articles by e.g. Stuart, at http://www2.warwick.ac.uk/fac/sci/maths/people/staff/andrew_stuart/cv/.

On the scale issue, it is not entirely clear to me what the issue is. (I apologise to the

authors.) Is the main issue that observations and numerical models represent reality at different scales? That would make sense. However, since point observations do not exist that problem can be treated directly via the likelihood in Bayes Theorem, and I don't see the need for the measure theory developed here. Furthermore, much work has been done in the data-assimilation community on this issue, called representation error. References that might be useful are Hodyss and Nichols Tellus A 2015, 67, 24822, http://dx.doi.org/10.3402/tellusa.v67.24822, and perhaps Van Leeuwen (2015), Representation errors and retrievals in linear and nonlinear data assimilation. Q.J.R. Meteorol. Soc., 141: 1612–1623. doi:10.1002/qj.2464. See also the excellent work by Bocquet et al. 2011. Bayesian control space for optimal assimilation of observations, I: Consistent multiscale formalism. Q. J. R. Meteorol. Soc. 137: 1340–1356.

My suggestion is that the authors consider these references and reconsider their findings in light of those works. What has been missed by the examples of papers referred above? Why is the new framework needed, using language that the average NPG reader interested in data assimilation can understand?

---

## Editor Comment (EC1) · O. Talagrand (Editor) · 4 Sep 2016

Dear Drs Feng Liu and Xin Li,

As you must have seen, two referees have sent their comments on your paper and the Interactive Discussion of the latter has now been closed.

Referee 1 has remained anonymous, while Referee 2 has let his name known. He is Prof. P. J. van Leeuwen, from Reading University in the UK. While both of them say that there may be potential interest in your paper, they also say that they have found it very hard to understand what you have done (I can say the same for me). Both ask for a major revision of your paper, and both mention specific references of books and papers that deal with the question of filtering for stochastic processes.

You have already submitted a response to the comments by Referee 1. His/her comments bear mostly on the use you make of the notions of measure and scale. But concerning the notations you use, with which both referees (as well as myself) have had difficulties, I stress that the minor changes you describe in items 1-3) of part 3 of your response will not solve those difficulties. Your response must be given at a more fundamental and conceptual level.

I follow the suggestion of the referees, and I ask you to prepare a major revision of your paper. Please clarify the notions of measure and scale you introduce, and the use you make of them. The difficulties that the referees (and I) have met may result in part from inconsistencies between your vocabulary and notations and what we are used to. So please pay particular attention to the references given by the referees, and make your new version as consistent as possible with the existing literature. In particular, should you introduce notions that you consider as new, explain precisely in what they are new.

Please prepare your new version in strict agreement with the instructions you have received from the Editorial Office of *Nonlinear Processes in Geophysics* (*NPG*). Your revised version will be submitted to further review, either by the two referees of the first version, or by other referees. As you must have been informed, submission of a revised manuscript does not necessarily ensure publication in *NPG*.

I thank you for having submitted your paper to *Nonlinear Processes in Geophysics*, and look forward to receiving a new version.

With regards,

Olivier Talagrand
Editor
Nonlinear Processes in Geophysics

---

## Author Comment (AC2) · 30 Oct 2016

**Responses to Referee 1**

We thank the anonymous Referee 1 for taking his/her valuable time to review our manuscript and provide us some very thoughtful and constructive comments. We apologize for our first response letter, which now seems inadequate. We had to update our reply after thoroughly changing this manuscript, especially the introduction of measure theory and definitions of scale and scale transformation. Here are the new point-by-point responses to the comments from Referee 1 (please forgive us for not marking revisions in the revised manuscript; many changes were made, and marking all revisions would make the paper a mess and difficult to read).

**1.  General reply**

In the original version, we only introduced the concept of measure and developed a vector-valued measure to define "scale" (see Sect. 2 in the original manuscript). However, this new measure did not exhibit enough rigorousness. After careful consideration, we decided to abandon the original idea and completely change the definition of "scale" by further introducing Lebesgue measure for the following reasons:

(1)  Lebesgue measure is generally accepted. Its definition is sound and its geometric meaning is similar with scale. Therefore, Lebesgue measure can be potentially applied in mathematical formalism of scale (in the original manuscript, we attempted to develop a measure that belonged to Earth observations and simulations, but the result was not better than that obtained using Lebesgue measure).

(2)  Demonstrating scale transformation is important in addition to the scale. This notion coincides with the concept of change of variable in the Lebesgue integral (in the revised manuscript, this concept was called Lebesgue integration by substitution). In the original manuscript (see Sect. 2.2), we implied the concept of scale transformation with abundant discussion. However, we introduced this concept in the revised manuscript (see Sect. 3.1) following Lebesgue measure for simplicity (see Sect. 3.1 in the revised manuscript).

(3)  Based on the original idea, the explanation and instances of this new measure contained abundant content, which made our presentation redundant and hard to understand. Therefore, we thoroughly

modified this content by introducing Lebesgue measure and other associated concepts. Being similar to our study in terms of scale and scale transformation, this content could provide a more concise and rigorous presentation. Moreover, the length of this article was reduced because no extra explanation was needed.

**Changes in the manuscript:** Correspondingly, the paper was completely rewritten. The title of the revised manuscript was rewritten as "Formulation of Scale Transformation in a Stochastic Data Assimilation Framework". We made this significant modification for the following reasons. First, defining the scale and scale transformation laid a foundation for our study and makes our work distinct from the previous studies. Second, the original title was insufficient because we did not reformulate the framework of a stochastic data assimilation, which was used only to investigate the expression of errors that were determined by scale transformation. Therefore, the new title is more suitable.

Sect. 1 was reorganized and Sect. 2 was retitled as "Basic knowledge", which mainly introduced the basic concepts and theorems of measure theory and stochastic calculus. Sect. 3 was retitled as "Reformulation of scale transformation in a data assimilation framework", where we first defined some essential concepts, such as the scale, scale transformation and variables. Then, we established a Bayesian description of data assimilation with time- and scale-dependent stochastic processes and formulated the effect of scale transformations on the posterior probability of the state.

In Sect. 2.1, which was retitled "Basic knowledge of measure theory", we introduced some basic concepts such as σ-algebra, measure, measure space, Lebesgue measure, Lebesgue integral, and so on. Two main references were used: "Billingsley, P.: Probability and Measure, 2nd ed., John Wiley & Sons, New York, 1986.", and "Bartle, R. G.: The Elements of Integration and Lebesgue Measure, Wiley, New York, 1995." The latter might be the book that Referee 1 recommended in the interactive comment. Indeed, some terminological incongruences exist between these two books, so we tried our best to make the exposition acceptable and explicit.

In Sect. 3.1, which was retitled "Definition of scale", we mainly developed the structures of "scale" and "scale transformation" by Lebesgue measure. Scale is logically similar to Lebesgue measure and some technicalities were also included in the previous section, so this section is more concise than that in the original manuscript. In addition, the revised definition of scale is also valid for the following sections of our study.

Please find the detailed information in the revised manuscript.

**2.  Language problem**

*"Unfortunately, the exposition is extremely hard to follow, and, after reading the article through carefully, I still don't understand what the authors are doing. Part of the problem is probably the language. The manuscript would profit considerably if the authors could find a sympathetic native English speaker to read it over."*

**Response:** Because we are non-native English speakers, our manuscript was thoroughly edited by a manuscript service company, American Journal Experts, before submission (Certificate Verification Key: 53B9-681E-0C9B-BC77-FF4C). However, the exposition could be improved further. The revised manuscript was totally rewritten and was re-edited by a professional native English speaking team. We ensure that we will provide a modified version of this manuscript with higher-quality English expression in the future.

**3.  Reference problems**

*"More importantly, I find much of the exposition puzzling. I don't know the book by Billingsley. In my day students in the USA learned measure theory from the texts by Bartle and Royden, and, relative to my background, much of the material is written very unconventionally."*
*"The Bayesian expression of DA in terms of the stochastic calculus appears in many places. The authors should consult the volume by Jazwinski and the recent work of P. J. van Leeuwen and M. Bocquet."*

**Response:** The literature by Billingsley, Bartle and Royden are all classic works of measure theory (the texts that you mentioned may be *The Elements of Integration and Lebesgue Measure* by Bartle R. G. and *Real analysis* by Royden H. L.) and were all frequently cited according to the search results of Google Scholar. In addition, we received help from the volume "Stochastic Processes and Filtering Theory" by Jazwinski A. H. (1970) during our study, which will be listed in 'References' in the revised manuscript. We also appreciate the recommended literature regarding Bayesian data assimilation in terms of

stochastic calculus. This literature introduced the latest advances in this research field, and the related papers will be cited in our present and future works.

**4. Why measures were defined as a vector-valued set function in the manuscript**

*"The usual intuition for the concept of measure is that measure is a generalization of the concepts of length, area and volume, and is thus a scalar valued set function. The authors' response at the end of section 2.1 to Prof. Talagrand's comment is inadequate. There is nothing intrinsically wrong with choosing a vector valued measure, but that choice requires more explanation than simply "the measure correspondingly turns to ..." The authors should explain why they want to define measure as a vector valued set function, rather than simply defining the measure of a rectangle in Euclidean space as its area. Again, maybe Billingsley defines it differently, but the Lebesgue measure of a rectangle is its area, not a vector whose components are the lengths of its sides as the authors assert on line 16 of page 6."*

**Response:** Dr. Talagrand also stressed this problem. In the revised manuscript, we completely changed the definition of scale by introducing Lebesgue measure. Detailed information can be found in "General reply" in this response and in the revised manuscript (main contents are in Sect. 2.1 and Sect. 3.1).

**5. Definition of scale**

*"I do not understand the definition of scale. First, measure is a function whose domain is the sigma field F, as noted at the very beginning of section 2.1. The integrals in the definition of scale, line 16, page 7, don't make sense to me. $A_0$ is a specific set. (Pardon my TeX, I don't know how to make superscripts, subscripts or special characters). In the analysis texts I learned from, \mu ($A_0$) is the area of the set $A_0$. I don't understand the expression \mu ($A_0$)d$A_0$. I cannot make sense of the second integral. The domain of the measure \mu is the \sigma-field F, as noted in the beginning of section 2.1. The Lebesgue measure is not a point function"*

**Response:** In the revised manuscript, we completely changed the definition of scale by introducing Lebesgue measure. The scale and scale transformation were formulated by Lebesgue integral and change of variable in the Lebesgue integral. Detailed information can be found in "General reply" in this response letter and in the revised manuscript (main contents are in Sect. 2.1 and Sect. 3.1).

**6. Regarding Figure 1**

*"It would help a great deal if there were more explanation of figure 1. In particular, after reading and rereading the last paragraph on page 8, I can't understand how C_2 can have the same measure as C_1 and C_3, and D_1 has the same measure as D_2, though they have the same "scale." The problem may be the terminology: As I recall my long-ago analysis classes, the common intuition for measure was that measure corresponds to area, and, in particular, the Lebesgue measure of a geometrical figure in the plane is its area."*

**Response:** We feel very sorry because the description of this figure was not enough to help you understand. As stated in the above response, we defined scale by Lebesgue measure. Thus, the explanation of Figure 1 was changed accordingly. The conclusion is more reasonable and concise because it was established by the definition of Lebesgue measure.

**Changes in the manuscript:** The paragraph before Figure 1 was updated as follows:

"Figure 1 shows the relationship between the Lebesgue measure and scale. The measure space $\Omega = [x: 0 \leq x_k \leq 4, k = 1,2]$ is regularly divided by the unit interval $A_0$. Let $m_{C1}^2$, $m_{C2}^2$ and $m_{C3}^2$ be the Lebesgue measures of disc measurements $C_1$, $C_2$ and $C_3$, respectively, and let $m_{D1}^2$ and $m_{D2}^2$ be the Lebesgue measures of diamond measurements $D_1$ and $D_2$. Then, $m_{C1}^2 = m_{C2}^2 = m_{C3}^2$ because they are the same function. That is, if $\{A_i\}$ is the set with the smallest volume that covers $C_1$, then similar sets $\{A_i + 2\}$ and $\{A_i \times 3 + 2\}$ can be used (with the origin located in the upper-left corner) to cover $C_3$ and $C_2$ with the smallest volumes, respectively. Here, $A_i + 2 = [x: x_k + 2, x_k \in A_i, k = 1,2]$ and $A_i \times 3 + 2 = [x: x_k \times 3 + 2, x_k \in A_i, k = 1,2]$., which proves that $m_{C1}^2$, $m_{C2}^2$ and $m_{C3}^2$ collect the desirable set based on the same scheme, so they are identical. Additionally, $\sum I^2(A_i \times 3 + 2)$ is much larger than $\sum I^2(A_i)$. Therefore, the scale of $C_2$ is not equal to the two other scales because the volumes of their

subsets are different. However, their scales are governed by one-dimensional rules because their measures are identical and the Jacobian matrices between them are diagonal. Similarly, $m_{D1}^2 = m_{D2}^2$; although their scales are different, they obey a one-dimensional rule."

**7. Inconsistency problems**

*"Finally, the manuscript seems inconsistent with itself. As examples, consider the abstract. "...measure theory was used to propose [a definition of] spatial scale ...[and the] Jacobian matrix [was used] to describe the change of scale. The Jacobian matrix is introduced on page 7 in the well known change of variables formula, change of scale by the Jacobian matrix is defined on page 8, and the Jacobian is not mentioned again until the summary. No further discussion of the effects of change of scale appears. Again, in the abstract, "...the variation range of this type of error is proportional to the scale gap, ..." I'm sure I'm not the only reader for whom the phrase "scale gap" conjures up ideas of inertial range from turbulence theory and similar notions. The term "scale gap" is never mentioned in the body of the article."*

**Response:** You mentioned two inconsistency problems. The first is why the Jacobian matrix and the change in scale (the former was included in the introduction of the change of variable in the Lebesgue integral, and the latter was renamed as "scale transformation") disappeared after Sect. 2. Actually, these concepts were not omitted but simplified by the one-dimensional rule (defined in Sect. 3.1 and Eq. (5)) to suit stochastic calculus. Then, we used this simplified version of scale changes to investigate the uncertainties in data assimilation (we also mentioned this problem in the second paragraph of Sect. 5 in the original manuscript). Although the one-dimensional transformation maybe the simplest case, the results were still complicated and some new components of uncertainty were discovered (Eq. 21~27). However, fully studying the scale transformation using Jacobian matrices is comprehensive and universal, and it will be launched in our following work rather than in this study. The second inconsistency is the phrase "scale gap" in the abstract. We apologize for the trouble with this word. Here, the term "scale gap" refers to the quadratic variation between $s_X$ and $s_Y$ (see Eq. 25~27). However, fully explaining this term in the abstract was difficult, so we had to use the term "scale gap". Replacing this term with "the difference between scales" may be more practical.

---

## Author Comment (AC3) · 30 Oct 2016

**Responses to Referee 2**

We thank Prof. van Leeuwen for his valuable comments and suggestions. Most importantly, he provided some valuable literature on representation error, which convinced us to reconsider our study in light of these previous works and helped us improve our study in line with the latest advances in studying representation error. Our point-by-point responses to the comments from Prof. van Leeuwen are as follows (please forgive us for not marking revisions in the revised manuscript; many changes were made, and a revised version would make the paper a mess and difficult to read):

**1. General reply**

In the original version, we considered some results from other researchers, including basic knowledge of scale, scale problem in geophysical processes, and data assimilation with respect to stochastic processes. However, we did not consider the representation error, which was also noted by Prof. van Leeuwen in his interactive comment.

In fact, we also devoted ourselves to studying the representation error. Some results of how to understand and quantify representation error with real world experiments have been published or are being considered for publication:

- Li, X.: Characterization, controlling, and reduction of uncertainties in the modeling and observation of land-surface systems, Sci. China Ser. D, 2014.
- Huang, G., Li, X., et al.: Representativeness errors of point-scale ground-based solar radiation measurements in the validation of remote sensing products, Remote Sens. Environ., 2016.
- Li, X. and Liu, F.: Can Point Measurements of Soil Moisture Be Used to Validate a Footprint-Scale Soil Moisture Product? IEEE Geosci. Remote S., 2016. (Submitted)
- Ran, Y. H., Li, X., et al.: Spatial representativeness and uncertainty of eddy covariance carbon flux measurement for upscaling net ecosystem productivity to field scale, Agric. Forest Meteorol., 2016.

We used "representativeness errors" in our previous studies, and we believe that the representation error is not limited to data assimilation. This factor may remarkably affect many fields that are associated with Earth observations and simulations.

We did not explicitly introduce representation error in the original manuscript because this study mainly focuses on the errors from scale transformation, which is only one component of representation error. However, we accept the advice from Prof. van Leeuwen to consider representation error in our study. According to this comment, some valuable works examined representation error in data assimilation, and considering these works should substantially improve our study.

**Changes in the manuscript:** In the revised manuscript, the corresponding text was added and the structure of the manuscript was reworked. The main changes are as follows:

(1) The title of the revised manuscript was rewritten as "Formulation of Scale Transformation in a Stochastic Data Assimilation Framework". We made this significant modification for the following reasons. First, defining the scale and scale transformation laid a foundation for our study and makes our work distinct from previous studies. Second, the original title was insufficient because we did not reformulate the framework of a stochastic data assimilation, which was used only to investigate the expression of errors that were determined by scale transformation. Therefore, the new title is more suitable.

(2) We rewrote the introduction. We reduced the original text and added some necessary reviews on representation error. The fundamental causes of representation error, the difference between representation error and the scale-dependent error, the latest advances in studying the representation error in data assimilation, and the room to improve these studies were summarized.

(3) Sect. 2 was also changed. This section's name was changed to "Basic knowledge". We stated the basic knowledge of measure theory in Sect. 2.1, and we introduced the basic knowledge of stochastic calculus in Sect. 2.2. Our study was mainly presented in Sect. 3. In Sect. 3.1, we defined the scale and scale transformation with Lebesgue measure. In Sect. 3.2, we introduced the definition of stochastic variables in data assimilation. The errors from scale transformation in a data assimilation framework were presented in Sect. 3.3. Then, Sect. 4 was divided into three subsections, namely, "Summary", "Discussion" and "Next step". In the last section, we listed the basic notations in measure theory and stochastic calculus, and the new notations in our study. In addition, we removed Sect. 4.4, "An example: the stochastic radiative transfer equation (SRTE)", which was not closely tied to the other sections of this study. We shifted Sect. 4.5 to Sect. 3.4, "Extension to n-dimensional data assimilation".

(4) At the end of Sect. 3.3, we considered a related problem regarding what the error would be if the initial state is not at the scale of the forecasting operator. The results are also based on our framework with respect to scale.

(5) An extra subsection was added to Sect. 4, which was titled "Discussion". In this section, we made further comparisons between our framework and previous treatments of representation error. After a careful literature survey, we believe that the improvement from our study is a general framework of stochastic data assimilation, which was presented to investigate scale-dependent errors. First, our framework is based on nonlinear forecasting operators and observation operators. The scale transformation, which is similar to the relationship between the model space and observation space, is also nonlinear. For simplicity, we let the scale transformation obey the one-dimensional rule (See Sect. 3.1), which is equal to a linear change in scale. However, our framework does not exclude nonlinear scale transformation, and one can track a more complicated integral path of the scale for this purpose. Second, our framework permits general Gaussian representation error by introducing a non-constant volatility function in Ito-formed states and observations. Finally, we further considered the heterogeneity of geophysical parameters. This improvement highly depended on the first two advantages, i.e., either the nonlinear scale transformation or general Gaussian representation error.

Please find the detailed information in the revised manuscript.

**2. Language problem**

*"The language is also not standard."*

**Response:** Indeed, we are non-native English speakers. Therefore, our manuscript was completely re-edited by a professional native English speaking team. We tried our best to provide a high-quality English expression manuscript that could be understood by average readers in data assimilation.

We updated the data assimilation terminologies by referring to some classic literature, such as

- Talagrand, O.: Assimilation of observations, an introduction, J. Meteorol. Soc. JPN, 75, 191-209, 1997.

- Ide, K., Ghil, M., Lorenc, A. C.: Unified Notation for Data Assimilation: Operational, Sequential and Variational, J. Meteorol. Soc. JPN, 75, 181-189, 1997.

The references that were provided by Prof. van Leeuwen were used to check the language. Correspondingly, some notations were changed as follows:

| Previous notation | Current notation |
|---|---|
| Observation/Measurement region | Observation footprint |
| Model units | Model unit |
| System state | State |
| dynamic model (operator), physical model (operator) | Forecasting model (operator) |
| System state space | Model space |
| Instrument error | Measurement error |

Additionally, some notations from measure theory and stochastic calculus, as well as the new notations that were defined in our study may seem strange to ordinary readers in data assimilation. These notations were all listed and provided corresponding explanations in Sect. 5, "Notation", in the revised manuscript.

**3. Innovation of this study**

*"There are several recent works that describe the filtering problem for stochastic processes, and I'm not sure the results of the authors are new in this respect."*

**Response:** Indeed, abundant literature exists on data assimilation based on stochastic processes. However, our study is very different from the available works and deduced some remarkable results. The previous works were mostly based on stochastic processes with respect to time. Our study, which introduced measure theory to investigate the natural structure of scale and treated scale variations similarly to time variations, constructed a framework that was based on stochastic calculus with respect to scale. Some components of representation error are associated with scale, so our study provided a more reasonable framework that can further improve our understanding of the structure of representation error in data assimilation.

**4.    Necessity of introducing measure theory**

*"I don't see the need for the measure theory developed here."*

**Response**: The scale in representation error is important, but a rigorous mathematical definition of scale is lacking. Simply regarding the resolution as the scale is not reasonable. For example, if two different observation footprints exist, one is a disc field with a diameter of 1 km and the other is a square field with side length of 1 km. Obviously, they have the same resolutions, but their scale is different. We introduced measure theory to further mathematically describe the geometric characteristics of an observation footprint or model unit, as well as the scale transformation. Measure theory laid the foundation of this study in terms of scale-dependent error in data assimilation.

**5.    Advantages of our study**

*"My suggestion is that the authors consider these references and reconsider their findings in light of those works. What has been missed by the examples of papers referred above? Why is the new framework needed, using language that the average NPG reader interested in data assimilation can understand?"*

**Response**: The advantages of our study include the general framework of stochastic data assimilation to investigate the scale-dependent errors, i.e., a framework that includes the nonlinear operators and nonlinear scale transformation (such as the relationship from model space to observation space), the general Gaussian representation error, and considering the heterogeneity of geophysical parameters. Although some of these factors were simplified to linear situations for more clear results, nonlinear results could also be deduced from this framework.

Please find the detailed information in the general reply of this response letter and the revised manuscript (main contents in Sect. 1 and Sect. 4.2).

---

## Author Comment (AC4) · 30 Oct 2016

**Responses to the editor's comments**

We appreciate the kind considerations from Dr. Olivier Talagrand on our manuscript in NPG Discussion. We made major revisions to our original manuscript based on all the comments from the editor and referees. The main amendments are as follows (please forgive us for not marking revisions in the revised manuscript; many changes were made, and marking all revisions would make the paper a mess and difficult to read):

**1. Structural changes**

We thoroughly revised the structure to make our manuscript more understandable. The title of the revised manuscript was rewritten as "Formulation of Scale Transformation in a Stochastic Data Assimilation Framework". We made this significant modification for the following reasons. First, defining the scale and scale transformation laid a foundation for our study and makes our work distinct from the previous studies. Second, the original title was insufficient because we did not reformulate the framework of a stochastic data assimilation, which was used only to investigate the expression of errors that are determined by scale transformation. Therefore, the new title is more suitable.

First, we rewrote Sect. 1. We reduced the original text and added some necessary reviews on representativeness error. Second, Sect. 2 was also changed to "Basic knowledge"; we stated the basic knowledge of measure theory in Sect. 2.1, and the basic knowledge of stochastic calculus in Sect. 2.2. Our study was mainly presented in Sect. 3. In Sect. 3.1, we defined the scale and scale transformation with Lebesgue measure. In Sect. 3.2, we introduced the definition of stochastic variables in data assimilation. The errors from scale transformation in a data assimilation framework were presented in Sect. 3.3. Then, Sect. 4 was divided into three subsections, namely, "Summary", "Discussion" and "Next Step". In the last section, we listed the basic notations in measure theory and stochastic calculus and the new notations in our study.

In addition, we removed Sect. 4.4, "An example: the stochastic radiative transfer equation (SRTE)", which was not closely tied to the other sections of this study. We shifted Sect. 4.5 to Sect. 3.4, "Extension to n-dimensional data assimilation".

**2.  Full introduction of Lebesgue measure to define "scale"**

In the original version, we developed a vector-valued measure to define the "scale" (see Sect. 2 in the original manuscript). Referee 1 argued that this measure mismatches with his/her basic knowledge of measure theory and leads to a poor understanding of the manuscript. Dr. Talagrand also stressed this problem. We accept this advice. After careful consideration, we decided to abandon the original idea and completely change the definition of the "scale" by further introducing Lebesgue measure.

**Changes in the manuscript:** Correspondingly, Sect. 2 was completely rewritten. In Sect. 2.1, which was retitled "Measure theory", we introduced some basic concepts such as σ-algebra, measure, measure space, Lebesgue measure, Lebesgue integral, and so on. In Sect. 3.1, "Definition of scale", we used Lebesgue measures to investigate the structures of scales and scale transformation.

**3.  Previous studies on representativeness error were included and compared to our study**

Prof. van Leeuwen noted that no reviews of representativeness error exist and that this topic requires a comparison between our study and previous work on representativeness error. We added the corresponding content in the revised manuscript.

**Changes in the manuscript:**

(1)   The introduction mainly considered previous work on representation error.

(2)   Sect. 4.2 was added to compare our study with the available works on representation error.

Please find this information in the revised manuscript (main contents are in Sect. 1 and Sect. 4.2).

**4.  Equation recompilation**

We reedited all the equations with Microsoft equation editor. The typesetting of the mathematical equations in the revised manuscript is better than that in the original manuscript.

**5.  Notation consistency**

Both the referees and Dr. Talagrand emphasized that the notations in the original manuscript were inconsistent. We seriously addressed this issue. The corresponding changes in the new version are as follows:

| Previous notation | Current notation |
| --- | --- |
| Scale change, change of scale | Scale transformation |

| | |
|---|---|
| Observation/measurement region | Observation footprint |
| Model units | Model unit |
| System state | State |
| dynamic model (operator), physical model (operator) | Forecasting model (operator) |
| System state space | Model space |
| Instrument error | Measurement error |

**6. New Notations**

Dr. Talagrand advised us to clearly explain the new notations that were introduced in our manuscript. We accept this advice. In the revised manuscript, we expanded the "Notation" section and classified all the notations into two types. One was the basic notations in measure theory and stochastic calculus, which were given only a full name, and their definitions can be found in Sect. 2.1 and Sect. 2.2. The other type was the new notations. We offered their full names and their detailed explanations and indices in the manuscript. Please find more information in the revised manuscript (Sect. 5).

**7. Other changes**

Many other small presentation changes were made in the current manuscript. Some of these changes involved our re-examination, and some were provided by a professional native English speaking team. We hope that these changes improved the manuscript substantially to meet the quality standards of NPG. However, the changes were not presented as revisions in our revised manuscript because so many revisions that presenting them all would make the paper a mess.

---

## Referee Report (RR1)

The authors have clearly made a conscientious effort to address my original concerns, and they have, for the most part succeeded. While the new version is much improved, I still can't say that I understand it completely. I think that, at this point, it's just a matter of wording, and the authors should be able to correct what seem to me to be obvious problems and produce an entirely acceptable manuscript fairly easily.

First, the authors should be careful to treat the Lebesgue measure rigorously correctly. A few examples:

1. On page 5: "Generally a Lebesgue measure on $\mathbb{R}^n$ assumes that $A$ is any subset of $\mathbb{R}^n$." This is not true. The $\sigma$-algebra $\mathcal{F}$ in the definition of the triplet $(\Omega, \mathcal{F}, \mu)$ (line 6 on p5) that defines the measure does not include all subsets of $\mathbb{R}^n$. There are subsets of $\mathbb{R}^n$ to which a Lebesgue measure cannot be consistently assigned. Construction of these so-called "unmeasurable sets" is described in the standard texts, as the authors know.

2. Same paragraph: instead of "Thus if $A$ is any subset of $\mathbb{R}^n$, one can collect ..." I suggest "for any $A \in \mathcal{F}$ one can collect ..."

3. p5, toward the bottom: "The Lebesgue measure of any subset in $\mathbb{R}^n$ also coincides with its volume." Again, there are subsets of $\mathbb{R}^n$ to which a Lebesgue measure cannot be consistently assigned.

4. p8: "...any bounded closed domain $A$" As before, $A$ must be Lebesgue measurable.

5. p11 lines 7-8 instead of "$A \in \mathbb{R}^n$", you want "$A \in \mathcal{L}^2$"

The definition of scale in section 3 is much better than the original, but it is still not clear. At this point some careful attention to the exposition should be sufficient. Here is the problem:

p8, lines 4-5: "We use Lebesgue measure on $\mathbb{R}^2$, i.e., $\mu_{iv}(A) = m^2(A) = inf\left(\sum_{i=1}^{+\infty} I^2(A_i)\right)$ where ... From a geometric perspective, the measure function refers to the shape of the subset, and the scale further indicates the size." OK, I understand that, say, in figure 1, you mean to say that disks $C_1$, $C_2$ and $C_3$ have the same shape, but they have different scales because they are different sizes. But the Lebesgue measure is the area, and you have defined it as it is defined in the books. By your definition of $m^2$, referring to figure 1, $m^2(C_2) > m^2(C_1)$, but on line 10 you write "$m_{C1}^2 = m_{C_2}^2 = m_{C_3}^2$ because they are the same function." I don't understand this at all. You mention a function you call $f$ but it plays no part in the definition of $m^2$. The statement "$m_{C1}^2 = m_{C_2}^2 = m_{C_3}^2$" is inconsistent with the stated definition of $m^2$.

I'm guessing that $C_1$, $C_2$ and $C_3$ are examples of footprints. If so, would you please say this explicitly?

What, exactly, are the functions associated with $C_1$, $C_2$ and $C_3$?

---

## Editor Decision (ED1)

Dear Drs Feng Liu and Xin Li,

I have received two reviews of the new version of your paper. The referees are the same as those of the previous version. In particular, referee 2, who has again let his name known, is Prof. P. J. van Leeuwen.

Both referees consider that your paper has been improved, and that the mathematical derivations are now clear. At the same time, both write that they still do not fully understand the logic and the significance of your work. Referee 1 writes *While the new version is much improved*, *I still can't say that I understand it completely* and adds *I think that, at this point, it's just a matter of wording*. Referee 2 is in a sense more critical and writes *I don't understand the stochastic equation in scale space, neither where it comes form nor how it helps solve the representation error problem*. Both of them ask for a major revision.

Referee 1's most important comments bear on the use you make of the Lebesgue measure. He/she mentions that the equality on 1. 10, p. 9, is in contradiction with the definition of the scale you give on ll. 11-12 of p. 8. This may be only a question of notation or of misunderstanding from the part of the referee, but it has to be clarified.

Referee 2's comments are at a more fundamental level. The most important ones are those relative to pp. 11-13 of the paper. They develop his general comment above on the significance of the stochastic equation in scale space.

Thinking of the general significance of your paper, I actually think as Editor that it would be useful to add a simple illustrative example, which would show explicitly how considering a stochastic scale transformation can impact the assimilation and the probability distribution that it produces. You have removed the section dealing with the Radiative Transfer Equation (RTE) which was included in the previous version of your paper, considering it was not closely tied to the other sections of the paper. I do not think it was explicit enough to show the impact on assimilation of a stochastic scale transformation, but it could possibly be used, with appropriate modifications, for that purpose. Without making it a condition for acceptance of the paper, I think an appropriate illustrative example, whether based on the RTE or not, would make the paper more understandable.

In addition to their basic comments, both referees also make a number of suggestions for editing (the first five comments of referee 1, and all comments of referee 2 down to p. 11, plus a few others). I myself add below of number of suggestions for corrections.

1. P. 5, l. 14, and p. 8, l.5. I suggest to add indices *i* as follows

 $A_i = [x: a_{k,i} \le x_k \le b_{k,i}, k = ...]$

2. P. 6, l. 17. I presume you mean

 $\forall t_1 > s_1 \ge t_2 > s_2$ , the increments  $W(t_1) - W(s_1)$  and  $W(t_2) - W(s_2)$  are independent.

- 3. P. 8, l. 15, ... a unit interval  $\dots \rightarrow \dots$  the unit square  $\dots$
- 4. Probabilities are denoted *p*() in some places (eq. 8, p. 10 for instance), and *P*()in other places (p. 11, ll. 2-4). Please use consistent notations.

Please revise your paper according to the comments and suggestions of the referees, as well as to my own ones. And please give a point-to-point response to all of these comments and suggestions. Should you disagree with one particular comment, or decide not to follow a particular suggestion, please state precisely your reasons for that. The revised manuscript will be submitted to two referees, who may, or may not, be those of your former versions. Submission of a revised manuscript does not necessarily ensure publication in *NPG*.

I thank you again for having submitted your paper to *Nonlinear Processes in Geophysics*, and look forward to receiving a new version.

With regards,

Olivier Talagrand Editor Nonlinear Processes in Geophysics

---

## Author Response (AR2)

**Authors' Responses to the Comments from Editor and Referees**

**General reply**

Based on the comments from editor and referees, we carefully reconsidered our manuscript. There are too many revisions such that the structure of the new manuscript is mess. Therefore both the tracked and the clean versions were provided to make reading as easy as possible. All the positions we mention in this response letter are according to the pages and lines of the clean version.

The revisions mainly consist of six parts:

1. Structure changes. First, we added a new Sect. 2.3 titled "Traditional formulation of data assimilation in the Bayesian theorem framework" in the revised manuscript and the corresponding text in Sect. 3.2 was removed, because this material is also a basic knowledge of our study. Second, according to editor's suggestion, we added examples to further explain the theoretical results. All the examples were organized in the new Sect. 3.4 titled "Examples: the stochastic radiative transfer equation (SRTE)".

2. Rewriting Sect.1 and Sect. 4. We rewritten the introduction to make the text more concise and comprehensible. The advances and scientific problem in our study were clearly presented. The title of Sect.4 was change to "Discussion & Conclusions", and accordingly there are only two subsections left. In the revised manuscript, Sect.4.1 titled "Discussion" mainly focused on the necessity of methodology, the advantages and limitations of our study, and Sect.4.2 titled "Conclusions" restated the major results.

3. Revisions according to the comments from editor and referees. Detail information can be found in the point-by-point responses to editor and referees.

4. The formulation of scale transformation between the scales of initial state and forecasting operator was removed (In the end of Sect. 3.3). This formulation was presented in the previous manuscript. After lots of reconsiderations and discussion, we believed that this formulation is not necessary to be introduce in this paper because the main scientific problem is to formulate the error in the

update step, and the error caused by the scale difference between initial state and forecasting operator will not have a significant impact on the formulation of the update step.

5.  To make one term in this paper only presents a single object, some similar terms in different fields were again clarified. Please check them according to the following form.

| | Terms in this paper | Explanation |
|---|---|---|
| *1* | Measure | Measure is a term for measure theory, and Observation is an estimation of the value for a geophysical variable. |
| | Observation/Observe | "Measurement" have the similar meaning with "Observation' but may be confused with "Measure", so "Measurement" will not be introduced. Also, "measurement error" will be replaced by "instrument error". |
| *2* | Footprint | A footprint is the observation footprint, and space is the measure space or state/observation space that a geophysical variable can evolve. We try to avoid the use of "field" or "region" to indicate the similar meaning. |
| | Space | |
| *3* | Variable | Variable is the geophysical variable or variable in state vector that can be observed by Earth observations. Parameter cannot be observed. In the mathematical formula, parameter also refer to the argument of a function. Random process indicates the stochastic process or Ito process only on the condition that a rigorous mathematical expression is involved. |
| | Parameter | |
| | Random process | |

6.  Other modifications that are based on the updated cognition on our study can also be found in the revised manuscript.

**Responses to the Editor's Comments**

We thank Dr. Olivier Talagrand once again for taking his valuable time to review our revised manuscript and given recognition of the improvement in our study. Based on all the comments from editor and referees, we have made major revisions of our manuscript. Here are the point-by-point responses to the new comments from editor.

1. *"Thinking of the general significance of your paper, I actually think as Editor that it would be useful to add a simple illustrative example, which would show explicitly how considering a stochastic scale transformation can impact the assimilation and the probability distribution that it produces. You have removed the section dealing with the Radiative Transfer Equation (RTE) which was included in the previous version of your paper, considering it was not closely tied to the other sections of the paper. I do not think it was explicit enough to show the impact on assimilation of a stochastic scale transformation, but it could possibly be used, with appropriate modifications, for that purpose. Without making it a condition for acceptance of the paper, I think an appropriate illustrative example, whether based on the RTE or not, would make the paper more understandable."*

**Response:**

We agree with you. An example can make our study more understandable. With some modification, we first added an example to explain how the scales of system state and observation can obey the one-dimensional rule based on the scales presented in Figure 1. Then the formula of likelihood function was deduced to two different cases. The first one is that the observation is the same physical quantity as the state. In the second case, a nonlinear observation operator, i.e., a stochastic radiative transfer equation (SRTE) is used as another example. One thing should be noticed in the second case is that we only offered the Ito process-formed state, observation and observation operator. These functions can be used to further deduce the likelihood function according to Eq. (22).

**Changes in the manuscript:**

The example with Stochastic Radiative Transfer Equation in the previous manuscript is a little complicated. So we reduced its expression. Besides, using the scales presented in Figure 1, we also added some more simple examples. Both the simple examples and the example based on SRTE are introduced in Sect. 3.4 titled "Examples: the stochastic radiative transfer equation (SRTE)".

*2. P. 5, l. 14, and p. 8, l.5. I suggest to add indices i as follows*

**Response:**

Yes, this makes the formula more clearly. We also add the indices *i* in the last paragraph of Sect. 3.1.

*3. P. 6, l. 17. I presume you mean $\forall t1 > s1 \geq t2 > s2$, the increments $W(t1) - W(s1)$ and $W(t2) - W(s2)$*

*are independent.*

**Response:**

Your suggestion makes the definition easy to understand, so we accept it and revised it in the new

version.

*4. P. 8, l. 15, ... a unit interval ...  →  ... the unit square ...*

**Response:**

Thanks for your suggestion. We also revised the other 3 words with the similar problem.

*5. Probabilities are denoted p()  in some places  (eq. 8, p. 10 for instance), and P()in other places (p.*

*11, ll. 2-4). Please use consistent notations.*

**Response:**

Thanks for your comment. We have made the revision accordingly.

**Responses to Referee 1's Comments**

We thank the anonymous Referee once again for taking his/her valuable time to review our revised manuscript and given recognition of the improvement in our study. Here are the point-by-point responses to the new suggestions.

**1. Lebesgue measurable subset**

*1. On page 5: "Generally a Lebesgue measure on $R^n$ assumes that A is any subset of $R^n$." This is not true. The σ-algebra $\mathcal{F}$ in the definition of the triplet $(\Omega, \mathcal{F}, \mu)$ (line 6 on p5) that defines the measure does not include all subsets of $R^n$. There are subsets of $R^n$ to which a Lebesgue measure cannot be consistently assigned. Construction of these so-called "unmeasurable sets" is described in the standard texts, as the authors know.*

*2. Same paragraph: instead of "Thus if A is any subset of $R^n$, one can collect..." I suggest "for any $A \in \mathcal{F}$, one can collect ..."*

*3. p5, toward the bottom: "The Lebesgue measure of any subset in $R^n$ also coincides with its volume." Again, there are subsets of $R^n$ to which a Lebesgue measure cannot be consistently assigned.*

*4. p8: "...any bounded closed domain A" As before, A must be Lebesgue measurable.*

*5. p11 lines 7-8 instead of "$A \in R^n$", you want "$A \in \mathcal{L}^2$"*

**Response:**

Comment 1: Yes, you are right. The expression in here is not rigorous. Here A should be a Lebesgue measurable subset of $R^n$.

Comment 2, 3 and 5: Yes, these sentences will be more rigorous based on your suggestions. We have made revisions accordingly.

Comment 4: Based on my knowledge, a bounded closed domain is Lebesgue measurable. So, I think this sentence might be right. Besides, in this sentence, *A* indicates the observed space of a footprint-scale observation, which are Lebesgue measurable.

**Changes in the manuscript:**

Comment 1. This sentence was removed. To make this statement more clear, in the next paragraph, we added some sentences behind the Lebesgue σ-algebra: "The construction of the Lebesgue σ-algebra

is based on the Caratheodory condition (Bartle, 1995, definition 13.3). Fortunately, almost all of the observation footprints and model units are finite and closed, leading them to be Lebesgue measurable. This consequently ensures the Lebesgue measure $m^n$ is a measure and the triple $(R^n, \mathcal{L}^n, m^n)$ is a measure spaces."

Comment 2. In this paragraph, if the first sentence was removed, then we only talk about the outer measure. I think the definition is valid for the outer measure of an arbitrary subset of $R^n$, so maybe no change is needed in here.

Comment 3. This sentence was changed to "The Lebesgue measure of a Lebesgue measurable subset in $R^n$ also coincides with its volume".

Comment 5: *"$A \in R^2$" was changed to "$A \in \mathcal{L}^2$"*

**2. The definition of scale**

*p8, lines 4-5: "We use Lebesgue measure on $R^2$, i. e., $\mu_{iv}(A) = m^2(A) = \inf\left\{\sum_{i=1}^{+\infty} I^2(A_i)\right\}$ where ...*

*From a geometric perspective, the measure function refers to the shape of the subset, and the scale further indicates the size." OK, I understand that, say, in Figure 1, you mean to say that disks $C_1$, $C_2$ and $C_3$ have the same shape, but they have different scales because they are different sizes. But the Lebesgue measure is the area, and you have defined it as it is defined in the books. By your definition of $m^2$, referring to Figure 1, $m^2(C_2) > m^2(C_1)$, but on line 10 you write "$m^2_{C1} = m^2_{C2} = m^2_{C3}$ because they are the same function." I don't understand this at all. You mention a function you call f but it plays no part in the definition of $m^2$. The statement "$m^2_{C1} = m^2_{C2} = m^2_{C3}$" is inconsistent with the stated definition of $m^2$.*

*I'm guessing that $C_1$, $C_2$ and $C_3$ are examples of footprints. If so, would you please say this explicitly? What, exactly, are the functions associated with $C_1$, $C_2$ and $C_3$?*

**Response:**

I am so sorry that I had not make it much clearer. Your understanding is correct. We defined the scale as the Lebesgue measure with respect to the observation footprint, i. e., $s = m^2(A)$. Therefore, there are two elements in this definition, the Lebesgue measure function $m^2(\cdot)$ and the observation footprint $A$. The two scales, $s_1$ and $s_2$, are equal to each other happens only when they have the same Lebesgue measure functions and the same observation footprints, say, $m^2_{s_1}(\cdot) = m^2_{s_2}(\cdot)$ and $A_{s_1} = A_{s_2}$. In Figure

1, $m_{C1}^2$, $m_{C2}^2$ and $m_{C3}^2$ have the same Lebesgue measure functions, that is, they all refer to a disc footprint, so $m_{C1}^2(\cdot) = m_{C2}^2(\cdot) = m_{C3}^2(\cdot)$. Your challenge is reasonable, here we made a mistake that we left out the note $(\cdot)$ in this formula, which misled the readers to believe that the Lebesgue measures of disc measurements $C_1$, $C_2$ and $C_3$ are equal in value. Here the Lebesgue measure functions $m_{C1}^2(\cdot), m_{C2}^2(\cdot)$ and $m_{C3}^2(\cdot)$ are associated with C$_1$, C$_2$ and C$_3$.

We introduced the real function $f$ because it is involved in the general definitions of Lebesgue integral and Lebesgue integration by substitution. However, $f$ is equal to 1 in the definitions of scale and scale transformation because the Lebesgue measure in $R^2$ is area. At p8, lines 9 and line 12, we have stated that $f \equiv 1$.

C$_1$, C$_2$ and C$_3$ are indeed the examples of footprints, we have clarified this in the revised manuscript.

**Changes in the manuscript:**

The formula $m_{C1}^2 = m_{C2}^2 = m_{C3}^2$ was changed to $m_{C1}^2(\cdot) = m_{C2}^2(\cdot) = m_{C3}^2(\cdot)$. In addition, we revised Fig.1 and removed the diamond observation footprint $D1$ and $D2$ to make Fig.1 more concise. We also added some necessary words in this paragraph to make the explanation of Fig. 1 clearer. Please find the detailed information in the new manuscript.

**Responses to Prof. van Leeuwen's Comments**

We thank Prof. van Leeuwen once again for taking his valuable time to review our revised manuscript and provide us some very thoughtful and constructive comments. The point-by-point responses to the new suggestions were classified by what these comments focus on.

**1. Scale**

*1. Abstract: Define the spatial scale issue, Do you mean that models and observations define scale differently? Or do you mean that scale is not well defined in general, which then hampers model-observation comparisons and data assimilation?*

*2. P8, 11: scale is defined as an area, so has physical dimension m^2. Typically scale is defined in terms of distance. That might be mentioned.*

*3. P8, 14: The standard scale depends on the units used, it is a different thing using meters of millimetres. Is this a useful definition? Or do you assume that all physical scales have been normalised? Please clarify.*

**Response:**

The understanding of scale issue and the definitions of scale and scale transformation play an important role in our work. If the observation footprints or model units are changed and associated variables present heterogeneities, the scale issue is inevitable. However, scale is not well understood currently. Defining scale in terms of distance is not adequate because distance is a one-dimensional quantity but scale generally refers to a two- or three-dimensional space. We believe the scale is related to the shape and size of observation footprint or model unit, so the Lebesgue measure on $R^2$ was used to define it.

Another reason for defining the scale is values for variables may change with scale in most of Earth observations and simulations. Scale may change much complicated (for example, form an irregular observation footprint to a square observation footprint), so how to quantify this change must be based on a rigorous definitions of scale and scale transformation.

Therefore, for *comment 1,* we think that scale is not well defined in general. Meanwhile, in the studies of model-observation comparisons or adding a new observation into data assimilation system, the

transformations between different scales result in remarkable error (also can be seen as a part of representativeness error). However, scale transformation was also not fully addressed.

For *comment 2*, as we have mentioned, distance is a one-dimensional quantity and unable to meet the definition of scale.

For *comment 3*, similarly, to use meters of millimetres is more reasonable for one-dimensional scale. But the scales that related to Earth observations, simulations and data assimilation should be regarded as two-dimensional or three-dimensional. We introduced unit area $A_0$. The standard scale, defined as the area of $A_0$, is significant because, on the one hand, it is a standard reference, by which one can make a quantitative comparison between different scales; on the other hand, the standard scale can be seen as a origin if we treat scale similarly to other physical quantities, such as time. Consequently we can develop the Brownian motion and stochastic calculus based on scale. Brownian motion and stochastic calculus both begin with the standard scale (for example, Lemma 1 and Eq. 12).

We introduced the standard scale, but that is not necessary to require all physical scales to be normalized by standard scale. However, if it is in need of a rigorous formulation of scale-dependent error, it's better to look to standard scale.

**Changes in the manuscript:**

We revised the manuscript accordingly as follows,

For comment 2, In paragraph 1, Sect. 1, we added some text after the first sentence: "Scale is traditionally defined in terms of distance, which is not adequate both because distance is a one-dimensional quantity but scale generally refers to a two- or three-dimensional space, and because scale may change much complicated (for example, form an irregular observation footprint to a square observation footprint).".

For comment 3, detail explanation was appended after the definition of the standard scale: "The standard scale can be regarded as a basic unit of scale in two-dimensions. It presents a standard reference, by which one can make a quantitative comparison between different scales. The standard scale is also the origin of scales that let scales vary similarly to other physical quantities, such as time." Additionally, after the paragraph on "Remark on Lemma 1", we added "Therefore, beginning with the standard scale, the Brownian motion and stochastic calculus with respect to scale can be further developed."

**2. Language improvements**

*1. P1, 25: 'increases with the difference'*

*2. P4, 16: 'the definition of scale uses measure theory'*

*3. P6, 19: typo in first equation: 'the first s1' should be 's2'.*

*4. P10, 16: 'discovered' should perhaps be 'described the origin of'*

*5. P13, 13 Typo in equation (15): X(s_X) = X_0 + ...*

**Response:**

Thanks for your comments on language improvements. We have made revisions accordingly except comment 1, which was replaced by the new introduction.

**3. Variables and parameters**

*1. P1, 28: parameters are chosen constants in time, variables are varying in time. So I guess the authors mean 'geophysical variables', or perhaps both. This runs through the whole manuscript.*

*2. P2, 14: Parameters cannot be collected. Values for variables can be collected. And they are not collected by Earth Observation techniques but by Earth observations.*

*3. P11, 7: the definition of 'variable' is unconventional and perhaps misleading. An element of state vector X is typically called a variable in the data-assimilation literature. Another name would be preferable.*

*4. P11, 7: What is the exact relation between state vector X and variable V?*

**Response:**

Thanks for these comments. It is really helpful that variable and parameter are different and we indeed mixed them up. As you stated, the term "geophysical parameter" is usually regarded as a spatial and temporal constant, which cannot be observed. But variable changes with space and time, and its value can be observed.

For *comment 3*, the term "variable" is a common concept both in geophysics and mathematics. In measure theory, it indicates a real-value function on a probability space $(\Omega, \mathcal{F}, P)$, but in data assimilation "variable" means a little differently. Here we in fact defined a general geophysical variable,

therefore, according to your advice, in the new manuscript, we use geophysical variable instead of "variable".

For *comment 4*, in our study, the $V$ is the stochastic version of an element of state vector $X$. We try to further introduce the mathematical definition of geophysical variable in the sense of measure theory (see page 10, line 6 and 7), and then study the Ito process-formed geophysical variable (see Eq. (9) and Eq. (10)).

**Changes in the manuscript:**

For *comment 1 and 2*, the term *"geophysical parameters"* was revised as *"geophysical variables"*, and "parameter" in the new manuscript only refers to "argument of a function". How to distinguish the other similar terms was also presented in part 5, General reply of this response letter.

For *comment 3 and 4, V* was revised as *"geophysical variable"*. After the definition of geophysical variable, we also appended "In n-dimensional data assimilation, a geophysical variable $V$ is related to an element of state vector $X$ at a specific scale $s$ and time $t$". Other related text will be revised in the new manuscript as well.

**4.  Stochastic differential equation with respect to scale**

*1. P11, 23: The authors introduce a stochastic process in scale space that operates at a time instance, so time is a constant, and the variable changes due to a process in scale space. Is this interpretation correct, and if so, what is this process physically? This is a crucial point for me, and the point where I get lost.*

*2. P12: I understand this as a formal derivation of the stochastic process in scale space up to line 23 (but, as mentioned, I don't understand the physical process behind this).*

*3. P13, 4: What does this equation mean? That X(s) changes due to changes in scale in the analysis step? If so, phi should also depend on Y, or at least on the scales in Y. Or does this equation describe a scale relation in X? So how X depends on scale? If so, what is the stochastic forcing?*
*Also, how to choose or estimate phi and sigma in an application?*

**Response:**

Yes, this is the most important problem in our study. In paragraph 2, Sect. 1, we stated that the scale issue is related to spatial heterogeneities and dynamic process variations among different scales. That's to say, if the study region is not homogeneous, the values of a variable that observed

at the same place may present differently between large scale and small scale (for example, between the larger footprint *C2* and the smaller footprint *C3* in Figure 1). Some physical processes also vary among different scales. For example, except the ones we mentioned in paragraph 2, Sect. 1, ground water flow process is governed by Darcy's law at the macro-scale and by the Navier–Stokes equations at the pore-scale (Narsilio, et al. 2009). The validity of Planck's law also depends on the scale (Li, et al. 1999).

Therefore, to understand the physical processes behind the scale issue should both consider the heterogeneities and the changes of dynamic processes among different scales. However, based on associated literatures, most of them are not very clear, let along to model these physical processes in a general theory study. Therefore, a sophisticated formulations is conceptualized in our manuscript but we believe this problem needs further study to make it more concrete.

For *comment 3,* we think that in the analysis step, time is invariant, but the state $X$ in the state space is mapped to the observational space, i.e., the scale of $X$ changes from $s_X$ (scale of state space) to $s_Y$ (scale of observation space). This process can be regarded as an Ito process of state with respect to scale, which can also be formulated by Eq. (15). Based on Eq. (12) (the integral form of Ito process), state $X$ in the state space is $X(s_X) = X_0 + \int_{s_0}^{s_X} \varphi(u)du + \int_{s_0}^{s_X} \sigma(u)dW(u)$ and state in the observational space is $X(s_Y) = X_0 + \int_{s_0}^{s_Y} \varphi(u)du + \int_{s_0}^{s_Y} \sigma(u)dW(u)$. Apparently, $X$ depends on scale.

In Eq. (13), as we stated, $\varphi(s)$ is the scale-dependent drift rate from standard scale to a specific scale, for example, $s_X$ or $s_Y$. $\varphi(s)$ accords to the physical processes of state with respect to scale, which currently may be hard to be formulated. $\sigma(s)$ can be regarded as the stochastic perturbation with respect to scale, which is needed to be further investigated. However, if assuming the perturbation at the scales is totally random or Gaussian, then $\sigma(u) = 1$. In our study, only the simplest case, i.e., $\varphi = 0$ and $\sigma = 1$, was considered. This means only the Gaussian perturbation presents when the scale is changed. However, the result (Eq. (23)) is still complicated.

**Changes in the manuscript:**

For *comment 1* and *comment 2,* we added some necessary explanations after Eq. (10): "To formulate $\varphi(s)$ should consider both the spatial heterogeneities and physical process variations among different scales. However, neither of them is well understood in a general theory study.

Therefore $\varphi(s)$ is conceptualized in Eq. (10)." And after Eq. (13) and Eq. (14), we stated "$\varphi(s)$ also implies the heterogeneities and physical processes from standard scale to a specific scale, which currently maybe hard to be formulate."

For *comment 3*, we added text after Eq. (14): "Therefore, according to Eq. (12), a state is $X(s_X) = X_0 + \int_{s_0}^{s_X} \varphi(u)du + \int_{s_0}^{s_X} \sigma(u)dW(u)$ in the state space and is $X(s_Y) = X_0 + \int_{s_0}^{s_Y} \varphi(u)du + \int_{s_0}^{s_Y} \sigma(u)dW(u)$ in the observation space. These formulas prove that the value of state varies with the changes of scale."

**5. Data assimilation**

*1. P12: I don't understand what assumption 1 means, 'in data assimilation' seems rather vague to me. Assumption 2 assumes that the model has a constant grid in space and time. That should perhaps me mentioned explicitly. So the grid is not finite element or finite volume, or adaptive in time. Assumption 3: What does 'scale dependent' mean here? That the scale changes when applying Bayes Theorem?*

*2. P13, 9: The `authors state that 'Assumption 3 implies that the scales of the state and observation are invariant when observational information is added in the analysis step'. I don't see that, please clarify.*

**Response:**

For assumption 1, the scale transformation between state space and observation space of data assimilation obeys a one-dimensional rule. The one-dimensional rule is defined in Sect. 3.1 and can make scale change in a sense of geometrical similarity (for example, form a square observation footprint to a smaller square observation footprint). By this assumption, the formulations of scale transformation in data assimilation can be extremely reduced, but turn out the same conclusions with the one without any assumption about scale transformation.

For assumption 2, in the forecasting step, the model unit and state scale are both supposed to be same and invariant. There is no scale transformation in this step.

For assumption 3, the term "scale dependent" means that the state, observation and observation operator are all dependent on scale, and they can vary with scale. But when the Bayesian theorem is applied, the scales of state and observation are actually not changed, and the scale transformation

only involves in the process that mapping the state vector from state space to observation space. Then we get an estimation of observation $H(X(s_X))$ in the observation space which is related to the state $X(s_X)$ defined in the state space.

**Changes in the manuscript:**

We add some necessary text after all the assumptions, which is supposed to be more explicit to express our intention.

For assumption 1, we added "In assumption 1, the one-dimensional rule ensures that scale changes in a sense of geometrical similarity (for example, form a larger square observation footprint to a smaller square observation footprint, or from $C_2$ to $C_3$ as presented in Figure 1). Additionally, the formulations of scale transformation can be extremely reduced, but turn out the same conclusions with the one without any assumption about scale transformation."

For assumption 2, it stated that "Assumption 2 indicates that the model unit and state scale are both supposed to be the same and invariant in space and time. So, there is no scale transformation in the forecasting step".

For assumption 3, after Eq. (14), the sentence "Assumption 3 implies that the scales of the state and observation are invariant when observational information is added in the analysis step" missed some necessary information. This sentence was replaced by "The scale transformation only involves in the process that mapping the state vector from state space to observation space".

*3. P13, 22: What does this equation mean? Is this the prior marginal pdf of X in scale space?*

*I am lost as this page. Where is the measurement uncertainty? That should also appear somewhere in p(y|x). I see only the scale part of the error. Is the assumption that the scale part is dominant? And again, what is this process in scale space that happens at observation time prior to calculating p(y|x). Note that assumption 2 mentions explicitly that the scale of state and observation do not change before the observation time.*

**Response:**

The equations mean the prior pdfs of state and observation with respect to scale in state space and observation space, respectively. They are different from the pdfs with respect to time, because their means are equal to the value at the standard scale and variances depends on the differences

between standard scale and the state space or observation space. These two prior pdfs are introduced into the Bayesian theorem that reformulated by scale.

We first clarify that the measurement error has little impact on the error caused by scale transformation (it will be also clarified in the new manuscript, see paragraph 3, Sect. 1). Actually, in some data assimilation literatures, for example, Lorenc (1995) and van Leeuwen (2014), the observational error is composed of two individual errors on the Gaussian assumption: measurement error and representativeness error (the error caused by scale transformation is the major component of representativeness error). Both of them are equally important. Meanwhile measurement error is independent with scale transformation. Therefore, it is not necessary to introduce the measurement error when formulate the scale transformation in data assimilation, but that should be stated in the manuscript.

Assumption 2 and 3 mentioned that both model unit and scale of state do not change before the analysis step, but if mapping the state vector from state space to observation space, the scale transformation occurs, and the state $X$, observation $Y$ and observation operator $H(\cdot)$ are all dependent on scale. Then the error caused by scale transformation can be formulated with $Y(s_Y) - H(s_X, X(s_X))$.

**Changes in the manuscript:**

For the equation, explanation was appended: "Eq. (17) and Eq. (18) are the prior PDFs of state and observation with respect to scale in state space and observation space, respectively. Compared with the PDFs with respect to time, their expectations are equal to the value at the standard scale, and the variances depend on the differences between the standard scale and the scale in state or observation space. These two prior PDFs are introduced into the Bayesian theorem that reformulated by scale. "

For the issue on measurement error, we appended some necessary text in paragraph 3, Sect. 1 to clarify that the measurement error is not necessary to introduce in this paper: "The representativeness error and instrument error make up the observation error of data assimilation. Under the Gaussian assumption, they are independent of each other (Lorenc, 1995; van Leeuwen, 2014). This study will not introduce the instrument error when formulate the scale transformation in data assimilation." Here "measurement error" is replaced by "instrument error" to avoid being confused with "measure".

**6. Data assimilation and stochastic calculus**

*1. My main issue is that I don't understand the stochastic equation in scale space, neither where it comes from nor how it helps solve the representation error problem.*

*2. P13, 28: I'm not sure what happens here. Why is there a stochastic equation for H? Why not use the pdf of X directly to find the uncertainty in H?*

*3. I seem to miss something fundamental related to the stochastic equation in scale space and hope the authors can clarify that. What I would understand is a transformation from state to observation space, which might be modelled by a stochastic process. The rationale for that is that the state and/or observation subgrid processes are unknown and treated randomly. For that, only equations (19) and (20) are needed, although I still don't see why a stochastic differential equation is used to model this transformation, why not define it directly as a nonlinear function from state to observation space? Then the likelihood can be formulated.*

**Response:**

Thanks for your comments. We believe it is worth to use the stochastic approach to solve the representation error problem based on the following reasons:

First, using the Ito process and stochastic calculus is essentially consistent with the definitions of scale, scale transformation and geophysical variable. In Sect. 3.1, the scale was defined as the Lebesgue measure with respect to the observation footprint, scale transformation presents the change between two different scales, and geophysical variable is a real mapping function on $R$. All of them are associated with corresponding measure spaces $(\Omega, \mathcal{F}, \mu)$. Therefore, it is natural to regard the state space and observation space as two different measure spaces, respectively, and each element of state (or observation) vector can be seen as a geophysical variable that mapping the state (or observation) measure space onto $R$. Correspondingly, stochastic calculus, which is defined for integrals of random processes with respect to random processes, was adopted.

Second, understanding the scale transformation between different scales can be further improved by stochastic calculus. As we stated, to map the state vector from state space to observation space should consider the transformation of scales, heterogeneities and physical processes. This can be illustrated by Eq. (12).

$$V(s) = V_0 + \int_{s_0}^{s} \varphi(u) du + \int_{s_0}^{s} \sigma(u) dW(u)$$

Eq. (12) is integrated from $s_0$ to $s$, which presents the scale transformation. The integral term $\varphi(u)$ combines physical processes with heterogeneities.

Third, using stochastic calculus can formulate the scale-dependent errors. The results are presented in Eq. (23) and Eq. (25), which are derivate from Eq. (20). Therefore, we believe that all the equations are needed.

Further, compared with nonlinear function, the stochastic equation can offer a more general framework for scale issue and representativeness error. For example, we used the one-dimensional rule to simplify the scale transformation. However, that is the simplest situation. If the scale changes randomly, say, from an irregular footprint to another irregular footprint, the stochastic equation can offer a double-integral or multiple-integral to further study the scale issue, such as

$$V(x,y) = V_0 + \iint \varphi(x,y) dx dy + \iint \sigma(x,y) dW_1(x) dW_2(y)$$

where $W_1(x)$ and $W_2(y)$ are two independent Brownian Motion.

**Changes in the manuscript:**

To explicitly explain why using stochastic calculus is benefit to understanding the scale transformation and the representation error, we reorganized the section of "Discussion". In Sect. 4.1 titled "Discussion", we restated the advantages of this study:

"The reasons that the methodology focuses on a stochastic framework are: First, the stochastic data assimilation framework is essentially consistent with the conceptions of scale and scale transformation. Both of them are associated with corresponding measure spaces $(\Omega, \mathcal{F}, \mu)$. Therefore, it is natural to regard the state space and observation space as two different measure spaces, respectively, and each element of state (or observation) vector can be seen as a geophysical variable that mapping the state (or observation) measure space onto $R$. Correspondingly, as the integrals of random processes with respect to random processes, stochastic calculus was adopted ultimately. Second, using stochastic calculus can also formulate the errors caused by scale transformations. The study proceeds with and improves the understanding of representativeness error in terms of scale. Results did not only prove the conventional point that the uncertainties of these errors mainly depend on the differences between scales, but indicated that the first-order

differential of the nonlinear observation operator should also be incorporated in representativeness error. Last, stochastic calculus can be extended to meet a general scale transformation and formulate corresponding representativeness error. This was unattainable in previous work. For example, if the scale changes randomly, say, from an irregular footprint to another irregular footprint, the stochastic equation can offer a multiple-integral to present this kind of a scale transformation, such as

$$V(x,y) = V_0 + \int_{Y_0}^{Y} \int_{X_0}^{X} \varphi(x,y)dxdy + \int_{Y_0}^{Y} \int_{X_0}^{X} \sigma(x,y)dW_1(x)dW_2(y), \text{ where } W_1(x) \text{ and } W_2(y)$$

are two independent Brownian Motion."

**7. Other problems**

*1. P2, 15: The 'therefore' is not a logical consequence. What does 'them' refer to?*

**Response:**

Here the term "them" refer to "forcing data and system states".

**Changes in the manuscript:**

We have rewritten the Sect. 1 titled "introduction". According to your comments, this sentence was changed to "Geophysical data are typically observed by various Earth observations, therefore to update the observation data in a data assimilation system may result in scale transformations between observation space and system state space".

*2. P2, 16: The fact that observation operators are nonlinear and complex has in principle nothing to do with mismatches between model units and observation footprints. The logical connection is not clear.*

**Changes in the manuscript:**

We realized this problem and have made some revision according to your comments. Related text will be changed to "If observation operator is strongly nonlinear and complex, errors caused by scale transformation is even more serious".

*3. P2, 16: Model units should be defined a bit better (I know this is difficult because it is unclear what scales a model with a certain grid box size represents.)*

**Response:**

We agree with you that to well define the model units is a little difficult, and its definition may vary with branches of geoscience. So in the definition of scale (page 8, line 10, previous manuscript), we have exemplified the model units.

**Changes in the manuscript:**

According to the previous problem, the term "model units" was removed in this sentence.

We further defined the model unit in paragraph 1, Sect. 2 as "The model unit is a specified subspace where a geophysical variable evolves in the model space. It could be a point, a rectangular grid, or an irregular unit such as a response unit (watershed, landscape patch and so on)".

4. *P3, 4: Van Leeuwen also discussed the spatial and temporal resolution differences as giving rise to representation error, and Lorenc 1986 was the first to discuss the observation operator as source of an extra error on top of the measurement error in data assimilation.*

**Response:**

Thanks for the guidance. The related references have been cited in the revised manuscript.

**Changes in the manuscript:**

Related text will be changed to "An important concept that is related to scale transformation in data assimilation is "representativeness error", which is associated with the inconsistency in spatial and temporal resolutions between states, observations and operators (Lorenc, 1986; Janjić and Cohn, 2006; van Leeuwen, 2014; Hodyss and Nichols, 2015)".

New reference:

Lorenc A C. Analysis methods for numerical weather prediction. Quarterly Journal of the Royal Meteorological Society, 1986, 112(474):1177–1194.

5. *P3, 7: 'According to the above…' The land surface dynamical processes have not been discussed, so the logical link is missing.*

**Response:**

In the new manuscript, the Sect. 1 was rewrote, so the related text about land surface dynamics is removed.

6. *P4, 7: Data assimilation does not necessarily result in first and second moments, the solution is the full pdf. For instance, data assimilation can describe multimodal pdfs.*

**Response:**

Thanks for the guidance. In the new manuscript, the Sect. 1 was rewrote, and the incorrect sentences was also removed.

7. *P7: I assume phi(t) is deterministic in eq (1), it is the drift term, so 'transition probability' is perhaps misleading.*

**Response:**

We agree with you. The term *'transition probability'* will be changed to "drift rate" in the new version. Correspondingly, the term *'volatility'* is also changed to "volatility rate".

8. *P11, 15: 'the variable varies with scale because of the scale issue' is unnecessary vague. Perhaps remove this part of the sentence?*

**Response:**

The unnecessary vague part will be removed in the new version.

9. *P11, 31: What are the exact relations between M and p, and eta and q?*

**Response:**

As we stated, Eq. (6) (In the revised manuscript, it changed to Eq. (5)) is a discrete-time forecasting system, and Eq. (9) is a continuous-time Ito process that obtaining the prediction of state. So $p(t)$ can be regarded as a continuous-time version of $M$ that obtaining the state on the interval $[0, t]$. $\eta$ is the model error, and $q(t)$ can be seen as the error caused by evolution of time. So $q(t)$ is one part of $\eta$.

Generally $q(t)dW(t)$ is Gaussian (Apte, et al., 2007) and can hardly be used to study the representativeness error. That is also one of the reasons that we define the scale and formulate stochastic processes with respect to scale.

**Changes in the manuscript:**

We removed this paragraph, and made the new paragraph instead:

[revised manuscript text omitted]

---

## Editor Decision (ED2)

Dear Drs Feng Liu and Xin Li,

I have received two reviews of the latest version of your paper. The referees are the same as those of the previous versions. In particular, referee 2, who has again let his name known, is Prof. P. J. van Leeuwen.

Both referees give credit to the work you have done to further improve your paper (referee 1 writes that *The authors have worked hard*, and referee 2 that they *have done a fantastic job*). They both consider that your paper is now basically suitable for publication. They nevertheless add comments and suggestions, stressing in particular that they think the paper will still be difficult to understand by the potential readers. They write *It's not as easy to read as it should be* (referee 1) and *I still expect the readers get lost* (referee 2). As Editor, I certainly agree with them on that aspect.

Referee 1 does not make specific requests about the paper, and actually writes that the paper can be accepted in its present form. He/she nevertheless makes two suggestions which could in his/her mind improve the clarity of the paper. I leave it to you to follow or not these suggestions.

Referee 2 is much more specific. He writes that the paper can be accepted subject to minor revisions. But he also raises a number of specific questions and makes a number of specific requests, which are intended at simplifying and clarifying the paper, but actually go beyond 'minor' revisions. Please consider all of them. I stress one (his point 8). It would certainly be good that you push your example in subsection 3.3 to its end, by showing the probability distributions that you obtain with and without using a Stochastic Differential Equation for the scale transformation.

Please revise your paper taking into consideration all comments and suggestions of both referees. When you send the revised version, please give a point-by-point response to all of these comments and suggestions. Should you disagree with one particular comment, or decide not to follow one particular suggestion, please state precisely your reasons for that.

Referee 1 mentions that the English of the paper must be improved. It is certainly in part because of the English that both referees (as well as myself) have had difficulties in understanding your paper  (for an example, I do not personally understand what you mean by the sentence *the formulations of scale transformation can be extremely reduced*, on p. 12, l. 12). If you can have your paper checked by a native English speaker, that will be very good. But I mention that, if your paper is accepted, it will be submitted to a (free of charge for you) copy-editing, intended in particular at correcting the English (with of course further check by you).

…/…

In agreement with Referee 2's request, I am formally asking for *further review by Editor*, but I may nevertheless send your new version to Referee 2, not for a further review, but to ask him if he thinks you have responded properly to his various comments and suggestions.

I looked forward to receiving your new version,

With regards,

Olivier Talagrand
Editor
*Nonlinear Processes in Geophysics*

---

## Author Response (AR3)

**Authors' Responses to the Comments from the Editor and Referees**

**General reply**

We truly thank Prof. Olivier Talagrand, Prof. van Leeuwen and the anonymous referee for their very thoughtful and constructive comments, which improved our study significantly compared with the initial version of this paper. In the new manuscript, we made revisions according to the suggestions from Prof. van Leeuwen, and the research content was rechecked by a professional native English speaking team. We hope our intentions in this paper have been conveyed as clearly as possible.

**Responses to the Editor's Comments**

In the Editor's comments, Prof. Olivier Talagrand stated "I do not personally understand what you mean by the sentence *the formulations of scale transformation can be extremely reduced*, on p. 12, l. 12".

**Response:**

This conclusion was based on assumption 1, which introduced the one-dimensional rule as defined in Sect. 3.1 to formulate the scale transformation. According to assumption 1, scales vary in one-dimensional space. Then, the geophysical variable can be formulated as

$$dV = \varphi(s)ds + \sigma(s)dW(s), \qquad (R1)$$

where $s$ is a one-dimensional scalar.

Without assumption 1, this formula may become more complex because the scale will change arbitrarily in this situation. Then, a multiple integral should be introduced. For example,

$$dV = \varphi(x,y)dxdy + \sigma(x,y)dW_1(x)dW_2(y), \qquad (R2)$$

where $W_1(x)$ and $W_2(y)$ are two independent Brownian motions.

Therefore, Eq. (R2) is reduced to Eq. (R1) based on assumption 1.

**Changes in the manuscript:**

In the new manuscript, we made this conclusion clearer. The sentence "Additionally, the formulations of scale transformation can be extremely reduced." was replaced by "Therefore, based on assumption 1, the scale only varies in one-dimensional space, meaning that the corresponding scale transformation is an integral over one-dimensional space."

**Responses to Prof. van Leeuwen's Comments**

The point-by-point responses to the suggestions are as follows.

**1. Question 1**

*The authors have done a fantastic job in clarifying all mathematics used, and I find the manuscript very easy to read as the argument s are now presented in a logical order (for me). Furthermore, the results are very interesting. However, I suggest a modifications along the following lines.*

*I still expect the readers get lost. Let me explain what I think their reasoning could be:*
*They understand that a geophysical variable is a random variable as function of scale. The reason is two-fold, at a certain scale the variable can have different values in different parts of the domain, and our prior knowledge of the random variable value at a certain scale is limited. This, then leads to a pdf $p(V,s)$ at each time instant. This joint pdf gives the likely range of V given likely ranges of s. It's marginal $p\_s(V)$ gives the pdf of V at each scale value s. This is as far as they understand the reasoning until page 10 line 17.*

*However, at 18 on that page the authors introduce an SDE for $V(s)$. This description has a different interpretation from the one above. Now the uncertainty, so the width of the pdf at a certain scale depends on the scale at which the SDE is started. This, then, means that the pdf $p(V,s)$ depends also on a starting scale $s\_0$, and a different $s\_0$ leads to a different pdf. But, as far as they can see, $s\_0$ is arbitrary, so this cannot work.*

*Even if one defines $s\_0$ as the model unit scale or the observational footprint, there are still problems. Why would the uncertainty grow the further we are away from that scale? My guess would be that the pdf $p\_s(V)$ as defined above will be narrower at large scales, but this is not what the SDE will give us. Furthermore, the manuscript assumes $s\_0$ to be the same for observation and model, and it is unclear why that is a good choice.*
*So their question will be: why is modelling the random character of the geophysical variable as function of scale via an SDE the right thing to do? Why can't one work directly with $p(V,s)$ as defined above?*
*If one does use $p(V,s)$ as defined above for both state and observation they won't see the extra terms in*

*20-27 arising, apart from those terms already present in the representation error literature.*

*It is important that the authors make sure that this interpretation is avoided.*

**Response:**

Thank you for your comments. We believe that the formulation based on SDE will offer a more general and accurate framework to understand the elements in this study, such as the scale, scale transformation, geophysical variable and representativeness error. The results support this contention. We have no intention of replacing the classic expressions by these formulas; rather, we are trying to understand the related problems from a new perspective and discover something interesting.

Other reasons why we used SDE to formulate the scale transformation were presented in the second paragraph, Sect. 4.1 (previous manuscript). We also stated this problem in the response to question 4.

As to the standard scale $s_0$, which may lead readers to get lost, we revised most of our formulas based on your suggestions in questions 5 and 6. We believe that the result is clearer than that in the previous version, as described in our responses to questions 5 and 6.

**2. Question 2**

*A completely different interpretation, and the one that the authors have in mind I think, is that one does know the variable at one scale and needs to transform that to a value for the variable at another scale. This is indeed what is needed to compare model results with observations. The interpretation of equation (10) is now that it described how to transform the variable value from the model unit, or the observational footprint, to another scale where the comparison will be made. That will in general be a deterministic part to this (e.g. due to statistical knowledge about the averaging process), and a stochastic part due to the heterogeneity of the variable. This should be made more clear.*

**Response:**

We indeed formulated Eq. (10) by dividing it into a deterministic part ($\varphi(s)$) and a stochastic part ($\sigma(s)$). $\varphi(s)$ refers to the deterministic processes including spatial heterogeneities and physical variations among different scales, and $\sigma(s)$ is the uncertainty caused by the scale transformation.

**Changes in the manuscript:**

In the new manuscript, we further clarified the components of Eq. (10). The paragraph following Eq. (10) was revised and now states "The formulation of $\varphi(s)$ should consider the spatial heterogeneities and physical process variations among different scales, which together constitute the deterministic part of a geophysical variable. However, neither of them is well understood in a general theoretical study. Therefore, $\varphi(s)$ is conceptualized in Eq. (10). Particularly, if the study region is homogeneous, then the values of a variable that are observed at the same place are identical between the large scale and fine scale, and $\varphi(s)$ can be left out. Due to the integral over the space of Brownian motion, $\sigma(s)$ is the stochastic part, meaning that scale transformation produces uncertainties."

**3.   Comment 3**

*The assumption is that the scale at which the comparison is made is larger than both the scale at which we know the model and the observation. This makes this problem very different from the main issue in dealing with representation errors in data assimilation, in which we do know the model only at a much courser scale then the observations, and the observations only at the fine scale. Then a direct comparison is not possible and extra information is needed to solve this problem.*

*This assumption is crucial to mention, and, although a limitation, it still makes this an interesting problem to study.*

**Response:**

In our study, it is not necessary to identify which scale is smaller. Technically, we formulated a reversible process of scale transformation, meaning that two different scales can transform into each other. Practically, there is no size requirement for the scales of the observations or model units. One may compare the model unit of a catchment-scale model with the point scale or a micro-scale model with large-scale remote sensing images.

**4.   Question 4**

*The authors need to add a motivation why an SDE is a good model for changing scales. What does the drift term represent? Does it represent that we can use more model units or more observations from different positions to describe the scale change via averaging? And if so, the SDE leads to an increasing*

*stochastic variance for an increasing scale separation. Is that realistic if one can do spatial averaging with increasing scale?*

**Response:**

 The reasons why we used SDE to model the scale transformation are as follows: First, we defined the scale and scale transformation by some basic concepts of measure theory, so that the elements of data assimilation were also determined. For example, the geophysical variables are related to the Ito processes with respect to the scale, and the state and observation spaces correspond to the probability measure spaces. Therefore, stochastic calculus, as the integrals of Ito processes with respect to Ito processes, was ultimately adopted. Second, data assimilation based on SDE provides a general framework to study the errors that are caused by scale transformation. It is widely accepted that the understanding of the uncertainty can be improved by using stochastic models rather than deterministic models. In the Ito process-formed geographical variable, the drift rate represents the determined elements, and the volatility rate represent the uncertainties with the Brownian motion. Also, SDE can offer multiple Ito processes to study the cases in which the scale changes randomly. Third, more accurate descriptions of the errors can be obtained by this stochastic framework. We believe that the first-order differential of the nonlinear observation operator should also be incorporated in the representativeness error based on our research.

 Although averaging multi-observations or model units to estimate a large-scale geophysical variable is possible in theory, it may inevitably introduce additional errors. The multi-point upscaling technique may start by estimating a representative value (v1) in the scale (s1) in which the observations are measured; this step involves spatial averaging. Then, we used the single-point value v1 to predict value (v2) at a larger scale (s2) by scale transformation. This method introduces two types of error, which are caused by the imperfection of the spatial sampling technique and the scale transformation, respectively. Essentially, scale transformation is a nonlinear process, which is totally different from spatial averaging, because the latter is linear estimation. This is why the errors were introduced.

**Changes in the manuscript:**

 In the second paragraph, Sect. 4.1 (previous manuscript), we explained the motivation for choosing SDE to formulate the scale transformation and added a new reason to this paragraph: "Third, the error caused by the scale transformation was presented in a general form. The drift and quadratic

variation of the error were formulated by Eq. (21) and Eq. (22), respectively, and both defined the probability distribution space of $p(Y|X)$."

**5. Questions 5 and 6**

*The authors implicitly assume that both model unit and observational footprint are at scale $s\_0$. Otherwise the $Y\_0$ in equation (21) is unknown and the integrations in (21) from $s\_0$ cannot be performed, so the equations cannot be used in practise. I think this assumption is not necessary, as long as $s\_0$ for Y and $s\_0$ for X are smaller than the scale at which the comparison is made they can be different.*

*Further on point 5, I actually think the authors should reformulate page 13. Assume the model is available at scale $s\_x$ and the observation at scale $s\_y$. One cannot refer to scale $s\_0$ as it is smaller then both of them, so $x\_0$ and $Y\_0$ are not known, so cannot be used.*

*Let us first look at the case $s\_x < s\_y$. We first have to transform the model values from scale $s\_x$ to $s\_y$ before we can use H. This is done via the SDE. Equation (19) needs to be rewritten as (assuming $sigma\_x = sigma\_y = 1$, and $phi\_x = phi\_y = 0$ as in the paper):*

*$H(s\_y, X(s\_y))$ = equation (19) with $s\_0$ replaced by $s\_x$, and $s\_x$ by $s\_y$.*

*Equation (20) becomes*

*$Y(s\_y) - H(s\_y, X(s\_x)) = Y(s\_y) - H(s\_x, X(s\_x)) + \frac{1}{2} int\_s\_x^s\_y H\_{xx} du +$*

*$int sx^s\_y dW - int\_s\_x^s\_y H\_x dW$*

*This leads to a new drift and variance term easily extracted from the above.*

*In this way all reference to $Y\_0$ and $X\_0$ is disappeared, and all results become intuitive.*

*Now consider $s\_x > s\_y$. In that case one uses the SDE on Y, and finds:*

*$Y(s\_y) - H(s\_y, X(s\_x)) = Y(s\_x) + int\_s\_y^s\_x dW) - H(s\_y, X(s\_x))$*

*And again drift and variance can easily be extracted.*

*(In fact, I would suggest the authors keep the phi's and sigma's as it is good to have the full set of equations.)*

**Response:**

It is very kind of Prof. van Leeuwen to provide us with such a wonderful suggestion to make our results clearer. We are very pleased to accept this suggestion, and we also present a further re-solution of this problem.

It is true that $s_0$ is arbitrary and both $s_0$ and $Y_0$ are not known. Based on the methodology of data assimilation, in the analysis step, only $s_X, s_Y, X(s_X), Y(s_Y)$, and $H(\cdot)$ are available, and $X(s_Y)$, the true state in observation space, is not known. Therefore, $H\big(s_Y, X(s_Y)\big)$ cannot be formulated because it is also not known and cannot introduce the scale-related errors; thus, we only formulated $H\big(s_X, X(s_X)\big)$. Accordingly, we still only formulated $Y(s_Y) - H\big(s_X, X(s_X)\big)$, and we believe that the results will not change upon determining which is smaller between $s_X$ and $s_Y$.

We also keep $\varphi(s)$ and $\sigma(s)$ in the new results; therefore, the full set of equations is presented. However, Eq. (19)~Eq. (25) are not explicit formulations of $\varphi(s)$ and $\sigma(s)$.

**Changes in the manuscript:**

According to your suggestion, we reformulated Eq. (19)~Eq. (25) and removed Eq. (26) and Eq. (27). The related text has also been revised.

For **Eq. (19)**, we stated "Eq. (19) can also be rewritten by replacing $s_0$ with $s_Y$, namely

$$H\big(s_X, X(s_X)\big) = H\big(s_Y, X(s_Y)\big) + \int_{s_Y}^{s_X}\left[H_s\big(u, X(u)\big) + \tfrac{1}{2}H_{xx}\big(u, X(u)\big)\right]du + \int_{s_Y}^{s_X} H_x\big(u, X(u)\big)\, dW(u)".$$

**Eq. (20)** was changed from

$$Y(s_Y) - H\big(s_X, X(s_X)\big)$$

$$= Y_0 + \int_{s_0}^{s_Y} dW(u) - \left[H(s_0, X_0) + \int_{s_0}^{s_X} H_s\big(u, X(u)\big)\, du + \int_{s_0}^{s_X} H_x\big(u, X(u)\big)\, dW(u) + \tfrac{1}{2}\int_{s_0}^{s_X} H_{xx}\big(u, X(u)\big)\, du\right]$$

$$= Y_0 - H(s_0, X_0) + \int_{s_0}^{s_Y} dW(u) - \left[H\big(s_X, X(s_X)\big) - H\big(s_0, X(s_0)\big)\right] - \tfrac{1}{2}\int_{s_0}^{s_X} H_{xx}\big(u, X(u)\big)\, du - \int_{s_0}^{s_X} H_x\big(u, X(u)\big)\, dW(u)$$

$$= Y_0 - \left[H\big(s_X, X(s_X)\big) + \tfrac{1}{2}\int_{s_0}^{s_X} H_{xx}\big(u, X(u)\big)\, du\right] + \left\{\int_{s_0}^{s_Y} dW(u) - \int_{s_0}^{s_X} H_x\big(u, X(u)\big)\, dW(u)\right\}$$

to

$$Y(s_Y) - H\big(s_X, X(s_X)\big)$$

$$= Y(s_Y) - \left[H\big(s_Y, X(s_Y)\big) + \int_{s_Y}^{s_X}\left[H_s\big(u, X(u)\big) + \tfrac{1}{2}H_{xx}\big(u, X(u)\big)\right]du + \int_{s_Y}^{s_X} H_x\big(u, X(u)\big)\, dW(u)\right]$$

$$= Y(s_Y) - \left[H\big(s_X, X(s_X)\big) + \tfrac{1}{2}\int_{s_Y}^{s_X} H_{xx}\big(u, X(u)\big)\, du\right] + \left\{\int_{s_X}^{s_Y} H_x\big(u, X(u)\big)\, dW(u)\right\}.$$

**Eq. (21)** was changed from $Y_0 - \left[H\big(s_X, X(s_X)\big) + \tfrac{1}{2}\int_{s_0}^{s_X} H_{xx}\big(u, X(u)\big)\, du\right]$ to $Y(s_Y) - \left[H\big(s_X, X(s_X)\big) + \tfrac{1}{2}\int_{s_Y}^{s_X} H_{xx}\big(u, X(u)\big)\, du\right].$

**Eq. (22)** was changed from $(s_Y - s_0) + \int_{s_0}^{s_X} H_x^2\big(u, X(u)\big)\, du$ to $\int_{s_X}^{s_Y} H_x^2\big(u, X(u)\big)du.$

**Eq. (23)** was changed from $p(Y|X) = N\left(Y_0 - \left[H(s_X, X(s_X)) + \frac{1}{2}\int_{s_0}^{s_X} H_{xx}(u, X(u))\,du\right], (s_Y - s_0) + \int_{s_0}^{s_X} H_x^2(u, X(u))\,du\right)$ to $p(Y|X) = N\left(Y(s_Y) - \left[H(s_X, X(s_X)) + \frac{1}{2}\int_{s_Y}^{s_X} H_{xx}(u, X(u))\,du\right], \int_{s_X}^{s_Y} H_x^2(u, X(u))\,du\right)$.

**Eq. (24)** and **Eq. (25)** were changed from

$$Y(s_Y) - X(s_X) = \begin{cases} Y_0 - X(s_X) + W(s_Y) - W(s_X), s_Y > s_X \\ Y_0 - X(s_X) + W(s_X) - W(s_Y), s_X > s_Y \end{cases} \text{ and } p(Y|X) = N\{Y_0 - X(s_X), |s_Y - s_X|\}$$

to

$$Y(s_Y) - X(s_X) = Y(s_Y) - X(s_X) + \int_{s_X}^{s_Y} dW(u) \quad \text{and} \quad p(Y|X) = N\{Y(s_Y) - X(s_X), |s_Y - s_X|\}\quad,$$

respectively.

In this new version of Eq. (24), it may seem strange that $\int_{s_X}^{s_Y} dW(u)$ is appended to the right-hand side. This, we further explained "In Eq. (24), the integral $\int_{s_X}^{s_Y} dW(u)$ can be regarded as the noise based on the increment of the Brownian motion with respect to the scale, and its expectation equals zero".

Additionally, we removed the assumptions of $\varphi_X(s) = \varphi_Y(s) = 0$ and $\sigma_X(s) = \sigma_Y(s) = 1$. Therefore, Eq. (15)~Eq. (18) were changed from

$$\begin{cases} X(s_X) = X_0 + \int_{s_0}^{s_X} dW(s) \\ Y(s_Y) = Y_0 + \int_{s_0}^{s_Y} dW(s) \\ X \sim N(X_0, s_X - s_0) \\ Y \sim N(Y_0, s_Y - s_0) \end{cases} \text{ to } \begin{cases} X(s_X) = X_0 + \int_{s_0}^{s_X} \varphi_X(s)ds + \int_{s_0}^{s_X} \sigma_X(s)dW(s) \\ Y(s_Y) = Y_0 + \int_{s_0}^{s_Y} \varphi_Y(s)ds + \int_{s_0}^{s_Y} \sigma_Y(s)dW(s) \\ X \sim N\left(X_0 + \int_{s_0}^{s_X} \varphi_X(s)ds, \int_{s_0}^{s_X} \sigma_X^2(s)ds\right) \\ Y \sim N\left(Y_0 + \int_{s_0}^{s_Y} \varphi_Y(s)ds, \int_{s_0}^{s_Y} \sigma_Y^2(s)ds\right) \end{cases}.$$

**6. Suggestion 7**

*It would be good if the authors would discuss methods to find the sigma's and the phi's.*

**Response:**

It is important to find the expressions of the sigma and phi for explaining the scale transformation. However, this depends on the special case of particular studies, and these studies are beyond the scope of this paper.

In Eq. (10), if a geophyscial variable is transformed from $s_0$ to a new scale, $\varphi(s)$ indicates the dynamic variation (it is typically nonlinear) and the heterogeneity (if being quantified by the generally accepted method, namely, geostatistics, it is somewhat linear; however, it is essentially still nonlinear.).

To obtain $\varphi(s)$ requires a full understanding of the random process associated with the scale transformation of a geophysical variable. Taking a vegetation radiative transfer model (Jacquemoud et al., 2009) as an example, SAIL (Scattering by Arbitrary Inclined Leaves) is the canopy level (scale) model, and PROSPECT is predominant at the leaf level (scale). These two models were then coupled with each other as a new model, PROSAIL, to simulate the spectral reflectance field from the leaf to the canopy scale. PROSAIL can be regarded as a potential model to formulate $\varphi(s)$, but further study is definitely required to find an adequate function based on PROSAIL.

$\sigma(s)$ indicates the noise or disturbances caused by multiple trivial events. If the noise is totally random, a Gaussian random field is included in the stochastic process of a geographic variable, and $\sigma(s) = 1$. Otherwise a mixed model should be considered, which leads to $\sigma(s) \neq 1$. For example, if $\sigma(s) = W(s)$, which means that the noise is a product of two Gaussian random fields, then the noise $\int W(s)dW(s)$ is a martingale with mean zero and a quadratic variation equal to $\int W^2(s)ds$ (based on the results of Karatzas et al., 1991).

**7. Suggestion 8**

*Example 3.4 would benefit from specifying the phi's and sigma's and pushing the example to the end, perhaps showing pictures of the pdf's involved.*

**Response:**

As we stated in Suggestion 7, the exact expressions of the sigma and phi are based on special cases, for example, to determine the distribution function of a leaf area index or aerosol. In this example, the phi and sigma are mainly related to the optical depth $\tau$.

In the new manuscript, we specified $\sigma(s)$ and $\varphi(s)$ in $\tau(s)$ (Eq. 27) and $I(\tau(s))$ (Eq. 29), and the example was pushed to the end. However, the graphs of these pdfs cannot be presented because the formulas in this example are still abstract and complex. Currently, it seems impossible to obtain a picture of the pdfs.

**Changes in the manuscript:**

**Eq. (27)**: $d\tau(s) = \varphi_\tau(s)ds + \sigma_\tau(s)dW(s)$

**Eq. (29)**: $I\left(\tau(s)\right) = C \cdot \exp\left[\int\left(\frac{\sigma_\tau^2(s)}{2\mu^2} + \frac{\varphi_\tau(s)}{\mu}\right)ds + \int\left(\frac{\sigma_\tau(s)}{\mu}\right)dW(s)\right]$

To the end of this section, we appended:

"the results in Sect. 3.3 could easily be applied here. For example, Eq. (20) and Eq. (23) become

$$Y(s_Y) - H(s_X, X(s_X)) = I(\tau(s_Y)) - I(\tau(s_X)) + \frac{1}{2}\int_{s_X}^{s_Y}\frac{1}{\mu^2}I(\tau)du + \int_{s_X}^{s_Y}\frac{1}{\mu}I(\tau)dW(u), \quad (30)$$

$$p(Y|X) = N\left(I(\tau(s_Y)) - I(\tau(s_X)) + \frac{1}{2}\int_{s_X}^{s_Y}\frac{1}{\mu^2}I(\tau)du, \int_{s_X}^{s_Y}\frac{1}{\mu^2}I^2(\tau)du\right). \quad (31)$$

Then, the posterior PDF of the data assimilation can be determined by Eq. (27), (29) and (31)."

**8. Question 9**

*I don't see how 3.5 is relevant and/or needed. Why all this trouble, why the one-dimensional rule? Different variables will have different scaling rules, but as far as I can see one can follow the scaling rule ideas for each independent observation separately. What is the problem?*

**Response:**

Your question is reasonable. Most parts of this paper are focused on one-dimensional data assimilation and obtained results by introducing a single geophysical variable. Therefore, it is better to extend the framework to multi-dimensions. We proved that the scale based on the product measure also obeys a one-dimensional rule (see Sect. 3.1) because this rule is very important for the other formulas related to the scale. We agree that if all the variables are independent from each other, the scaling rules can be applied separately. Thus, if Sect. 3.5 is removed, there is no substantial effect on the main conclusions. We believe that a more challenging case is how to deal with correlated variables, and this more complicated problem might be studied in our future work.

**Changes in the manuscript:**

In the new version, we deleted Sect. 3.5, and all the problems have been removed.

**9. Question 10**

*As far as I can see, one can just define scale as the Lebesque integral, why do we need section 2.1?*

**Response:**

I think that to keep Sect. 2.1 is logically sound for our study because, first, it provides more basic knowledge about measure theory, such as the definitions of measure, measure space and σ-algebra, which can help readers understand the concepts of a Lebesgue measure and Lebesgue integral. Second, this basic knowledge also comprises important elements of the definitions of scale, scale transformation and

geophysical variable. Third, the state and observation spaces have a close relationship with the measure space; thus, we believe it is warranted to present a complete description of measure space. Lastly, it is necessary that the basic mathematical background of the paper be clarified precisely, as was required by the Editor.

**References**

[revised manuscript text omitted]

---

## Editor Decision (ED3)

Dear Drs Feng Liu and Xin Li,

I think your paper is now basically acceptable. I however also think that some further editing is necessary. I just give two examples.

1. Taken literally, eq. (20) makes no sense, since it would require that the sum of the two jntegrals on the rhs of the last line to be equal to zero. I do not think that results from a typographical error, since you give an explanation for a similar apparent inconsistency in eq. (24). But please explain.

2. On a less basic point, the Proof, ll. 22-25, p. 11, contains useless repetitions. Please shorten it (actually, the proven result is obvious).

Please go carefully through your paper, and correct all ambiguities or inconsistencies. When you send your final version, please mention clearly all changes you will have made.

With regards,

Olivier Talagrand
Editor
*Nonlinear Processes in Geophysics*

---

## Author Response (AR4)

**Authors' Responses to the Comments from Editor**

Dear Prof. Talagrand,

We truly thank you for your consideration and constructive comments. We have rechecked the paper carefully and further make some revisions including the presentations and responses to your comments. Besides, we changed the title of our institute from "Cold and Arid Regions Environmental and Engineering Research Institute" to "Northwest Institute of Eco-Environment and Resources", because it was renamed recently. Although they are the same institute, we are not sure the change is permitted by NPG. We can cancel it if this change is forbidden.

Best wishes,

Feng Liu

**Comment 1.**

*Taken literally, eq. (20) makes no sense, since it would require that the sum of the two jntegrals on the rhs of the last line to be equal to zero. I do not think that results from a typographical error, since you give an explanation for a similar apparent inconsistency in eq. (24). But please explain.*

**Response:**

Thanks for your comments. Your challenge is reasonable. Eq. (19) ~ Eq. (25) have revised a lot and we indeed made a mistake in the last manuscript. The formulas are not correct and need to be revised.

The last version is revised based on the referee's suggestion that $s_0$ should be replaced by $s_X$ or $s_Y$, and remove the assumption $\varphi = 0$ and $\sigma = 1$. However, we cancelled the assumption but did not revise these formulas accordingly, so Eq. (19) ~ Eq. (25) were not explicit formulations about $\varphi$ and $\sigma$.

**Changes in the manuscript:**

In the new manuscript, Eq. (19) ~ Eq. (24) are changed to

$$H\big(s_Y, X(s_Y)\big)$$

$$= H(s_0, X_0) + \int_{s_0}^{s_Y} H_s \, du + \int_{s_0}^{s_Y} H_x \sigma_X \, dW(u) + \int_{s_0}^{s_Y} H_x \varphi_X \, du + \frac{1}{2} \int_{s_0}^{s_Y} H_{xx} \sigma_X^2 \, du$$

$$= H(s_0, X_0) + \int_{s_0}^{s_Y} \left[ H_s + H_x \varphi_X + \frac{1}{2} H_{xx} \sigma_X^2 \right] du + \int_{s_0}^{s_Y} H_x \sigma_X \, dW(u).$$

$$= H\big(s_X, X(s_X)\big) + \int_{s_X}^{s_Y} \left( H_s + H_x \varphi_X + \frac{1}{2} H_{xx} \sigma_X^2 \right) du + \int_{s_X}^{s_Y} H_x \sigma_X \, dW(u) \tag{19}$$

$$Y(s_Y) - H\big(s_Y, X(s_Y)\big) = Y(s_Y) - \left[H\big(s_X, X(s_X)\big) + \int_{s_X}^{s_Y}\left(H_s + H_x\varphi_X + \tfrac{1}{2}H_{xx}\sigma_X^2\right)du\right] +$$

$$\int_{s_X}^{s_Y}(-H_x\sigma_X)\,dW(u). \tag{20}$$

$$Y(s_Y) - \left[H\big(s_X, X(s_X)\big) + \int_{s_X}^{s_Y}\left(H_s + H_x\varphi_X + \tfrac{1}{2}H_{xx}\sigma_X^2\right)du\right]. \tag{21}$$

$$\int_{s_X}^{s_Y} H_x^2\sigma_X^2\,du. \tag{22}$$

$$p(Y|X) = N\left(Y(s_Y) - \left[H\big(s_X, X(s_X)\big) + \int_{s_X}^{s_Y}\left(H_s + H_x\varphi_X + \tfrac{1}{2}H_{xx}\sigma_X^2\right)du\right], \int_{s_X}^{s_Y} H_x^2\sigma_X^2\,du\right). \tag{23}$$

where $\varphi$ and $\sigma$ are kept in these formulas. And for simplicity, Eq. (24) is still based on the assumption of $\varphi = 0$ and $\sigma = 1$:

$$Y(s_Y) - X(s_Y) = Y(s_Y) - X(s_X) - \int_{s_X}^{s_Y} dW(u). \tag{24}$$

**Comment 2.**

*On a less basic point, the Proof, ll. 22-25, p. 11, contains useless repetitions. Please shorten it (actually, the proven result is obvious).*

**Response:**

The lemma is obvious and easy to be proved, so we agree with you.

**Changes in the manuscript:**

The proof was removed, and we explain it in the following paragraph:

[revised manuscript text omitted]

---

## Editor Decision (ED4)

Dear Drs Feng Liu and Xin Li,

After reading this new version of your paper, I still have a few suggestions for modifications (page and line numbers refer below to the version of the paper in which you have included explicitly, in red, your latest modifications).

1. P. 6, Eq. (4). *x(t)* is here an Ito process, just as *I(t)* before. I suggest you use the same notation.

2. P. 6, l. 30. … *the well-accepted Bayesian theory of data* assimilation.

3. P. 13, l. 4. The two equations on this line state the same result for two different arguments $s_X$ and $s_Y$. Actually, in view of the eqs (15) and (16) that follow, I think the corresponding sentence is useless. The sentence *These formulas prove* … can be put after Eqs (15) and (16).

4. P. 13, l. 24. I understand *x* in *H(s, x)* is the same thing as *X* before. Use the same notation.

5. It seems that, from the last line of p. 14, $\sigma_X$ is assumed to be equal to 1. Why not keep an explicit $\sigma_X$ ?

6. P. 16, l. 1. Why not put an absolute value for the variance of *X* (and remove the sentence *It should be noted* …) ?

7. And, finally, change the first line of acknowledgments to *We thank the editor, Dr Talagrand, and* …

With regards,

Olivier Talagrand
Editor
*Nonlinear Processes in Geophysics*

---

## Author Response (AR5)

Dear Prof. Talagrand,

Thank you for your hard working and constructive suggestions, which significantly improved our study. We have made the corresponding revisions. The point-by-point responses to the suggestions are in the following text.

Best wishes,

Feng Liu

**Authors' Responses to the Comments from Editor**

**Comment 1.**

*P. 6, Eq. (4). x(t) is here an Ito process, just as I(t) before. I suggest you use the same notation.*

**Comment 2.**

*P. 6, l. 30. ... the well-accepted Bayesian theory of data assimilation.*

**Comment 3.**

*P. 13, l. 4. The two equations on this line state the same result for two different arguments sX and sY. Actually, in view of the eqs (15) and (16) that follow, I think the corresponding sentence is useless. The sentence These formulas prove ... can be put after Eqs (15) and (16).*

**Comment 6.**

*P. 16, l. 1. Why not put an absolute value for the variance of X (and remove the sentence It should be noted ...) ?*

**Comment 7.**

*And, finally, change the first line of acknowledgments to We thank the editor, Dr Talagrand, and ...*

**Response:**

Thanks for your suggestions. We have adopted all the above suggestions. Please find the detail information in the new manuscript.

**Comment 4.**

*P. 13, l. 24. I understand x in H(s, x) is the same thing as X before. Use the same notation.*

**Response:**

Actually $x$ in *H(s, x)* is different from *X*. Here $x$ is an argument of *H*. Although $x$ always indicates the state vector, it is not the same as an instance of state vector *X*. Furthermore, the first- and second-order derivatives of *H* are notated as $H_x$ and $H_{xx}$, respectively, where the notation $x$ cannot be replaced by *X*.

But we can use another letter to replace $x$ in order not to cause confusion. As comment 1, $x$ was replaced by *I*. Therefore *H* can also be written as *H(s, I)*, and its first- and second-order derivatives are $H_I$ and $H_{II}$.

**Changes in the manuscript:**

In the new manuscript, all the related notations were changed accordingly. But in Sect. 3.4, SRTE is still defined as $H\big(s, x(s)\big)$, because we used $I(\tau)$ to present the state vector radiation intensity ($I$ is the widely accepted notation for radiation intensity in a radiative transfer equation). In this section we believed that $x$ and $I(\tau)$ cannot be mixed up.

**Comment 5.**

*It seems that, from the last line of p. 14, $\sigma X$ is assumed to be equal to 1. Why not keep an explicit $\sigma X$?*

**Response:**

The reasons that we did not keep an explicit $\sigma_X$ are, on the one hand, the formulas with explicit $\sigma$ and $\varphi$ were given in Eqs. (21) and (23), and on the other hand, in Eqs. (24) and (25), we tried to deduce the simplest version of Eqs. (21) and (23) to present the direct relationship between the scale transformation and $p(Y|X)$, where $\sigma$ and $\varphi$ should not be included in these equations. So the state *X* is assumed to be only with the scale-dependent Gaussian noises, resulting $\sigma_X$=1 and $\varphi_X$=0.